# Large Language Model-driven Large Neighborhood Search for Large-Scale MILP Problems

## Abstract

Large Neighborhood Search (LNS) is a widely used method for solving large-scale Mixed Integer Linear Programming (MILP) problems. The effectiveness of LNS crucially depends on the choice of the search neighborhood. However, existing strategies either rely on expert knowledge or computationally expensive Machine Learning (ML) approaches, both of which struggle to scale effectively for large problems. To address this, we propose LLM-LNS, a novel Large Language Model (LLM)-driven LNS framework for large-scale MILP problems. Our approach introduces a dual-layer self-evolutionary LLM agent to automate neighborhood selection, discovering effective strategies with scant small-scale training data that generalize well to large-scale MILPs. The inner layer evolves heuristic strategies to ensure convergence, while the outer layer evolves evolutionary prompt strategies to maintain diversity. Experimental results demonstrate that the proposed dual-layer agent outperforms state-of-the-art agents such as FunSearch and EOH. Furthermore, the full LLM-LNS framework surpasses manually designed LNS algorithms like ACP, ML-based LNS methods like CL-LNS, and large-scale solvers such as Gurobi and SCIP. It also achieves superior performance compared to advanced ML-based MILP optimization frameworks like GNN&GBDT and Light-MILPopt, further validating the effectiveness of our approach.

## 1 Introduction

Mixed Integer Linear Programming (MILP) is a versatile and widely used mathematical framework for solving complex optimization problems across various domains, including transportation management (Klanšek, 2015), bin packing (Fleszar, 2022), and production planning (Adrio et al., 2023). MILPs are challenging to solve efficiently due to their NP-hard nature (Kim et al., 2021) and the exponential growth of the search space as problem size increases (Vázquez et al., 2018). To address these challenges, researchers have developed two primary approaches (Zhang et al., 2023): exact algorithms, such as branch-and-bound, and heuristic-based approximation methods.

While exact algorithms like branch-and-bound (Boyd & Mattingley, 2007; Morrison et al., 2016) are effective for small to medium-sized problems, they struggle with the computational demands of larger instances. This has led to the rise of heuristic methods, particularly Large Neighborhood Search (LNS) (Ahuja et al., 2002; Mara et al., 2022), which iteratively improves solutions by destroying and repairing parts of the current solution, allowing for exploration of large neighborhoods without full re-optimization (Song et al., 2020; Ye et al., 2023a). However, LNS performance depends heavily on neighborhood selection, which is often hand-crafted and requires significant domain expertise. Designing these operators can be labor-intensive and prone to *cold-start issues*, where limited prior knowledge is available to guide the search (Zhang et al., 2023).

In recent years, machine learning (ML) techniques, including reinforcement learning (Wu et al., 2021; Song et al., 2020) and imitation learning (Sonnerat et al., 2021; Nair et al., 2020), have been applied to automate the design of neighborhood selection strategies. These methods aim to learn heuristic strategies from training datasets, reducing reliance on expert knowledge and allowing the algorithms to adapt to new, homogeneous instances. However, ML-based LNS approaches come with their own challenges. For reinforcement learning, *slow convergence* is a common issue (Beggs,

2005), particularly in large-scale MILP problems, due to the vast search space and the need for extensive exploration before identifying effective strategies. On the other hand, imitation learning requires large amounts of high-quality, labeled data, which can be *computationally expensive* to generate using expert algorithms (Huang et al., 2023b). As a result, both hand-crafted and ML-based methods struggle to efficiently solve large-scale MILP problems.

The rise of Large Language Models (LLMs) offers a promising solution to these challenges. Unlike traditional hand-crafted methods, LLMs come pretrained with vast general knowledge, allowing them to reason about complex tasks and learn problem structures with minimal training data, thus avoiding *cold-start issues*. Additionally, LLMs can adapt to new problems through interactive reasoning, reducing the need for extensive exploration and addressing the *slow convergence* of reinforcement learning. Furthermore, LLMs can dynamically generate heuristic strategies without relying on large labeled datasets, which significantly reduces the *computational overhead* typically associated with imitation learning (Yang et al., 2024; Lange et al., 2024). While LLMs have shown potential in generating strategies for combinatorial optimization problems(Ye et al., 2024; Elhenawy et al., 2024), they often lack the problem-specific refinement needed to produce efficient heuristics without additional guidance (Plaat et al., 2024). Approaches like FunSearch (Romera-Paredes et al., 2024) and Evolution of Heuristic (EOH) (Liu et al., 2024) combine LLMs with evolutionary algorithms (Simon, 2013), but rely on fixed strategies, limiting solution diversity and leading to poor convergence due to insufficient directionality. This underscores the need for a more adaptive framework to fully harness LLMs for large-scale MILP problems.

In this paper, we propose LLM-LNS, a novel Large Language Model-driven Large Neighborhood Search framework designed specifically for solving large-scale MILP problems, which can discover effective neighborhood selection strategies for LNS with scant small-scale training data that generalize well to large-scale MILPs. Our key innovations are as follows:

- **Dual-layer Self-evolutionary LLM Agent**: We propose a novel LLM agent with a dual-layer self-evolutionary mechanism for automatically generating heuristic strategies. The inner layer evolves both thoughts and code representations of heuristic strategies, ensuring convergence, while the outer layer evolves evolutionary prompt strategies to maintain diversity, preventing the search process from getting trapped in local optima.

- **Differential Memory for Directional Evolution**: We introduce differential evolution in the agent to guide both crossover and variation. By feeding the fitness values of parent strategies back into the LLM, we leverage its memory to learn how to evolve from less effective to more effective strategies. This feedback mechanism enables the LLM to act as an optimizer, identifying promising directions and leading to more efficient improvements.

- **Application to Neighborhood Selection in LNS**: We apply the proposed dual-layer LLM agent to the neighborhood selection strategy generation in LNS. By utilizing only a small amount of training data from small-scale problems, the LLM agent can discover new neighborhood selection strategies that generalize well to large-scale MILP problems.

- **Comprehensive Experimental Validation**: We validate the effectiveness of our proposed LLM-LNS at two levels. First, we test its agent's performance on heuristic generation tasks of combinatorial optimization problems, demonstrating its superiority over state-of-the-art methods such as FunSearch (Romera-Paredes et al., 2024) and EOH (Liu et al., 2024). Second, we evaluate its performance on large-scale MILP problems, where it outperforms traditional LNS methods (e.g., ACP (Ye et al., 2023a)), ML-based LNS methods (e.g., CL-LNS (Huang et al., 2023b)), and leading solvers like Gurobi (Gurobi Optimization, LLC, 2023) and SCIP (Maher et al., 2016). Furthermore, our proposed LLM-LNS surpasses modern ML-based optimization frameworks for large-scale MILP, such as GNN&GBDT (Ye et al., 2023c) and Light-MILPopt (Ye et al., 2023b). These results confirm the effectiveness of our proposed LLM-LNS in solving large-scale optimization problems.

## 2 RELATED WORK

### 2.1 MIXED INTEGER LINEAR PROGRAMMING

Mixed Integer Linear Programming (MILP) problems represent a class of combinatorial optimization problems characterized by a linear objective function subject to a set of linear constraints, where

some or all decision variables are restricted to integer values. An MILP can be defined as follows:

$$\min_x c^T x, \quad \text{subject to} \quad Ax \le b, \, l \le x \le u, \, x_i \in \mathbb{Z}, \, i \in \mathcal{I}, \tag{1}$$

where $x$ represents the decision variables, with $n \in \mathbb{Z}$ denoting the dimensionality of the integer variables and $l, u, c \in \mathbb{R}^n$ corresponding to the lower bounds, upper bounds, and coefficients of the decision variables, respectively. The matrix $A \in \mathbb{R}^{m \times n}$ and the vector $b \in \mathbb{R}^m$ define the linear constraints of the problem. The set $\mathcal{I} \subseteq \{1, 2, \ldots, n\}$ denotes the indices of variables that are constrained to integer values. A feasible solution to the MILP problem satisfies all constraints, and the optimal solution minimizes the objective function value. (Artigues et al., 2015; Pisaruk, 2019)

## 2.2 Large Neighborhood Search

Large Neighborhood Search (LNS) is a widely used heuristic for solving MILP problems. It iteratively improves solutions by exploring predefined neighborhoods around a current solution. However, the effectiveness of LNS heavily relies on the neighborhood selection strategy, as poor choices can lead to stagnation in local optima.

Several approaches have been proposed to address this challenge. One common method is random-LNS (Song et al., 2020), which randomly partitions integer variables into disjoint subsets and optimizes one subset in each iteration while fixing the others. However, random-LNS uses a fixed neighborhood size and overlooks correlations between decision variables, limiting its performance. To overcome these drawbacks, the Adaptive Constraint Partitioning (ACP) framework (Ye et al., 2023a) introduces a dynamic strategy that adjusts the neighborhood size, optimizing all decision variables associated with randomly selected constraints in each iteration. This ensures that highly correlated variables are optimized together, improving performance. Similar strategies have been explored in other works (Huang et al., 2023a; Han et al., 2023), but they still rely on manually designed heuristics, requiring expert knowledge and lacking adaptability to new problem instances.

To address this limitation, machine learning methods have been applied to automate neighborhood selection. Reinforcement learning (RL) approaches define reward functions based on solution improvements, allowing models to learn promising neighborhoods through interaction with the problem (Wu et al., 2021; Song et al., 2020; Nair et al., 2020). Imitation learning, on the other hand, uses large amount of large-scale sampling (Huang et al., 2023b; Zhou et al., 2023) or expert algorithms (Sonnerat et al., 2021) to guide the selection process. While these techniques reduce reliance on handcrafted strategies, RL struggles with convergence in large-scale MILP problems, and imitation learning requires extensive sampling, making it computationally expensive. This highlights the need for more efficient, automatically designed neighborhood selection strategies.

## 2.3 Large Language Model for Heuristic Strategy Design

The rise of Large Language Models (LLMs) has opened new possibilities for generating heuristic strategies to solve combinatorial optimization problems (Yang et al., 2024; Lange et al., 2024). LLMs excel at generating high-level ideas and reasoning over complex tasks, but they often lack problem-specific knowl-

Table 1: Comparison of Features Between Fun-Search, EOH, and LLM-LNS.

| | FunSearch | EOH | **LLM-LNS** |
|---|---|---|---|
| Heuristic Evolution | ✓ | ✓ | ✓ |
| Thought Evolution | × | ✓ | ✓ |
| Prompt Evolution | × | × | ✓ |
| Directional Evolution | × | × | ✓ |

edge, limiting their ability to create effective heuristics without additional guidance (Plaat et al., 2024). To overcome these limitations, recent works have integrated LLMs with evolutionary algorithms (EA) to iteratively refine heuristics.

FunSearch (Romera-Paredes et al., 2024) is a notable attempt that combines LLMs with evolutionary frameworks. FunSearch uses LLMs to generate functions, which are then evolved through an evolutionary search process. This approach has demonstrated success in outperforming hand-crafted algorithms on specific optimization problems. However, FunSearch is computationally expensive, often requiring millions of LLM queries to identify effective heuristic functions, which limits its practicality in many real-world applications. A more recent approach, Evolution of Heuristic (EOH) (Liu et al., 2024), builds on the strengths of LLMs and evolutionary computation while addressing some of FunSearch's limitations. EOH introduces a novel evolutionary paradigm where heuristics,

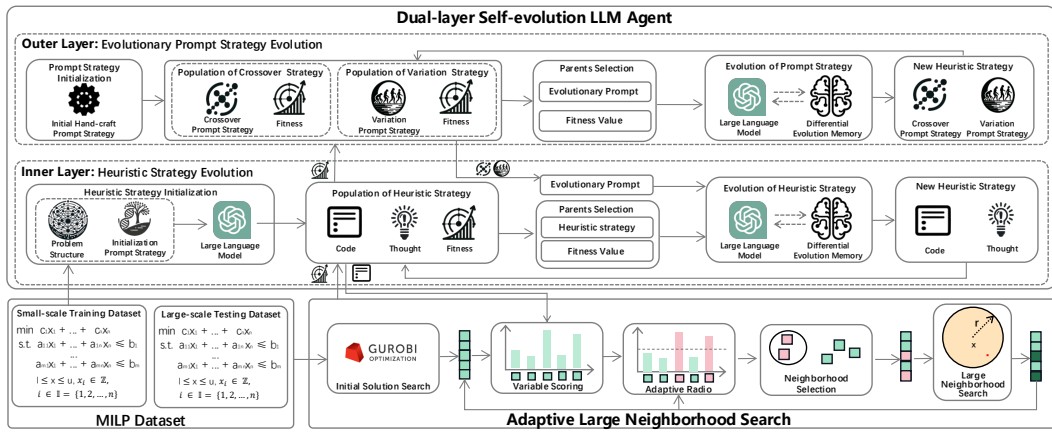

Figure 1: An overview of the proposed LLM-LNS framework. The framework consists of a dual-layer self-evolutionary LLM agent for solving large-scale MILP problems. In the outer layer, evolutionary prompt strategies are generated and passed to the inner layer, where heuristic strategies are evolved. A differential memory mechanism uses fitness feedback to refine these strategies across iterations. The refined strategies are fed into the Adaptive Large Neighborhood Search process, which iteratively improves solutions with the support of solvers like Gurobi.

represented as natural language "thoughts," are translated into executable code by LLMs. These thoughts and their corresponding code are evolved within an EA framework, enabling the efficient generation of high-performance heuristics. As shown in Table 1, while FunSearch and EOH have advanced the integration of LLMs with evolutionary algorithms, they still have limitations. All methods focus on *Heuristic Evolution* for generating strategies, but FunSearch evolves only at the code level and lacks *Thought Evolution*. Meanwhile, EOH incorporates Thought Evolution but uses fixed evolutionary strategies, lacking *Prompt Evolution* to enhance solution diversity. Additionally, both methods lack *Directional Evolution*, where crossover operations are guided by differential memory to improve efficiency and adaptability. These limitations reduce their ability to guide the search effectively, often leading to premature convergence. These challenges highlight the need for more adaptive frameworks to fully harness LLMs in large-scale optimization tasks.

## 3  METHOD

In this section, we introduce LLM-LNS, a Large Language Model-driven Large Neighborhood Search framework designed to solve large-scale MILP problems. As shown in Figure 1, the framework is composed of two main components: a **Dual-layer Self-evolutionary LLM Agent** and a **Adaptive Large Neighborhood Search** process.

### 3.1  DUAL-LAYER SELF-EVOLUTIONARY LLM AGENT

The Dual-layer Self-evolutionary LLM Agent is the core component of our framework, responsible for generating and evolving heuristic and prompt strategies. The **Dual-layer Self-evolutionary Structure** consists of an Inner Layer that evolves heuristic strategies to accelerate convergence, and an Outer Layer that evolves evolutionary prompt strategies to enhance diversity in heuristic generation. Another key innovation is the incorporation of **Differential Memory for Directional Evolution**, which accelerates convergence by learning the direction of improvement from less effective strategy to better ones. Together, these innovations ensure a balance between exploration and exploitation, significantly improving the efficiency and preventing stagnation in local optima.

### 3.1.1  DUAL-LAYER SELF-EVOLUTIONARY STRUCTURE

The Dual-layer Self-evolutionary Structure is the core component of the LLM-LNS framework. It is designed to evolve both evolutionary prompt strategies and heuristic strategies in a synergistic

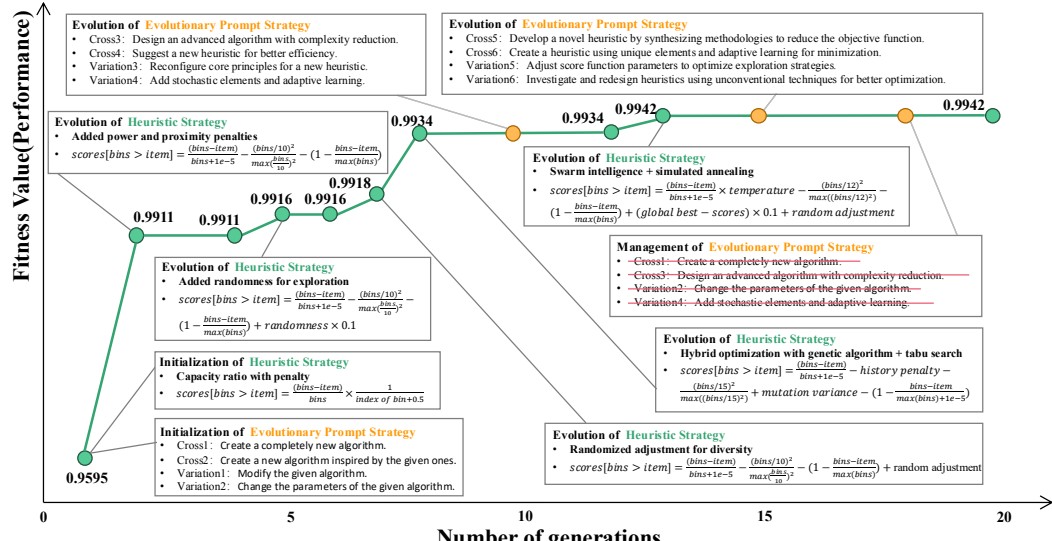

Figure 2: Evolution of Dual-layer Self-evolutionary LLM Agent for online bin packing. We outline the key thoughts and the corresponding code snippets of the best heuristics produced in some generations during the evolution of heuristic strategies. Additionally, we highlight the evolution of evolutionary prompt strategies, which dynamically adapt the prompt strategies to guide the LLM in generating more effective and diverse heuristics. Both strategies contribute to the overall improvement in performance and convergence throughout the evolutionary process.

manner, leveraging LLMs for automated heuristic design and refinement. This dual-layered structure mimics the heuristic development process of human experts, ensuring a balance between exploration and exploitation throughout the search process.

**Inner Layer: Heuristic Strategy Evolution.** The Inner Layer focuses on evolving heuristic strategies, which consist of both natural thought and corresponding code implementations, with an emphasis on *convergence*. Key aspects of Inner Layer, as illustrated in Figure 1 and Figure 2, include:

- *Initialization of Heuristic Strategies*: The initial set of heuristics is generated by feeding the structural information from small-scale training problems, along with an initialization prompt strategy, into the LLM. This produces the first generation of heuristic strategies. For example, at generation 1, a basic heuristic is initialized with a fitness value of 0.9595, based on a capacity ratio with penalty calculation, and is expressed both in natural language thought and executable code.

- *Evolution of Heuristic Strategies*: In each generation, new heuristic strategies are evolved by selecting parent strategies from the current heuristic population. As detailed in Appendix G, strategies with higher fitness values are more likely to be selected as parents. These parents are then combined with evolutionary strategies, selected from the Outer Layer's population of prompt strategies (e.g., crossover or variation prompts), to guide the LLM in generating new offspring strategies. For instance, at generation 5, randomness is introduced for exploration, achieving a fitness value of 0.9916. By generation 8, the evolution process incorporates hybrid optimization techniques, such as genetic algorithms combined with tabu search, resulting in a fitness value of 0.9934. This iterative process enables the LLM to continually refine strategies and explore new solution spaces.

- *Evaluation and Final Selection*: After new heuristic strategies are generated, they are evaluated by integrating them into the Adaptive Large Neighborhood Search process, where each heuristic is applied to solve small-scale instances from the training dataset. The performance of each strategy is measured by its objective function value, which serves as its fitness score. After multiple iterations of evolution and evaluation, the best-performing heuristic strategies are identified based on their fitness. By generation 20, advanced techniques like swarm intelligence and simulated annealing are incorporated, and the final best

strategy—achieving a fitness value of 0.9942—is selected for output. This iterative evaluation ensures that the framework converges to the most effective heuristic.

**Outer Layer: Evolutionary Prompt Strategy Evolution.** The Outer Layer focuses on evolving evolutionary prompt strategies, which guide the LLM in generating new heuristic strategies. The emphasis in this layer is on *exploration* to maintain diversity and prevent premature convergence in the heuristic strategy population. The key stages of Outer Layer, as illustrated in Figure 1 and Figure 2, include:

- *Initialization of Prompt Strategies*: The initial set of evolutionary prompt strategies is hand-crafted and designed to perform basic crossover and variation operations, instructing the LLM on how to combine or modify existing heuristic strategies in the inner layer. For example, at generation 1, basic prompt strategies like Cross1 and Cross2 are set, which help the LLM generate new heuristic strategies by recombining or tweaking existing ones.

- *Evolution of Prompt Strategies*: As the evolution progresses, more complex prompt strategies are introduced to address stagnation in the heuristic population. Specifically, if the top-$l$ individuals in the heuristic population remain unchanged for $t$ consecutive generations, we infer that the evolution may have converged to a local optimum. This triggers the evolution of new prompt strategies. As shown in Figure 2, signs of stagnation were observed in both the 10th and 15th generations. In response, new prompt strategies were generated to overcome the local optimality issue. At generation 10, prompts such as Cross3 and Cross4 were designed to enhance efficiency and reduce algorithmic complexity. By generation 15, even more advanced strategies like Variation5 and Variation6 were introduced, incorporating stochastic elements and adaptive learning to increase diversity and explore new heuristic possibilities. This systematic evolution of prompt strategies helps ensure that the heuristic population continues to evolve and does not get trapped in local optima.

- *Evaluation and Management of Prompt Strategies*: To ensure the efficiency and effectiveness of the prompt strategy population, each prompt strategy is evaluated based on the performance of the heuristic strategies it generates. Specifically, for each prompt strategy, the top-$k$ performing heuristic strategies it produces are tracked, and the average fitness score of these heuristics is used as the fitness score for the prompt strategy itself. This fitness-based evaluation allows us to manage the prompt population and control its size. As the number of prompt strategies increases over generations, underperforming strategies are pruned to prevent excessive growth and focus on the most effective strategies. For example, by generation 18, several underperforming prompt strategies (e.g., the four worst-performing strategies) are removed, as shown in Figure 2. This pruning process ensures that only the most effective prompt strategies continue to evolve, maintaining both diversity and efficiency in the evolutionary process. For parameter details, see Appendix A.

The synergy between the **Inner Layer** and **Outer Layer** drives rapid evolution of effective heuristics and novel evolutionary prompt strategies, as shown in Figure 2. Early generations focus on basic principles, but with the introduction of advanced prompt strategies, such as complexity reduction and adaptive learning, the system quickly adapts to overcome local optima. Notably, the sharp performance improvements between generations 5 to 15 demonstrate the framework's ability to autonomously discover and refine creative strategies, leading to continuous enhancements in heuristic performance. This dual-layered approach ensures efficient exploration and exploitation, enabling the LLM-LNS framework to tackle large-scale problems with minimal human intervention.

### 3.1.2 DIFFERENTIAL MEMORY FOR DIRECTIONAL EVOLUTION

In our Dual-layer Self-evolutionary LLM Agent, both heuristic strategies and evolutionary prompt strategies evolve through a process that incorporates *Differential Memory for Directional Evolution*. This mechanism allows the LLM to leverage the fitness history of strategies, learning from the differences between higher- and lower-performing strategies to guide the generation of improved candidates. Differential memory enables the LLM to act as both a generator and an optimizer, dynamically refining strategies over successive generations.

At each generation $t$, the LLM is provided with a set of $m$ *strategy-thought-fitness* tuples:

$$S^{(t)} = \{\langle H_i^{(t)}, \text{thought}_i, f(H_i^{(t)}) \rangle\}_{i=1}^m, \tag{2}$$

---

**Algorithm 1** Adaptive Large Neighborhood Search (ALNS)

---

**Require:** Initial solution $\mathbf{x}_0$, initial neighborhood size $k$, time limit $T$, threshold $\epsilon$, iteration limit $p$, minimum and maximum neighborhood sizes $k_{\min}, k_{\max}$, decision variable count $n$, adjustment rate $u\%$ (percentage)
 1: Initialize solution $\mathbf{x} \leftarrow \mathbf{x}_0$, set time $t \leftarrow 0$
 2: **while** $t < T$ **do**
 3:     Compute variable scores using LLM agent
 4:     Select top-$k$ variables to form neighborhood
 5:     Solve subproblem using solver within neighborhood
 6:     Update solution $\mathbf{x}$ if improved
 7:     **if** time spent in neighborhood exceeds limit **then**
 8:         $k \leftarrow \max(k_{\min}, k - \lceil u\% \cdot n \rceil)$                    ▷ Reduce search radius by $u\%$ of $n$
 9:     **else if** improvement in objective $< \epsilon$ for $p$ consecutive iterations **then**
10:         $k \leftarrow \min(k_{\max}, k + \lceil u\% \cdot n \rceil)$                    ▷ Expand search radius by $u\%$ of $n$
11:     **end if**
12:     Update time $t$
13: **end while**
14: **return** $\mathbf{x}$

---

where $H_i^{(t)}$ represents the $i$-th parent heuristic strategy selected for this generation, thought$_i$ is its corresponding natural language description, and $f(H_i^{(t)})$ is its fitness score. The size of $S^{(t)}$ is $m$, which is a predefined parameter representing the number of parent strategies used in a single evolutionary operation. These tuples encapsulate both the structural and performance information of the selected parent strategies, providing the necessary context for generating offspring strategies.

To generate the next generation of strategies $H^{(t+1)}$, the LLM employs a *meta-prompt* $p_{\text{meta}}$, which combines two key components: a directive $p_{\text{learn}}$ that instructs the LLM to learn from the differences between higher- and lower-performing strategies, emphasizing traits that contribute to higher fitness; and an *evolutionary prompt strategy* $p_{\text{evo}}$, provided by the Outer Layer, which specifies the goals and rules for the evolutionary operation, such as crossover, mutation, or hybrid operations. The generation process can be formalized as:

$$H_i^{(t+1)} = \mathcal{M}(p_{\text{meta}} \| S^{(t)}), \tag{3}$$

where $\mathcal{M}$ is the LLM model, $p_{\text{meta}} = \langle p_{\text{learn}}, p_{\text{evo}} \rangle$ is the meta-prompt, and $S^{(t)}$ represents the strategy-thought-fitness tuples from the current generation. By integrating these components, the LLM generates new strategies $H^{(t+1)}$ that are informed by past evolutionary performance and aligned with the objectives defined by the Outer Layer. This iterative feedback-refinement loop ensures that the LLM dynamically balances exploration and exploitation. Differential memory accumulates across generations, enabling the LLM to focus on areas of the search space that demonstrate promise while avoiding stagnation in local optima. The result is an increasingly proficient evolution process, accelerating convergence toward optimal solutions while maintaining population diversity.

## 3.2 ADAPTIVE LARGE NEIGHBORHOOD SEARCH

Adaptive Large Neighborhood Search (ALNS) dynamically adjusts neighborhood size and leverages the Dual-layer Self-evolutionary LLM Agent for variable scoring and selection. At each iteration $t$, the LLM agent computes scores $s_i^{(t)}$ for decision variables $x_i$ based on their potential to improve the objective value. The top-$k$ variables are selected to form the neighborhood $\mathcal{N}^{(t)}$:

$$\mathcal{N}^{(t)} = \{x_i \mid \text{rank}(s_i^{(t)}) \leq k\}, \tag{4}$$

where $\mathcal{N}^{(t)}$ is the neighborhood at iteration $t$, and $k$ is the current neighborhood size. A subproblem is then solved within $\mathcal{N}^{(t)}$, and the solution $\mathbf{x}$ is updated if an improvement is found.

The neighborhood size $k$ is adaptively adjusted based on search progress. If the improvement in the objective value falls below a threshold $\epsilon$ for $p$ consecutive iterations, $k$ is expanded to explore a broader search space $k \leftarrow \min(k_{\max}, k + \lceil u\% \cdot n \rceil)$, where $u\%$ is the adjustment rate and $n$ is the total number of decision variables. Conversely, if the time spent solving subproblems within the neighborhood exceeds a predefined limit, $k$ is reduced to focus on a smaller subset of variables $k \leftarrow \max(k_{\min}, k - \lceil u\% \cdot n \rceil)$.

Table 2: **Online Bin Packing Heuristic Comparison.** This table compares the performance of various bin packing heuristics based on the fraction of excess bins (lower values indicate better performance) across different Weibull distribution instances.

| | 1k_C100 | 5k_C100 | 10k_C100 | 1k_C500 | 5k_C500 | 10k_C500 | Avg |
|---|---|---|---|---|---|---|---|
| First Fit | 5.32% | 4.40% | 4.44% | 4.97% | 4.27% | 4.28% | 4.61% |
| Best Fit | 4.87% | 4.08% | 4.09% | 4.50% | 3.91% | 3.95% | 4.23% |
| FunSearch | 3.78% | **0.80%** | **0.33%** | 6.75% | 1.47% | 0.74% | 2.31% |
| EOH | 4.48% | 0.88% | 0.83% | 4.32% | 1.06% | 0.97% | 2.09% |
| Ours | **3.58%** | 0.85% | 0.41% | **3.67%** | **0.82%** | **0.42%** | **1.63%** |

The key innovation of ALNS lies in the use of the LLM agent to generalize variable selection strategies. The agent is trained on small-scale MILP problems and learns to rank variables based on their impact on the objective function, enabling it to generalize these strategies to larger, more complex problems. This transfer of knowledge ensures that neighborhood selection is both adaptive and intelligent, allowing the method to efficiently navigate the vast search space of large-scale MILPs.

By leveraging the LLM agent's ability to learn and generalize, ALNS dynamically balances exploration and exploitation, focusing computational resources on the most promising regions of the solution space. The pseudocode in Algorithm 1 outlines the overall process, where the adaptive control of $k$ ensures faster convergence to high-quality solutions while maintaining computational efficiency.

## 4 EXPERIMENT

To validate the effectiveness of the proposed LLM-LNS framework, we conduct two sets of experiments. First, we evaluate our proposed Dual-layer Self-evolutionary LLM Agent on heuristic generation tasks for combinatorial optimization problems, comparing it against methods like FunSearch (Romera-Paredes et al., 2024) and EOH (Liu et al., 2024). Second, we assess the full LLM-LNS framework on large-scale MILP problems, where it is compared against traditional LNS methods (e.g., ACP (Ye et al., 2023a)), ML-based LNS approaches (e.g., CL-LNS (Huang et al., 2023b)), the SOTA solvers like Gurobi (Gurobi Optimization, LLC, 2023) and SCIP (Maher et al., 2016), and modern ML optimization frameworks such as GNN&GBDT (Ye et al., 2023c) and Light-MILPopt (Ye et al., 2023b). More experimental results and details are provided in the Appendices A to D.

### 4.1 HEURISTIC GENERATION FOR COMBINATORIAL OPTIMIZATION PROBLEMS

In this section, we evaluate the performance of the Dual-layer Self-evolutionary LLM Agent in generating heuristic strategies for well-known combinatorial optimization problems. We focus on two widely studied problems: Online Bin Packing (Seiden, 2002) and the Traveling Salesman Problem (TSP) (Hoffman et al., 2013). Our method is compared against several hand-crafted heuristics, state-of-the-art machine learning methods, and other automatically designed heuristics.

#### 4.1.1 ONLINE BIN PACKING

The objective of the Online Bin Packing problem is to allocate a collection of items into the fewest possible bins of fixed capacity. We follow the experimental setup from Romera-Paredes et al. (2024), using Weibull distribution instances with varying numbers of items (1k to 10k) and bin capacities (100 and 500). The performance of each method is measured by the fraction of excess bins used, where lower values indicate better performance. We compare our method against several baselines, including hand-crafted heuristics First Fit (Tang et al., 2016) and Best Fit (Shor, 1991), which are widely used in practice, as well as automatically generated heuristics FunSearch (Romera-Paredes et al., 2024) and EOH (Liu et al., 2024), which represent state-of-the-art approaches.

As shown in Table 2, our method consistently achieves the best performance across different problem sizes and capacities, with an average excess bin fraction of 1.63%, outperforming both hand-crafted heuristics and automatically generated methods. In particular, our approach excels on the 10k items, capacity 500 instance, achieving a fraction of excess bins of 0.42%, outperforming FunSearch (0.74%) and EOH (0.97%). This result highlights the strong scalability and generalization ability of our method, making it particularly effective in handling large-scale, high-capacity scenarios.

Table 3: **Traveling Salesman Problems Heuristic Performance Evaluation.** This table provides a comparison of the relative distance to the best-known solutions for different routing heuristics (lower values indicate better performance) on a subset of TSPLib benchmark instances.

| | rd100 | pr124 | bier127 | kroA150 | u159 | kroB200 | Avg |
|---|---|---|---|---|---|---|---|
| NI | 19.91% | 15.50% | 23.21% | 18.17% | 23.59% | 24.10% | 20.75% |
| FI | 9.38% | 4.43% | 8.04% | 8.54% | 11.15% | 7.54% | 8.18% |
| Or-Tools | **0.01%** | 0.55% | 0.66% | 0.02% | 1.75% | 2.57% | 0.93% |
| AM | 3.41% | 3.68% | 5.91% | 3.78% | 7.55% | 7.11% | 5.24% |
| POMO | **0.01%** | 0.60% | 13.72% | 0.70% | 0.95% | 1.58% | 2.93% |
| LEHD | **0.01%** | 1.11% | 4.76% | 1.40% | 1.13% | 0.64% | 1.51% |
| EOH | **0.01%** | **0.00%** | 0.42% | 0.29% | **-0.01%** | **0.26%** | 0.16% |
| Ours | **0.01%** | **0.00%** | **0.01%** | **0.00%** | **-0.01%** | 0.44% | **0.08%** |

### 4.1.2 TRAVELING SALESMAN PROBLEM

The Traveling Salesman Problem (TSP) is a classic combinatorial optimization problem where the goal is to find the shortest route that visits all given locations exactly once. We evaluate our method on a subset of TSPLib benchmark instances (Reinelt, 1991), with performance measured by the relative distance to the best-known solutions (lower values indicate better performance). We compare our method against two types of baselines: hand-crafted heuristics and AI-generated heuristics. The hand-crafted heuristics include Nearest Insertion (NI) and Farthest Insertion (FI) (Rosenkrantz et al., 1977), two widely used constructive heuristics. We also include Google OR-Tools (Perron & Furnon), a popular solver, using its default settings and the recommended local search option. Beyond EOH (Liu et al., 2024), we compare against the Attention Model (AM) (Kool et al., 2018), POMO (Kwon et al., 2020), and LEHD (Luo et al., 2023), all of which are ML-based methods.

As shown in Table 3, our method achieves the best average performance with a 0.08% gap to the best-known solutions, outperforming both hand-crafted heuristics and neural network-based methods. Notably, on the bier127 instance, our method achieves a relative distance of just 0.01% to the best-known solution, significantly outperforming EOH (0.42%) and other baselines, including LEHD (4.76%) and AM (5.91%). This substantial improvement highlights the effectiveness of our approach in solving challenging instances of the TSP.

It is important to note that both the Online Bin Packing and TSP problems use the same GPT-4o-mini LLM, with identical settings: 20 iterations and a population size of 20 for Online Bin Packing, and 10 for the TSP problem. Despite these identical settings, our method consistently outperforms EOH in both problems, showcasing the superior efficiency of the dual-layer self-evolutionary mechanism in exploring the solution space. This mechanism allows our method to dynamically adapt and refine solutions, resulting in better overall performance with the same computational resources. These results underscore the robustness and scalability of our approach, offering a promising direction for solving large-scale combinatorial optimization problems using LLMs.

### 4.2 PERFORMANCE OF LLM-LNS ON LARGE-SCALE MILP PROBLEMS

To validate the effectiveness of the proposed LLM-LNS framework for large-scale MILP problems, we evaluate its performance on four widely-used benchmark datasets: Set Covering (SC) (Caprara et al., 2000), Minimum Vertex Cover (MVC) (Dinur & Safra, 2005), Maximum Independent Set (MIS) (Tarjan & Trojanowski, 1977), and Mixed Integer Knapsack Set (MIKS) (Atamtürk, 2003). Initially, LLM-LNS is trained on smalle-scale problems with tens of thousands of variables and constraints and then tested on large-scale instances (Table 4) to assess its scalability and generalization.

Table 4: The size of one real-world case study in the internet domain and four widely used NP-hard benchmark MILPs.

| Problem | Scale | Number of Variables | Number of Constraints |
|---|---|---|---|
| SC (Minimize) | $SC_1$ | 200000 | 200000 |
| | $SC_2$ | 2000000 | 2000000 |
| MVC (Minimize) | $MVC_1$ | 100000 | 300000 |
| | $MVC_2$ | 1000000 | 3000000 |
| MIS (Maximize) | $MIS_1$ | 100000 | 300000 |
| | $MIS_2$ | 1000000 | 3000000 |
| MIKS (Maximize) | $MIKS_1$ | 200000 | 200000 |
| | $MIKS_2$ | 2000000 | 2000000 |

We compare LLM-LNS with several state-of-the-art baselines, including heuristic LNS methods like Random-LNS (Song et al., 2020), Adaptive Constraint Propagation (ACP) (Ye et al., 2023a), and the learning-based CL-LNS framework

Table 5: **Comparison of objective values on large-scale MILP instances across different methods.** For each instance, the best-performing objective value is highlighted in bold. The - symbol indicates that the method was unable to generate samples for any instance within 30,000 seconds, while * indicates that the GNN&GBDT framework could not solve the MILP problem.

| | $SC_1$ | $SC_2$ | $MVC_1$ | $MVC_2$ | $MIS_1$ | $MIS_2$ | $MIKS_1$ | $MIKS_2$ |
|---|---|---|---|---|---|---|---|---|
| Random-LNS | 16140.6 | 169417.5 | 27031.4 | 276467.5 | 22892.9 | 223748.6 | 36011.0 | 351964.2 |
| ACP | 17672.1 | 182359.4 | 26877.2 | 274013.3 | 23058.0 | 226498.2 | 34190.8 | 332235.6 |
| CL-LNS | - | - | 31285.0 | - | 15000.0 | - | - | - |
| Gurobi | 17934.5 | 320240.4 | 28151.3 | 283555.8 | 21789.0 | 216591.3 | 32960.0 | 329642.4 |
| SCIP | 25191.2 | 385708.4 | 31275.4 | 491042.9 | 18649.9 | 9104.3 | 29974.7 | 168289.9 |
| GNN&GBDT | 16728.8 | 252797.2 | 27107.9 | 271777.2 | 22795.7 | 227006.4 | * | * |
| Light-MILPOPT | 16108.1 | 160015.5 | 26950.7 | 269571.5 | 22966.5 | 230432.9 | 36125.5 | 362265.1 |
| LLM-LNS(Ours) | **15802.7** | **158878.9** | **26725.3** | **268033.7** | **23169.3** | **231636.9** | **36479.8** | **363749.5** |

(Huang et al., 2023b). Additionally, we include traditional solvers like Gurobi (Gurobi Optimization, LLC, 2023) and SCIP (Maher et al., 2016), as well as modern ML-based frameworks such as GNN&GBDT (Ye et al., 2023c) and Light-MILPopt (Ye et al., 2023b). To ensure a fair comparison, Gurobi is used as the sub-solver in the neighborhood search step across all methods. For LLM-LNS, the neighborhood selection strategy is trained over 20 iterations on smaller problems before being applied to larger instances. Detailed results and discussions are provided in the Appendix D.

The experimental results, summarized in Table 5, show that LLM-LNS consistently outperforms traditional LNS-based heuristics and learning-based methods. Unlike hand-crafted LNS strategies, which are typically static and less effective as problem complexity increases, LLM-LNS dynamically adapts through its dual-layer self-evolutionary mechanism, enabling more efficient exploration of the solution space. Even compared to state-of-the-art learning-based LNS methods like CL-LNS, LLM-LNS demonstrates superior performance. Although CL-LNS represents one of the most advanced learning-based approaches, it often fails to complete sampling within an acceptable time for large-scale instances, and even when results are obtained, the solution quality is significantly lower. This highlights the challenges faced by existing LNS-based methods when dealing with large and complex MILP problems, while underscoring the robustness and adaptability of LLM-LNS.

In addition, LLM-LNS shows a clear advantage over traditional solvers like Gurobi and SCIP, as well as learning-based methods such as GNN&GBDT and Light-MILPopt. While traditional solvers perform competitively on smaller instances, their performance degrades significantly as the problem size increases. Similarly, learning-based methods struggle with large-scale MILPs, finding it difficult to efficiently explore the exponentially growing solution space. In contrast, LLM-LNS consistently delivers superior results across both small and large-scale problems, offering a scalable and efficient solution. These findings suggest that LLM-LNS not only bridges the gap between traditional and learning-based methods, but also opens new avenues for scalable optimization in large-scale MILPs.

Overall, the experimental results demonstrate the effectiveness of our proposed innovations. In the first set of experiments, we validate the capability of the **Dual-layer Self-evolutionary LLM Agent** to autonomously generate competitive heuristic strategies for combinatorial optimization problems, consistently outperforming state-of-the-art methods such as FunSearch and EOH. This confirms the agent's ability to balance exploration and exploitation, as guided by the **Differential Memory for Directional Evolution**. In the second set, we apply the **LLM-LNS framework** to large-scale MILP problems, where it not only surpasses traditional LNS methods and advanced solvers like Gurobi and SCIP, but also demonstrates superior scalability compared to modern ML-based frameworks. These results highlight the success of applying our LLM agent to **neighborhood selection in LNS**, showcasing its generalization to complex, large-scale problems with minimal training data.

## 5 CONCLUSION

In this paper, we propose **LLM-LNS**, a Large Language Model-driven LNS framework for solving large-scale MILP problems, utilizing a dual-layer self-evolutionary LLM agent to automate heuristic strategy generation. Experiments show that LLM-LNS consistently outperforms traditional solvers, learning-based methods, and state-of-the-art LNS frameworks. Future work will explore new agent architectures and broader optimization problems, aiming to further enhance the integration of LLMs with optimization techniques. The code of LLM-LNS will be open-sourced after the paper review.

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

## APPENDIX

This Appendix contains four sections, each addressing a specific aspect of the experimental setup and results. Below is a brief overview of each section:

- **Parameter Settings** (Appendix A): This section describes key experimental parameters, including the number of top-performing heuristic strategies evaluated, thresholds for stagnation detection, and criteria for evolutionary convergence. Parameter values for Bin Packing (BP), Maximum Vertex Covering (MVC), and Mixed Integer Knapsack Set (MIKS) are also outlined.

- **Evolutionary Process of LLM-LNS** (Appendix B): This section explains the co-evolution of the inner and outer layers in the Dual-layer Self-Evolutionary LLM Agent. It includes comparisons between the Evolution of Heuristic (EoH) method and the proposed dual-layer approach for problems like Bin Packing and Traveling Salesman Problem (TSP).

- **Convergence Analysis of LLM-LNS** (Appendix C): This section analyzes the convergence behavior of the LLM-LNS method compared to EoH. Faster convergence rates, superior solution quality, and greater stability in problems like Online Bin Packing and Traveling Salesman Problem are demonstrated through graphs and figures.

- **Supplementary Experiments for LLM-LNS on Large-Scale MILP Problems** (Appendix D): This section presents the performance of LLM-LNS on large-scale Mixed Integer Linear Programming (MILP) problems, evaluated with different subsolvers (e.g., SCIP) and compared to traditional and learning-based methods. Error bar comparisons highlight solution consistency and reliability.

- **Ablation Study of the Dual-Layer Self-evolutionary LLM Agent** (Appendix E): This section evaluates the contributions of the dual-layer framework, analyzing the roles of Prompt Evolution (outer layer) and Directional Evolution (inner layer). Results from small- and large-scale datasets highlight their complementary effects on convergence, diversity, and performance.

- **Additional Validation Experiments** (Appendix F): This section presents experiments validating the stability, generalization, and robustness of LLM-LNS, with deeper insights into its scalability and consistency.

- **Population Management Strategy** (Appendix G): This section details the population management strategy in LLM-LNS, including ranking-based selection of parent strategies, criteria for identifying poorly performing strategies, and methods for maintaining diversity and quality. Mathematical formalizations illustrate how this mechanism balances exploration and exploitation to improve strategy quality over generations.

- **Limitations and Future Directions** (Appendix H): This section discusses the limitations of the proposed framework and outlines potential future directions to enhance its scalability and applicability.

These appendices provide a comprehensive overview of the experimental setup, evolutionary process, convergence analysis, and supplementary experiments, offering a deeper understanding of the performance and robustness of the LLM-LNS method in solving complex combinatorial optimization problems.

## A EXPERIMENTAL SETTINGS

In this section, we detail the parameter settings used in our experiments for both the Dual-layer Self-evolutionary LLM Agent and the Adaptive Large Neighborhood Search (ALNS). We also provide an overview of the standard MILP problem instances used in this study.

### A.1 DUAL-LAYER SELF-EVOLUTIONARY LLM AGENT PARAMETERS

The following key parameters were used for the evolutionary process of the LLM agent:

- $h$: Represents the number of top-performing heuristic strategies used to evaluate each prompt strategy. For each prompt strategy, the top-$h$ heuristics it generates are tracked,

and their average fitness score is used as the fitness score for the prompt strategy. In our experiments, $h$ is set to half of the population size. Specifically:

– For **Bin Packing (BP)** and **Traveling Salesman Problem (TSP)**, the population sizes are 20 and 10, respectively, so $h$ is set to 10 and 5.

– For the four MILP problems—**Maximum Vertex Covering (MVC)**, **Set Covering (SC)**, **Independent Set (IS)**, and **Mixed Integer Knapsack Set (MIKS)**—the population size is 4, so $h$ is set to 2.

• $l$: Denotes the number of top individuals in the heuristic population that are monitored for stagnation. If the top-$l$ individuals remain unchanged for $t$ generations, we infer that the evolution has potentially converged to a local optimum, triggering the introduction of new prompt strategies. In all our experiments, $l$ is set to 4.

• $t$: The number of consecutive generations during which the top-$l$ individuals must remain unchanged before stagnation is detected. In all our experiments, $t$ is set to 3.

## A.2 ADAPTIVE LARGE NEIGHBORHOOD SEARCH (ALNS) PARAMETERS

For ALNS, we use the following parameters:

• **Neighborhood size** $k$: Set to half of the decision variable count $n$. This represents the number of decision variables selected to form the search neighborhood in each iteration.

• **Time limit** $T$: The maximum allowed runtime for solving the problem.

• **Threshold** $\epsilon$: Represents the minimum improvement in the objective function to continue exploring the current neighborhood. We set $\epsilon = $ 1e-3.

• **Iteration limit** $p$: The number of consecutive iterations with improvements below the threshold $\epsilon$ before expanding the neighborhood size. We set $p = 3$.

• **Minimum and maximum neighborhood sizes** $k_{\min}, k_{\max}$: These are set to $k_{\min} = 0$ and $k_{\max} = n$ (the total number of decision variables in the problem).

• **Adjustment rate** $u\%$: Specifies the percentage of decision variables $n$ by which the neighborhood size is adjusted during expansion or reduction. In our experiments, we set $u\% = 10$.

## A.3 DATASETS FOR HEURISTIC EVOLUTION

To ensure a fair comparison with state-of-the-art methods such as EOH, we adopted the same dataset configurations as those used in EOH for heuristic evolution. For example, in the online bin packing problem, the evaluation dataset consists of five sets of instances, each containing 5,000 items generated from a Weibull distribution. These instances cover a wide range of item counts and container capacities, ensuring the diversity and representativeness of the problem settings. Similarly, for the traveling salesman problem (TSP), we utilized 64 randomly selected instances from TSP100, which were also used in EOH's experiments. These instances provide a well-established basis for evaluating heuristic performance in combinatorial optimization tasks.

For MILP problems, we followed a similar design approach to that used in the online bin packing problem. Specifically, we employed five small-scale MILP problems, each involving tens of thousands of decision variables and linear constraints. These smaller-scale problems serve as a foundation for heuristic evolution, allowing the method to generalize effectively to larger-scale MILP problems with hundreds of thousands or even millions of decision variables. This demonstrates the scalability and practical applicability of our approach when addressing large-scale optimization challenges.

## A.4 EXPERIMENTAL SETTINGS FOR ALGORITHM DESIGN

Our proposed dual-layer agent framework is designed to evolve heuristics for solving combinatorial optimization problems, specifically targeting Online Bin Packing (BP) and the Travelling Salesman Problem (TSP). The dual-layer architecture is responsible for learning and refining heuristic

strategies for these problems, enabling efficient and scalable solutions. Below, we provide detailed descriptions of the experimental settings for each problem.

For **Online Bin Packing**, we adopt the settings described in (Romera-Paredes et al., 2024) and (Liu et al., 2024) to design heuristics for determining suitable bin allocations for incoming items (Angelopoulos et al., 2023). The task of the dual-layer agent is to design a scoring function that assigns items to bins. The inputs to the agent include the size of the item and the remaining capacities of the bins, while the output is a set of scores for the bins. The item is then assigned to the bin with the highest score. This process is iterated for each incoming item, allowing the agent to dynamically adapt its scoring strategy based on the evolving state of the bins.

For the **Travelling Salesman Problem (TSP)**, we use the dual-layer agent to design heuristics for Guided Local Search (GLS) (Voudouris et al., 2010). GLS introduces perturbations and dynamically adjusts the objective landscape to help escape local optima, enabling broader exploration of the solution space. A critical task in GLS is updating the distance matrix to guide the local search towards more promising regions. In this context, the dual-layer agent is tasked with producing heuristics for updating the distance matrix. The inputs include the current distance matrix, the current route, and the number of edges, while the output is an updated distance matrix. GLS then applies local search operators iteratively on the updated landscape to refine the solution. In our experiments, we utilize two common local search operators: the relocate operator and the 2-opt operator, which are widely recognized for their effectiveness in TSP optimization (Arnold & Sörensen, 2019).

These settings are aligned with those used in EOH to ensure fair comparisons and reproducibility. Detailed descriptions of the inputs, outputs, and operators are provided in the appendix of the manuscript to further clarify our experimental configurations.

We also emphasize that no seed heuristics, expert-written code, or prior knowledge were manually introduced during the experiments. All heuristic strategies were initialized automatically by the large language model (LLM), ensuring fairness in the comparisons.

## A.5 MILP PROBLEM OVERVIEW

We use a set of standard problem instances based on four canonical MILP problems: Maximum Independent Set (MIS), Minimum Vertex Covering (MVC), Set Covering (SC), and Mixed Integer Knapsack Set (MIKS). Below are the formal definitions of these problems.

**Maximum Independent Set problem (MIS)**: The Maximum Independent Set problem has applications in network design, where one might need to select the largest subset of mutually non-interacting entities, such as devices in a wireless network to avoid interference. Another common application is in social network analysis, where independent sets can represent groups of users who do not have direct connections, useful for targeting non-overlapping communities.

Consider an undirected graph $\mathcal{G} = (\mathcal{V}, \mathcal{E})$, where a subset of nodes $\mathcal{S} \subseteq \mathcal{V}$ is called an independent set if no edge $e \in \mathcal{E}$ exists between any pair of nodes in $\mathcal{S}$. The MIS problem seeks to find an independent set of maximum cardinality. The binary decision variable $x_v$ indicates whether node $v \in \mathcal{V}$ is part of the independent set ($x_v = 1$) or not ($x_v = 0$). The problem can be formulated as:

$$
\begin{aligned}
\max \quad & \sum_{v \in \mathcal{V}} x_v \\
\text{s.t.} \quad & x_u + x_v \le 1, \quad \forall (u, v) \in \mathcal{E}, \\
& x_v \in \{0, 1\}, \quad \forall v \in \mathcal{V}.
\end{aligned}
\tag{5}
$$

**Minimum Vertex Covering problem (MVC)**: The Minimum Vertex Covering problem is widely used in resource allocation, where one needs to ensure that every interaction (edge) between pairs of objects (nodes) is covered by a resource. For example, in network security, this problem can be used to efficiently place security agents or sensors such that all communication links are monitored.

Given an undirected graph $\mathcal{G} = (\mathcal{V}, \mathcal{E})$, a subset of nodes $\mathcal{S} \subseteq \mathcal{V}$ is called a covering set if for any edge $e \in \mathcal{E}$, at least one of its endpoints is included in $\mathcal{S}$. The MVC problem aims to find a covering set of minimum cardinality. The binary decision variable $x_v$ indicates whether node $v \in \mathcal{V}$ is part of the covering set ($x_v = 1$) or not ($x_v = 0$). The problem is formulated as:

$$\min \quad \sum_{v \in \mathcal{V}} x_v$$
$$\text{s.t.} \quad x_u + x_v \geq 1, \quad \forall (u,v) \in \mathcal{E}, \tag{6}$$
$$x_v \in \{0,1\}, \quad \forall v \in \mathcal{V}.$$

**Set Covering problem (SC)**: The Set Covering problem is fundamental in facility location, where one must select the minimum number of locations (subsets) to serve all customers (elements of the universal set). It is also used in airline crew scheduling, where the goal is to assign the minimum number of crews to cover all flights.

Given a finite universal set $\mathcal{U} = \{1, 2, \ldots, n\}$ and a collection of $m$ subsets $S_1, \ldots, S_m$ of $\mathcal{U}$, each subset $S_i$ is associated with a cost $c_i$. The SC problem involves selecting a combination of these subsets such that every element in $\mathcal{U}$ is covered by at least one of the selected subsets, while minimizing the total cost. The binary decision variable $x_i$ indicates whether subset $S_i$ is selected ($x_i = 1$) or not ($x_i = 0$). The problem is formulated as:

$$\min \quad \sum_{i=1}^{m} c_i x_i$$
$$\text{s.t.} \quad \sum_{i=1}^{m} x_i \cdot \mathbf{1}_{\{j \in S_i\}} \geq 1, \quad \forall j \in \mathcal{U}, \tag{7}$$
$$x_i \in \{0,1\}, \quad \forall i \in \{1, \ldots, m\}.$$

**Mixed Integer Knapsack Set problem (MIKS)**: The Mixed Integer Knapsack Set problem is commonly used in logistics, resource allocation, and portfolio selection problems. It models situations where some resources can be allocated fractionally while others must be fully included or excluded. For example, in supply chain management, some goods can be shipped partially, while others must be shipped as a whole.

The MIKS problem is a generalization of the knapsack problem that involves both continuous and binary decision variables. Given $N$ sets and $M$ items, each item must be covered by at least one of the sets. The objective is to minimize the total cost of the selected sets, where some sets can be partially selected. Let $x_i$ represent the decision variable for set $i$, where $x_i = 1$ indicates full selection, and $0 \leq x_i \leq 1$ allows partial selection. The problem is formulated as:

$$\min \quad \sum_{i=1}^{N} c_i x_i$$
$$\text{s.t.} \quad \sum_{i:j \in S_i} x_i \geq 1, \quad \forall j \in \{1, 2, \ldots, M\}, \tag{8}$$
$$0 \leq x_i \leq 1, \quad \forall i \in \{1, 2, \ldots, N\},$$
$$x_i \in \{0,1\} \text{ or } [0,1], \quad \forall i \in \{1, 2, \ldots, N\}.$$

# B    EVOLUTIONARY PROCESS OF LLM-LNS

## B.1    EVOLUTIONARY PROCESS OVERVIEW

In this appendix, we provide a detailed breakdown of the experimental results and the evolution of heuristic strategies generated by our proposed **Dual-layer Self-Evolutionary LLM Agent**. The following sections offer a comprehensive analysis of how the inner and outer layers of the LLM agent collaborate to generate and refine heuristic strategies across various combinatorial optimization problems, including **Online Bin Packing (bp_online)**, the **Traveling Salesman Problem (TSP)**, and large-scale MILP instances such as **Maximum Vertex Covering (MVC)**, **Set Covering (SC)**, **Independent Set (IS)**, and **Mixed Integer Knapsack Set (MIKS)**.

- **Inner and Outer Layer Prompt Initialization and Evolution**: As shown in Sec. B.2, our approach leverages a dual-layer architecture, where the **inner layer** evolves heuristic strategies by modifying solution components, while the **outer layer** evolves the prompt structure guiding the inner layer, balancing exploration and exploitation. The inner layer prompts iteratively generate heuristics by scoring decision variables based on their contributions to the objective function and constraints, with randomness included to avoid local optima. This enables the LLM to reason about the problem structure and generate high-quality strategies, even without extensive domain expertise. The **outer layer** maintains diversity by evolving prompt structures to prevent premature convergence on suboptimal solutions. Both layers adapt based on past performance, allowing the LLM to refine its strategy generation over time.

- **Heuristic Improvement Through Dual-layer Self-evolutionary LLM Agent**: As shown in Sec. B.3, we demonstrates the progression of heuristic strategies, starting from initial random strategies and gradually evolving into more effective ones through the dual-layer self-evolutionary process. The initial strategies are simple and focus on ranking decision variables based on their contributions to the objective function and constraints. Over time, the LLM agent introduces additional complexity, such as incorporating randomness and penalizing larger deviations from the current solution, improving the robustness of the generated heuristics. The progression of the population is guided by the outer layer, which adjusts the structure and focus of prompts to encourage exploration and avoid premature convergence. The inner layer then refines specific solution components in response to the prompts, iteratively improving the performance of the heuristic strategies. As seen from the evolution of objective scores, the dual-layer system enables the generation of increasingly effective heuristics, balancing exploration with exploitation to achieve superior results in various problem instances.

- **Heuristic Strategies for Bin Packing Online: EoH vs. Dual-Layer Self-Evolution LLM Agent**: As shown in Sec. B.4, both the *Evolution of Heuristic (EoH)* method and our Dual-layer Self-Evolution LLM Agent utilize LLM-based evolutionary processes to generate heuristic strategies for the *Bin Packing Online* problem. The strategy generated by EoH approach, while leveraging LLM to evolve heuristics, focuses primarily on a hybrid scoring system that combines utilization ratios, dynamic adjustments, and an exponentially decaying factor. This method is effective but tends to rely on a more static set of features and parameters, which limits its adaptability across diverse problem instances. In contrast, our Dual-layer Self-Evolution LLM Agent incorporates a more dynamic and adaptive strategy. By combining nonlinear capacity scaling, relative size assessment, and historical penalties for overutilized bins, our approach allows for greater flexibility and adaptability. Specifically, the generated heuristics dynamically adjust based on remaining capacity, item size, and previous bin usage, thereby balancing local search with global optimization. This adaptability enables our agent to discover and refine more efficient strategies that minimize the number of bins used. The results clearly demonstrate that while both methods use LLM-based evolution, our dual-layer approach consistently outperforms the EoH method in terms of solution quality and computational efficiency. The dual-layer system's ability to evolve both the heuristic strategies and the prompt structures ensures that it can fine-tune solutions more effectively, leading to superior bin utilization and fewer bins required overall. This highlights the strength of our approach in generating more robust and context-aware heuristics.

- **Heuristic Strategies for Traveling Salesman Problem (TSP): EoH vs. Dual-Layer Self-Evolution LLM Agent**: Similar to the *Bin Packing Online* problem, both the *Evolution of Heuristic (EoH)* method and our Dual-layer Self-Evolution LLM Agent use LLM-based evolutionary processes to generate heuristic strategies for the *Traveling Salesman Problem (TSP)*. As shown in Sec. B.5, the strategy generated by EoH method employs a randomized approach that adjusts the edge distance matrix by increasing the distances of a random proportion of edges, while rewarding a smaller subset of unused edges. This method encourages exploration but tends to apply uniform adjustments without fully accounting for the global structure of the solution. In contrast, strategy generated by our Dual-layer Self-Evolution LLM Agent introduces a more sophisticated edge distance adjustment mechanism. It dynamically explores alternative routes by incorporating an inverse frequency factor, which penalizes frequently used edges and rewards less frequently used ones. This

adaptive mechanism gradually resets excessively amplified distances, promoting diversification and improving the exploration of the solution space. Furthermore, it balances exploitation by focusing on refining the most promising routes based on past tours, leading to faster convergence towards a global optimum. The results clearly demonstrate that while both methods are effective in exploring new routes, the dual-layer approach consistently outperforms the EoH method in terms of solution quality and convergence speed. By incorporating a more nuanced edge adjustment process and dynamically adapting to the problem context, the Dual-layer Self-Evolution LLM Agent achieves superior results in minimizing the total distance, making it a more robust and efficient solution for the TSP.

- **Evolutionary Path of the Dual-Layer Self-Evolution LLM Agent**: As illustrated in Sec. B.6, we trace the evolutionary process of the LLM agent in solving Maximum Vertex Cover (MVC) problem, detailing how heuristic strategies evolve step by step through the inner and outer layers, gradually converging to optimized solutions. Initially, the agent generates simple heuristics that focus on ranking decision variables based on their impact on the objective function and constraint violation, incorporating randomness to encourage exploration. These early strategies serve as a foundation for further refinement. As the process evolves, the outer layer refines the prompt instructions, guiding the inner layer to develop more sophisticated heuristics. The LLM begins to incorporate additional factors, such as the absolute difference from the initial solution and a more nuanced treatment of constraints. This results in improved exploration of the solution space, as well as better handling of both the objective function and constraints. In the later stages, the agent integrates more advanced techniques, such as hybrid methods combining genetic algorithms with local search, to enhance convergence speed and solution quality. The final heuristics represent a co-evolutionary approach that balances exploration and exploitation, leading to significantly optimized solutions. The evolution of prompts, from the initial simplistic forms to highly specialized instructions, demonstrates the power of the dual-layer architecture in improving both the heuristic strategies and the problem-solving process itself.

- **Evolutionary Result of the Dual-Layer Self-Evolution LLM Agent**: Finally, we present the results achieved by the LLM agent after the completion of the entire evolutionary process across three challenging combinatorial optimization problems: Set Covering (SC), Maximum Independent Set (MIS), and Mixed Integer Knapsack Set (MIKS). As detailed in Sec. B.7, the final heuristics generated by the Dual-layer Self-Evolution LLM Agent are compared with those produced by traditional methods and state-of-the-art approaches, demonstrating significant improvements in solution quality and computational efficiency. For the Set Covering problem (SC), the LLM agent's final heuristic achieves a superior balance between minimizing the number of selected sets and satisfying the constraints. By dynamically adjusting penalties and incorporating random exploration, the agent efficiently navigates the solution space, outperforming traditional methods in both the objective score and constraint satisfaction. In the Maximum Independent Set (MIS) problem, the LLM agent leverages simulated annealing principles combined with adaptive scoring of decision variables. This approach not only ensures thorough exploration but also accelerates convergence towards high-quality solutions. The agent's ability to balance objective contributions with constraint violations leads to a considerable reduction in the total error, as reflected in the final objective score. Lastly, for the Mixed Integer Knapsack Set (MIKS) problem, the LLM agent adopts a hybrid strategy that integrates genetic algorithms and simulated annealing. This allows for a more diversified search process, strategically selecting decision variables based on their contributions to the objective function and constraint interactions. The agent's solution demonstrates a significant improvement over existing methods, particularly in how it dynamically adapts to varying problem constraints while maintaining computational efficiency.

In summary, the proposed Dual-layer Self-Evolutionary LLM Agent effectively generates and refines heuristic strategies for diverse combinatorial optimization problems. Leveraging the complementary roles of its inner and outer layers, it balances exploration and exploitation to discover high-quality, context-aware strategies. Its adaptability in evolving both problem-solving heuristics and guiding prompts ensures superior solution quality and computational efficiency. From online bin packing to large-scale MILP problems, the agent consistently outperforms traditional and state-of-the-art methods, demonstrating robustness, scalability, and evolutionary refinement.

## B.2 INNER AND OUTER LAYER PROMPT INITIALIZATION AND EVOLUTION

---

**Prompt for Generating Initial Heuristic Strategies**

Given an initial feasible solution and a current solution to a Mixed-Integer Linear Programming (MILP) problem, with variables' lower_bound, upper_bound and coefficient in objective function. We want to improve the current solution using Large Neighborhood Search (LNS).

The task can be solved step-by-step by starting from the current solution and iteratively selecting a subset of decision variables to relax and re-optimize. In each step, most decision variables are fixed to their values in the current solution, and only a small subset is allowed to change. You need to score all the decision variables based on the information I give you, and I will choose the decision variables with high scores as neighborhood selection. To avoid getting stuck in local optima, the choice of the subset can incorporate a degree of randomness.

First, describe your new algorithm and main steps in one sentence. The description must be inside a brace. Next, implement it in Python as a function named select_neighborhood. This function should accept 5 input(s): 'initial_solution', 'current_solution', 'lower_bound', 'upper_bound', 'objective_coefficient'. The function should return 1 output(s): 'neighbor_score'. 'initial_solution', 'current_solution', 'lower_bound', 'upper_bound' and 'objective_coefficient' are numpy arrays. 'neighbor_score' is also a numpy array that you need to create manually. The i-th element of the arrays corresponds to the i-th decision variable. All are Numpy arrays. I don't give you 'neighbor_score' so that you need to create it manually. The length of the 'neighbor_score' array is the same as the length of the other arrays.

Do not give additional explanations.

---

**(Cross) Initial Prompt for Heuristic Strategies Evolution**

Given an initial feasible solution and a current solution to a Mixed-Integer Linear Programming (MILP) problem, with variables' lower_bound, upper_bound and coefficient in objective function. We want to improve the current solution using Large Neighborhood Search (LNS).

The task can be solved step-by-step by starting from the current solution and iteratively selecting a subset of decision variables to relax and re-optimize. In each step, most decision variables are fixed to their values in the current solution, and only a small subset is allowed to change. You need to score all the decision variables based on the information I give you, and I will choose the decision variables with high scores as neighborhood selection. To avoid getting stuck in local optima, the choice of the subset can incorporate a degree of randomness.

I have 5 existing algorithm's thought, objective function value with their codes as follows: No.1 algorithm's thought, objective function value, and the corresponding code are: ...
No.2 algorithm's thought, objective function value, and the corresponding code are: ...
...
No.5 algorithm's thought, objective function value, and the corresponding code are: ...

Please help me create a new algorithm that has a totally different form from the given ones.

First, describe your new algorithm and main steps in one sentence. The description must be inside a brace. Next, implement it in Python as a function named select_neighborhood. This function should accept 5 input(s): 'initial_solution', 'current_solution', 'lower_bound', 'upper_bound', 'objective_coefficient'. The function should return 1 output(s): 'neighbor_score'. 'initial_solution', 'current_solution', 'lower_bound', 'upper_bound' and 'objective_coefficient' are numpy arrays. 'neighbor_score' is also a numpy array that you need to create manually. The i-th element of the arrays corresponds to the i-th decision variable. All are Numpy arrays. I don't give you 'neighbor_score' so that you need to create it manually. The length of the 'neighbor_score' array is the same as the length of the other arrays.

Do not give additional explanations.

---

**(Cross) Initial Prompt Strategies**

1. Please help me create a new algorithm that has a totally different form from the given ones.
2. Please help me create a new algorithm that has a totally different form from the given ones but can be motivated from them.

---

**(Cross) Prompt for Prompt Strategies Evolution**

We are working on solving a minimization problem. Our objective is to leverage the capabilities of the Language Model (LLM) to generate heuristic algorithms that can efficiently tackle this problem. We have already developed a set of initial prompts and observed the corresponding outputs. However, to improve the effectiveness of these algorithms, we need your assistance in carefully analyzing the existing prompts and their results. Based on this analysis, we ask you to generate new prompts that will help us achieve better outcomes in solving the minimization problem.

I have 5 existing prompts with objective function value as follows:
No.1 prompt's tasks assigned to LLM, and objective function value are: ...
No.2 prompt's tasks assigned to LLM, and objective function value are: ...
...
No.5 prompt's tasks assigned to LLM, and objective function value are: ...

Please help me create a new prompt that has a totally different form from the given ones but can be motivated from them.

Please describe your new prompt and main steps in one sentence. Do not give additional explanations.

## B.3 HEURISTIC IMPROVEMENT THROUGH DUAL-LAYER SELF-EVOLUTIONARY LLM AGENT

**Heuristic 1 (Obj Score: 5375.52145)**
Rank decision variables based on their penalty contribution and the difference from current solution, incorporating randomness in scoring.

```python
import numpy as np
def select_neighborhood(n, m, k, site, value,
        constraint, initial_solution,
        current_solution, objective_coefficient):
    neighbor_score = np.zeros(n)
    variable_difference = np.zeros(n)
    for i in range(m):
        lhs = sum(value[i][j] * current_solution[
                site[i][j]] for j in range(k[i]))
        penalty = max(0, lhs - constraint[i])
        for j in range(k[i]):
            var_index = site[i][j]
            difference = current_solution[
                    var_index] - initial_solution[
                    var_index]
            neighbor_score[var_index] += penalty *
                    difference
    neighbor_score += objective_coefficient * np.
        random.rand(n)
    return neighbor_score
```

**Heuristic 2 (Obj Score: 5383.05876)**
Rank decision variables based on their objective contribution and impact on current solution deviation, with randomness included in the scoring process.

```python
import numpy as np
def select_neighborhood(n, m, k, site, value,
        constraint, initial_solution,
        current_solution, objective_coefficient):
    neighbor_score = np.zeros(n)
    variable_contribution = np.zeros(n)
    for i in range(m):
        lhs = sum(value[i][j] * current_solution[
                site[i][j]] for j in range(k[i]))
        deviation = lhs - constraint[i]
        for j in range(k[i]):
            var_index = site[i][j]
            contribution = value[i][j] * (
                    initial_solution[var_index] -
                    current_solution[var_index])
            neighbor_score[var_index] +=
                    contribution
    neighbor_score += objective_coefficient + np.
        random.rand(n)
    return neighbor_score
```

**Heuristic 3 (Obj Score: 5384.8486)**
This modified algorithm ranks decision variables based on their contribution to the total current solution's objective function value and their degree of constraint satisfaction.

```python
import numpy as np
def select_neighborhood(n, m, k, site, value,
        constraint, initial_solution,
        current_solution, objective_coefficient):
    neighbor_score = np.zeros(n)
    for i in range(m):
        lhs = sum(value[i][j] * current_solution[
                site[i][j]] for j in range(k[i]))
        for j in range(k[i]):
            if lhs > constraint[i]:
                neighbor_score[site[i][j]] +=
                        objective_coefficient[site[i
                        ][j]] * (lhs - constraint[i])
            else:
                neighbor_score[site[i][j]] +=
                        objective_coefficient[site[i
                        ][j]] * (constraint[i] - lhs)
    neighbor_score += np.random.rand(n) * 0.1
    return neighbor_score
```

**Heuristic 4 (Obj Score: 5384.95417)**
Rank decision variables by their contribution to the objective function and difference from initial values, while also weighing their frequency of use in the constraints.

```python
import numpy as np
def select_neighborhood(n, m, k, site, value,
        constraint, initial_solution,
        current_solution, objective_coefficient):
    score = np.zeros(n)
    frequency = np.zeros(n)
    for i in range(m):
        lhs = sum(value[i][j] * current_solution[
                site[i][j]] for j in range(k[i]))
        deviation = lhs - constraint[i]
        for j in range(k[i]):
            var_index = site[i][j]
            contribution = value[i][j] * np.abs(
                    initial_solution[var_index] -
                    current_solution[var_index])
            score[var_index] += contribution
            frequency[var_index] += 1
    neighbor_score = score / (frequency + 1e-5) +
        objective_coefficient + np.random.rand(n)
    return neighbor_score
```

**Prompt Designed by LLM**
Develop an algorithm that combines the strengths of existing heuristics while introducing random perturbations to enhance exploration and minimize the objective function more effectively.

**Heuristic (Obj Score: 5374.19865)**
Rank decision variables based on their contribution to the objective function and incorporate the absolute difference from the initial solution while adding a degree of randomness to the scores.

```python
import numpy as np

def select_neighborhood(n, m, k, site, value, constraint, initial_solution, current_solution,
        objective_coefficient):
    neighbor_score = np.zeros(n)
    for i in range(m):
        lhs = sum(value[i][j] * current_solution[site[i][j]] for j in range(k[i]))
        for j in range(k[i]):
            var_index = site[i][j]
            difference = np.abs(current_solution[var_index] - initial_solution[var_index])
            neighbor_score[var_index] += (constraint[i] - lhs) * difference
    neighbor_score += objective_coefficient * np.random.rand(n)
    return neighbor_score
```

## B.4 HEURISTIC STRATEGIES FOR BIN PACKING ONLINE: EoH VS. DUAL-LAYER SELF-EVOLUTION LLM AGENT

---

**Heuristic Designed by EoH**

**Description**
The heuristic incorporates a weighted average of the utilization ratio, dynamic adjustment, and an exponentially decaying factor, with different parameter settings to minimize the number of used bins.

**Code**

```python
import numpy as np
def heuristic(item, bins):
    diff = bins-item # remaining capacity
    exp = np.exp(diff) # exponent term
    sqrt = np.sqrt(diff) # square root term
    ulti = 1-diff/bins # utilization term
    comb = ulti * sqrt # combination of utilization and square root
    adjust = np.where(diff > (item * 3), comb + 0.8, comb + 0.3)
        # hybrid adjustment term to penalize large bins
    hybrid_exp = bins / ((exp + 0.7) *exp)
        # hybrid score based on exponent term
    scores = hybrid_exp + adjust
        # sum of hybrid score and adjustment
    return scores
```

---

**Heuristic Designed by Dual-layer Self-evolution LLM Agent**

**Description**
The new algorithm employs a hybrid optimization strategy that combines nonlinear penalties for historical usage, adaptive capacity scaling, and a relative size assessment, facilitating a balance between local and global search for optimal bin assignment.

**Code**

```python
import numpy as np
def score(item, bins):
    feasible_bins = bins[bins > item]
    scores = np.zeros_like(bins)
    if len(feasible_bins) == 0:
        return scores
    # Nonlinear capacity scaling that enhances the desire for larger
        spaces
    remaining_capacity = feasible_bins - item
    capacity_scaling = np.log1p(remaining_capacity) * (
        remaining_capacity / np.max(remaining_capacity))
    # Relative size assessment: quadratic term comparing item size with
         bin capacities
    relative_size_effect = (item ** 2 / feasible_bins) * 50  # Scale to
         moderate impact
    # Nonlinear penalty based on historical usage counts to deter
        overutilization
    historical_count = np.arange(len(feasible_bins)) + 1  # Simulating
        historical usage
    penalty_factor = np.power(1.5, historical_count)  # Exponential
        penalty for higher usage
    # Combining scores: enhanced capacity scaling, moderated size
        assessment, and historical penalties
    scores[bins > item] = capacity_scaling - relative_size_effect -
        penalty_factor
    return scores
```

## B.5 HEURISTIC STRATEGIES FOR TRAVELING SALESMAN PROBLEM: EOH VS. DUAL-LAYER SELF-EVOLUTION LLM AGENT

---

**Heuristic Designed by EoH**

**Description**

This algorithm uses a randomized approach to update the edge distance matrix by randomly selecting a proportion of edges to increase their distances while uniformly rewarding a smaller proportion of unused edges to encourage exploration.

**Code**

```python
import numpy as np
def update_edge_distance(edge_distance, local_opt_tour, edge_n_used):
    N = edge_distance.shape[0]
    updated_edge_distance = edge_distance.copy()
    # Parameters for randomization
    increase_factor = 2.0
    decrease_factor = 0.9
    random_selection_ratio = 0.3  # percentage of edges to randomly adjust
    # Identify all edges used in the local optimal tour
    used_edges = set()
    for i in range(len(local_opt_tour)):
        start = local_opt_tour[i]
        end = local_opt_tour[(i + 1) % len(local_opt_tour)]
        used_edges.add((min(start, end), max(start, end)))
    # Randomly select a proportion of edges to increase distance
    all_edges = [(i, j) for i in range(N) for j in range(N) if i != j]
    np.random.shuffle(all_edges)
    num_edges_to_increase = int(len(all_edges) * random_selection_ratio)
    for edge in all_edges[:num_edges_to_increase]:
        start, end = edge
        # If the edge is used in the local optimal tour, apply a higher increase
        if (min(start, end), max(start, end)) in used_edges:
            updated_edge_distance[start, end] *= increase_factor
            updated_edge_distance[end, start] *= increase_factor
        else:
            updated_edge_distance[start, end] *= decrease_factor
            updated_edge_distance[end, start] *= decrease_factor
    return updated_edge_distance
```

---

**Heuristic Designed by Dual-layer Self-evolution LLM Agent**

**Description**

The new algorithm refines the edge distance adjustment mechanism by incorporating an acceptance heuristic that dynamically explores alternative routes while gradually resetting excessively amplified distances, thus promoting diversification and improved convergence towards a global optimum.

**Code**

```python
import numpy as np
def update_edge_distance(edge_distance, local_opt_tour, edge_n_used):
    # Create a copy of the edge distance matrix for updates
    updated_edge_distance = np.copy(edge_distance)
    # Extract the number of nodes
    num_nodes = edge_distance.shape[0]
    # Calculate the inverse frequency factor for each edge
    inverse_frequency_factor = np.max(edge_n_used) - edge_n_used + 1
    # Update the edge distance based on the local optimal tour
    for i in range(len(local_opt_tour)):
        # Get the current and next node in the local optimal tour
        current_node = local_opt_tour[i]
        next_node = local_opt_tour[(i + 1) % len(local_opt_tour)]
        # Apply the inverse frequency factor to decrease the edge weight
        updated_edge_distance[current_node, next_node] *= inverse_frequency_factor[
            current_node, next_node]
        updated_edge_distance[next_node, current_node] *= inverse_frequency_factor[
            next_node, current_node]
    return updated_edge_distance
```

## B.6 EVOLUTIONARY PATH OF THE DUAL-LAYER SELF-EVOLUTION LLM AGENT

**Heuristic (Obj Score: 5400.48176)**
The algorithm ranks decision variables based on their impact on the objective function and how they relate to the violated constraints, incorporating a degree of randomness.

**Code**

```python
import numpy as np
def select_neighborhood(n, m, k, site, value, constraint,
        initial_solution, current_solution, objective_coefficient):
    neighbor_score = np.zeros(n)
    violated_constraints = 0
    for i in range(m):
        lhs = sum(value[i][j] * current_solution[site[i][j]] for
                j in range(k[i]))
        if lhs > constraint[i]:
            violated_constraints += 1
            for j in range(k[i]):
                neighbor_score[site[i][j]] +=
                        objective_coefficient[site[i][j]]
    if violated_constraints > 0:
        neighbor_score /= violated_constraints
    randomness = np.random.rand(n) * 0.1
    neighbor_score += randomness
    return neighbor_score
```

**Heuristic (Obj Score: 5374.19865)**
Rank decision variables based on their contribution to the objective function and incorporate the absolute difference from the initial solution while adding a degree of randomness to the scores.

**Code**

```python
import numpy as np
def select_neighborhood(n, m, k, site, value, constraint,
        initial_solution, current_solution, objective_coefficient):
    neighbor_score = np.zeros(n)
    for i in range(m):
        lhs = sum(value[i][j] * current_solution[site[i][j]] for
                j in range(k[i]))
        for j in range(k[i]):
            var_index = site[i][j]
            difference = np.abs(current_solution[var_index] -
                    initial_solution[var_index])
            neighbor_score[var_index] += (constraint[i] - lhs) *
                    difference
    neighbor_score += objective_coefficient * np.random.rand(n)
    return neighbor_score
```

**Heuristic (Obj Score: 5373.34904)**
Develop a co-evolutionary heuristic approach that integrates genetic algorithms with local search techniques to enhance convergence speed and minimize the objective function for the specified problem.

**Code**

```python
import numpy as np
def select_neighborhood(n, m, k, site, value, constraint,
        initial_solution, current_solution, objective_coefficient):
    neighbor_score = np.zeros(n)
    for i in range(m):
        lhs = sum(value[i][j] * current_solution[site[i][j]] for
                j in range(k[i]))
        for j in range(k[i]):
            var_index = site[i][j]
            difference = np.abs(current_solution[var_index] -
                    initial_solution[var_index])
            neighbor_score[var_index] += (constraint[i] - lhs) *
                    difference
    random_adjustment = np.random.rand(n)
    adaptive_mutation_rate = np.clip(np.abs(objective_coefficient
            ), 0.1, 1.0)
    neighbor_score += adaptive_mutation_rate * random_adjustment
    return neighbor_score
```

**Initial Prompts**

- (Cross) Please help me create a new algorithm that has a totally different form from the given ones.
- (Cross) Please help me create a new algorithm that has a totally different form from the given ones but can be motivated from them.
- (Variation) Please assist me in creating a new algorithm that has a different form but can be a modified version of the algorithm provided.
- (Variation) Please identify the main algorithm parameters and assist me in creating a new algorithm that has a different parameter settings of the score function provided.

**Current Prompts**

- (Cross) Develop a modified heuristic algorithm that utilizes a hybrid approach, combining elements of simulated annealing and genetic algorithms, to optimize the given minimization problem.
- (Cross) Design a modified heuristic algorithm for the minimization problem by incorporating elements of simulated annealing with a unique cooling schedule.
- (Variation) Please identify the main algorithm parameters and assist me in creating a new algorithm that has a different parameter settings of the score function provided.
- (Variation) Develop an algorithm that combines the strengths of existing heuristics while introducing random perturbations to enhance exploration and minimize the objective function more effectively.

**Final Prompts**

- (Cross) Develop a hybrid heuristic algorithm for the minimization problem that combines genetic algorithms with tabu search to enhance local search capabilities while maintaining diversity in the solution population.
- (Cross) Develop a co-evolutionary heuristic approach that integrates genetic algorithms with local search techniques to enhance convergence speed and minimize the objective function for the specified problem.
- (Variation) Design a novel optimization strategy that integrates genetic algorithms with dynamic programming principles to enhance the search for optimal solutions, focusing on adaptive mutation rates to effectively minimize the objective function value.
- (Variation) Design a novel optimization framework that integrates particle swarm optimization with genetic algorithms, focusing on adaptive mutation strategies to enhance convergence speed and minimize the objective function value.

## B.7 EVOLUTIONARY RESULT OF THE DUAL-LAYER SELF-EVOLUTION LLM AGENT

### B.7.1 EVOLUTIONARY RESULT OF SET COVERING PROBLEM

**Heuristic (Obj Score: 3339.39339)**
This algorithm computes scores based on the penalty incurred by each variable when deviating from the current solution and evaluates the impact on constraint satisfaction.

**Code**

```python
import numpy as np
def select_neighborhood(n, m, k, site, value, constraint, initial_solution, current_solution,
        objective_coefficient):
    neighbor_score = np.zeros(n)
    for i in range(m):
        lhs_value = sum(value[i][j] * current_solution[site[i][j]] for j in range(k[i]))
        for j in range(k[i]):
            variable_index = site[i][j]
            if lhs_value >= constraint[i]:
                penalty = lhs_value - constraint[i]
                contribution = penalty * value[i][j]
                neighbor_score[variable_index] += contribution
            else:
                contribution = value[i][j]
                neighbor_score[variable_index] -= contribution
    costs = np.abs(current_solution - initial_solution) * (objective_coefficient + 1e-5)
    with np.errstate(divide='ignore', invalid='ignore'):
        neighbor_score = np.divide(neighbor_score, costs, where=costs != 0)
    neighbor_score -= np.min(neighbor_score)
    neighbor_score /= np.max(neighbor_score) if np.max(neighbor_score) != 0 else 1
    rand_factor = np.random.rand(n) * 0.1
    neighbor_score += rand_factor
    return neighbor_score
```

**Final Prompts**

- (Cross) Please help me create a new algorithm that has a totally different form from the given ones.
- (Cross) Please help me create a new algorithm that has a totally different form from the given ones but can be motivated from them.
- (Variation) Please assist me in creating a new algorithm that has a different form but can be a modified version of the algorithm provided.
- (Variation) Please identify the main algorithm parameters and assist me in creating a new algorithm that has a different parameter settings of the score function provided.

### B.7.2 EVOLUTIONARY RESULT OF MAXIMUM INDEPENDENT SET PROBLEM

**Heuristic (Obj Score: -4634.0636)**
This new heuristic approach combines the principles of simulated annealing with the adaptive scoring of decision variables based on their contributions to violated constraints while incorporating randomness to enhance exploration of the solution space.

**Code**

```python
import numpy as np
def select_neighborhood(n, m, k, site, value, constraint, initial_solution, current_solution,
        objective_coefficient):
    neighbor_score = np.zeros(n)
    current_objective_value = np.dot(current_solution, objective_coefficient)
    variable_contributions = np.zeros(n)
    for i in range(m):
        lhs_value = sum(value[i][j] * current_solution[site[i][j]] for j in range(k[i]))
        if lhs_value > constraint[i]:
            for j in range(k[i]):
                var_index = site[i][j]
                variable_contributions[var_index] += (value[i][j] * (current_solution[var_index] == 1))
    for index in range(n):
        improvement = objective_coefficient[index] - variable_contributions[index]
        neighbor_score[index] = improvement + (current_solution[index] * 0.5)
    temperature = np.random.uniform(0.1, 1.0)
    randomness = np.random.uniform(-temperature, temperature, size=n)
    neighbor_score += randomness
    return neighbor_score
```

**Final Prompts**

- (Cross) Develop a novel hybrid algorithm that combines local search and simulated annealing techniques to explore the solution space and minimize the objective function more effectively.
- (Cross) Design a novel optimization algorithm inspired by the existing methods, focusing on adaptive parameter tuning to enhance convergence toward better solutions.
- (Variation) Design a novel heuristic approach inspired by the principles of simulated annealing to optimize the following problem parameters.
- (Variation) Please identify the main algorithm parameters and assist me in creating a new algorithm that has a different parameter settings of the score function provided.

### B.7.3 Evolutionary Result of Mixed Integer Knapsack Set Problem

**Heuristic (Obj Score: -3612.99096)**
This novel algorithm enhances diversity in the solution search process by strategically selecting decision variables based on both their objective contributions and constraint interactions, while incorporating a degree of random exploration.

**Code**

```python
import numpy as np
def select_neighborhood(n, m, k, site, value, constraint, initial_solution, current_solution,
      objective_coefficient):
    neighbor_score = np.zeros(n)
    contribution_scores = objective_coefficient * current_solution
    neighbor_score += contribution_scores
    for i in range(m):
        lhs_value = sum(value[i][j] * current_solution[site[i][j]] for j in range(k[i]))
        if lhs_value > constraint[i]:
            for j in range(k[i]):
                var_index = site[i][j]
                penalty = (lhs_value - constraint[i]) / max(1, np.sum(value[i]))
                neighbor_score[var_index] -= penalty * value[i][j] * np.random.uniform(0.8, 1.2)
    local_search_factor = (initial_solution - current_solution) ** 2
    neighbor_score += local_search_factor
    randomness = np.random.rand(n) * 0.1
    neighbor_score += randomness
    if np.max(neighbor_score) > 0:
        neighbor_score /= np.max(neighbor_score)
    return neighbor_score
```

**Final Prompts**

- (Cross) Design a hybrid heuristic algorithm that combines elements of genetic algorithms and simulated annealing to explore the solution space efficiently.

- (Cross) Develop a multi-phase heuristic optimization strategy that integrates particle swarm optimization with tabu search to dynamically adapt search parameters and enhance convergence rates.

- (Variation) Develop an algorithm that incorporates a novel optimization strategy, diverging from previous approaches, to enhance the objective function's outcome by exploring alternative parameter tuning techniques.

- (Variation) Please identify the main algorithm parameters and assist me in creating a new algorithm that has a different parameter settings of the score function provided.

## C  Convergence Analysis of LLM-LNS

### C.1  Evolutionary Progress in Combinatorial Optimisation Problem

Across both two combinatorial optimization problems Online Bin Packing and Traveling Salesman Problem, LLM-LNS consistently shows superior convergence and final solution quality compared to EOH.

In the Online Bin Packing problem shown in Figure 3, LLM-LNS shows better convergence behavior from the early stages. As the generations progress, LLM-LNS steadily improves and consistently outperforms EOH. The reduced variance in later generations highlights the stability of the LLM-LNS approach, which efficiently balances exploration and exploitation. Its dual-layer structure allows it to thoroughly explore the solution space, avoiding premature convergence and reaching a higher overall objective score. In contrast, EOH exhibits larger fluctuations and fails to achieve the same level of performance, indicating its limitations in maintaining robust progress during the evolutionary process.

In the Traveling Salesman Problem shown in Figure 4, although LLM-LNS starts with a less favorable initial population compared to EOH, it quickly demonstrates its advantage. Initially, EOH performs better, but it stagnates after the first 8 generations, showing little improvement afterward. Meanwhile, LLM-LNS continues to refine its solutions and steadily decreases the objective score. This indicates that the dual-layer structure of LLM-LNS effectively prevents it from getting trapped in local optima, maintaining a high level of exploration even in later generations. By the end of the evolutionary process, LLM-LNS surpasses EOH, achieving better overall results.

In both problems, LLM-LNS's ability to maintain diversity early in the process, combined with its strong convergence in later stages, gives it a clear advantage over EOH. The dual-layer evolutionary strategy ensures that LLM-LNS avoids stagnation, allowing for continuous improvement and ultimately leading to superior performance in solving combinatorial optimization problems.

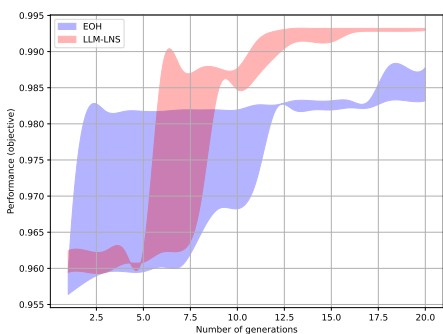
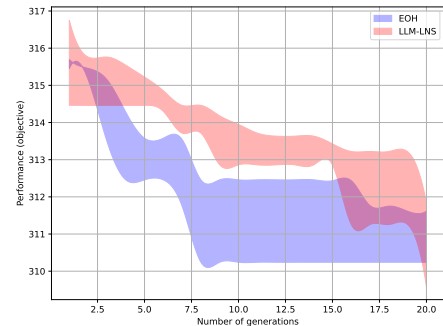

Figure 3: Evolutionary Progress of Heuristic Strategies in Online Bin Packing

Figure 4: Evolutionary Progress of Heuristic Strategies in Traveling Salesman Problem

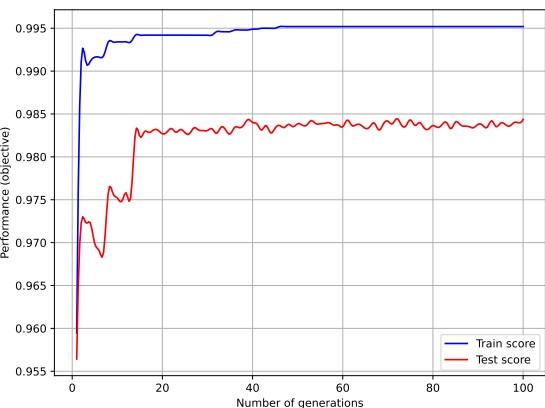

Figure 5: Convergence of Training and Testing Scores in 100-Generation of Online Bin Packing Problem.

## C.2    CONVERGENCE ANALYSIS OF GENERATIONS

In the Online Bin Packing Problem, we conducted 100 generations of iterative training using the proposed dual-layer strategy. Figure 5 shows the convergence trends for both the training and testing scores over these 100 generations. The results provide interesting insights into the behavior of our model during the evolutionary process, particularly in terms of how the training and testing losses evolve differently.

The training loss demonstrates a clear and consistent downward trend throughout the generations. Initially, the training score starts relatively high, but quickly drops within the first few generations. This rapid initial improvement indicates that the evolutionary algorithm is highly effective at optimizing the objective function within the training set. As the generations progress, the training score continues to decrease, eventually converging to a very low value. This steady decline suggests that the model is successfully adapting to the problem, continually refining its population and reducing the training objective. The absence of significant fluctuations in later generations implies that the model has reached a stable state, effectively minimizing the training loss with little variance.

On the other hand, the testing loss follows a somewhat different pattern. Initially, we observe a sharp decline in the testing score, which mirrors the behavior of the training score. However, after this initial drop, the testing score does not continue to improve as steadily as the training score. Instead, it stabilizes around a certain value and begins to exhibit small fluctuations. This behavior suggests that while the model is able to generalize to a degree, it encounters more variability in

Table 6: Comparison of objective values on large-scale MILP instances across different methods using SCIP as optimizer. For each instance, the best-performing objective value is highlighted in bold. The - symbol indicates that the method was unable to generate samples for any instance within 30,000 seconds, while * indicates that the GNN&GBDT framework could not solve the MILP problem.

| | $SC_1$ | $SC_2$ | $MVC_1$ | $MVC_2$ | $MIS_1$ | $MIS_2$ | $MIKS_1$ | $MIKS_2$ |
|---|---|---|---|---|---|---|---|---|
| Random-LNS | 16164.2 | 171655.6 | 27049.6 | 277255.3 | 22892.9 | 222076.8 | 691.7 | 6870.1 |
| ACP | 17743.4 | 192791.2 | 27432.9 | 281862.4 | 23058.0 | 216008.8 | 29879.2 | 7913.5 |
| CL-LNS | - | - | 31285.0 | - | 15000.0 | - | - | - |
| Gurobi | 17934.5 | 320240.4 | 28151.3 | 283555.8 | 21789.0 | 216591.3 | 32960.0 | 329642.4 |
| SCIP | 25191.2 | 385708.4 | 31275.4 | 491042.9 | 18649.9 | 9104.3 | 29974.7 | 168289.9 |
| GNN&GBDT | 16728.8 | 261174.0 | 27107.9 | 271777.2 | 22795.7 | 227006.4 | * | * |
| Light-MILPOPT | 16147.2 | 166756.0 | 26956.8 | 269771.3 | 22963.6 | 230278.1 | 36125.5 | **357483.8** |
| LLM-LNS(Ours) | **15950.2** | **161732.8** | **26763.4** | **268825.5** | **23137.19** | **230682.8** | **36147.7** | 350468.7 |

the testing data compared to the training data. These fluctuations could be attributed to the inherent complexity or diversity of the unseen test instances, which the model has not been directly optimized for.

This phenomenon is reminiscent of the behavior observed during neural network training, where the training loss continues to decrease as the model becomes more specialized in fitting the training data, while the testing loss reaches a plateau and may exhibit some fluctuations. In this case, the testing loss reflects the model's ability to generalize beyond the training set. The fact that the testing score does not continue to decrease beyond a certain point suggests that the model may have reached its limit in terms of generalization, possibly due to overfitting to the training data. However, the steady fluctuations in the testing score indicate that the model remains adaptable and does not suffer from severe overfitting, as there is no significant increase in the testing loss.

Overall, the divergence between the training and testing scores in later generations highlights the trade-off between optimization and generalization. While the dual-layer evolutionary strategy is highly effective at optimizing the training set, it must also balance the need for generalization to unseen data. The oscillation of the testing score around a stable value suggests that the model is reasonably robust but may benefit from additional techniques to further enhance its generalization performance, such as regularization or early stopping strategies in future iterations.

In summary, the convergence analysis of the 100-generation experiment reveals that while the training loss continues to decrease, the testing loss stabilizes with slight fluctuations. This behavior is indicative of a model that has successfully optimized for the training data while maintaining a reasonable level of generalization, akin to patterns observed in neural network training processes.

# D    SUPPLEMENTARY EXPERIMENTS FOR LLM-LNS ON LARGE-SCALE MILP PROBLEMS

## D.1    PERFORMANCE OF LLM-LNS USING SCIP AS THE SUBSOLVER

In this supplementary set of experiments, we further evaluate the performance of LLM-LNS by incorporating SCIP as the subsolver for large-scale MILP problems. The results, summarized in Table 6, provide a comprehensive comparison across various methods using SCIP, offering deeper insights into the robustness and adaptability of LLM-LNS when faced with different solver strategies.

As seen in the results, LLM-LNS continues to demonstrate superior performance across most instances, consistently outperforming traditional LNS-based methods, learning-based frameworks such as GNN&GBDT, and even advanced solvers like Gurobi and SCIP. The highlighted bold values indicate that LLM-LNS achieves the best objective values in the majority of cases, reinforcing its scalability and effectiveness in large-scale MILP problems.

However, an interesting observation arises in the MIKS instances, where Light-MILPopt outperforms LLM-LNS. This can be attributed to the unique challenges posed by MIKS in large-scale settings. Specifically, MIKS requires significantly more resources for neighborhood searches as the problem size increases, compared to smaller-scale instances. SCIP, as an optimizer, employs a different strategy for solving MIKS, which likely influences the performance of LLM-LNS when

Table 7: Comparison of standard deviation values on large-scale MILP instances across different methods using Gurobi as optimizer.

| | $SC_1$ | $SC_2$ | $MVC_1$ | $MVC_2$ | $MIS_1$ | $MIS_2$ | $MIKS_1$ | $MIKS_2$ |
|---|---|---|---|---|---|---|---|---|
| Random-LNS | 37.5 | 258.1 | 88.4 | 243.0 | 72.1 | 243.0 | 98.2 | 584.0 |
| ACP | 38.4 | 1039.3 | 71.6 | 403.5 | 60.3 | 928.8 | 118.2 | 649.2 |
| CL-LNS | - | - | 617.7 | - | 277.5 | - | - | - |
| Gurobi | 28.8 | 143.4 | 77.2 | 287.3 | 48.8 | 147.5 | 69.0 | 225.7 |
| SCIP | 13823.6 | 298211.7 | 107.3 | 262.0 | 57.5 | 85.8 | 73.2 | 242313.7 |
| GNN&GBDT | 360.1 | 3800.4 | 93.8 | 950.4 | 119.3 | 4738.8 | * | * |
| Light-MILPOPT | 1.0 | 145.7 | 79.4 | 209.4 | 52.1 | 133.1 | 41.7 | 272.5 |
| LLM-LNS(Ours) | 17.7 | 144.2 | 79.7 | 198.1 | 55.2 | 147.6 | 70.2 | 170.4 |

Table 8: Comparison of standard deviation values on large-scale MILP instances across different methods using SCIP as optimizer.

| | $SC_1$ | $SC_2$ | $MVC_1$ | $MVC_2$ | $MIS_1$ | $MIS_2$ | $MIKS_1$ | $MIKS_2$ |
|---|---|---|---|---|---|---|---|---|
| Random-LNS | 18.8 | 250.3 | 79.0 | 234.8 | 72.1 | 401.7 | 18.1 | 36.2 |
| ACP | 30.8 | 6338.3 | 77.2 | 217.6 | 60.3 | 946.4 | 1829.7 | 943.8 |
| CL-LNS | - | - | 617.7 | - | 277.5 | - | - | - |
| Gurobi | 28.8 | 143.4 | 77.2 | 287.3 | 48.8 | 147.5 | 69.0 | 225.7 |
| SCIP | 13823.6 | 298211.7 | 107.3 | 262.0 | 57.5 | 85.8 | 73.2 | 242313.7 |
| GNN&GBDT | 51.4 | 5587.6 | 91.4 | 474.0 | 80.0 | 660.4 | * | * |
| Light-MILPOPT | 37.7 | 693.4 | 77.3 | 216.9 | 51.6 | 151.7 | 80.0 | 1045.8 |
| LLM-LNS(Ours) | 20.4 | 169.5 | 82.6 | 188.7 | 54.3 | 75.9 | 68.7 | 1197.5 |

scaling to larger instances. In smaller-scale problems, LLM-LNS may have learned more aggressive strategies that are effective in those scenarios, but these strategies may lead to timeout issues in larger instances due to the increased computational complexity and extended iteration times required for SCIP. As a result, the overall improvement in performance is limited in these larger MIKS problems.

Despite these challenges, LLM-LNS still exhibits competitive performance in MIKS, managing to outperform many other methods, including Gurobi and traditional LNS strategies. The occasional time-out or reduced efficiency in MIKS does not overshadow the fact that LLM-LNS remains a robust and scalable solution across a wide range of large-scale MILP problems.

In conclusion, these supplementary experiments highlight the adaptability and robustness of LLM-LNS when using different subsolvers, including SCIP. Although challenges remain in specific problem instances like MIKS, LLM-LNS consistently delivers superior performance across most problem types, demonstrating its ability to generalize across solvers and problem scales. The results reinforce the notion that LLM-LNS effectively bridges the gap between traditional solvers and learning-based methods, offering a scalable solution for large-scale combinatorial optimization problems.

## D.2 COMPARISON OF STANDARD DEVIATION VALUES

The comparison of standard deviation (SD) values across different methods using both Gurobi and SCIP as sub-optimizers reveals several key insights into the stability of various approaches when solving large-scale MILP problems. Standard deviation reflects the consistency of the solutions; lower values indicate that the method is more stable and produces less variation in different runs.

As shown in Table 7, for the experiments using Gurobi, LLM-LNS consistently demonstrates low standard deviation values across most instances, indicating that it not only achieves superior objective values but does so with high stability. For example, in $SC_1$, $MVC_2$, and $MIKS_2$, LLM-LNS has SD values of 17.7, 198.1, and 170.4, respectively, which are comparable to or lower than other methods. Light-MILPopt also shows excellent stability in $SC_1$ and $MIKS_1$, with SD values of 1.0 and 41.7, respectively, although its performance fluctuates more in other instances. In contrast, Random-LNS and ACP exhibit higher variability, especially in $SC_2$ and $MIKS_2$, where ACP's SD reaches as high as 1039.3 and 649.2, respectively, suggesting a lack of robustness in these instances. Gurobi itself also shows moderate consistency, while methods like CL-LNS fail to generate results for certain instances, indicating poor scalability for large problems.

As shown in Table 8, when SCIP is used as the optimizer, the trends remain somewhat similar. LLM-LNS continues to show stable performance, particularly in $SC_1$ and $MVC_2$, with SD values of 20.4 and 188.7, respectively. However, SCIP itself exhibits extremely high variability in some instances,

Table 9: Comparison of error bar on large-scale MILP instances across different methods using Gurobi as optimizer.

| | $SC_1$ | $SC_2$ | $MVC_1$ | $MVC_2$ | $MIS_1$ | $MIS_2$ | $MIKS_1$ | $MIKS_2$ |
|---|---|---|---|---|---|---|---|---|
| Random-LNS | 65.4 | 318.3 | 142.1 | 350.8 | 104.4 | 333.6 | 158.9 | 808.8 |
| ACP | 56.8 | 1787.2 | 120.6 | 574.8 | 83.6 | 1233.0 | 173.7 | 742.7 |
| CL-LNS | - | - | 892.6 | - | 406.3 | - | - | - |
| Gurobi | 39.7 | 252.7 | 119.6 | 349.0 | 64.7 | 183.1 | 103.8 | 319.7 |
| SCIP | 25238.2 | 533457.2 | 165.2 | 402.1 | 96.9 | 103.6 | 94.6 | 433463.8 |
| GNN&GBDT | 511.3 | 5504.8 | 148.7 | 1522.6 | 160.1 | 7887.9 | * | * |
| Light-MILPOPT | 1.4 | 206.4 | 121.6 | 289.8 | 78.8 | 216.6 | 63.3 | 420.1 |
| LLM-LNS(Ours) | 27.9 | 187.9 | 125.4 | 289.8 | 82.2 | 199.3 | 111.7 | 259.2 |

Table 10: Comparison of error bar on large-scale MILP instances across different methods using SCIP as optimizer.

| | $SC_1$ | $SC_2$ | $MVC_1$ | $MVC_2$ | $MIS_1$ | $MIS_2$ | $MIKS_1$ | $MIKS_2$ |
|---|---|---|---|---|---|---|---|---|
| Random-LNS | 33.2 | 362.1 | 123.3 | 368.2 | 104.4 | 531.3 | 26.1 | 51.5 |
| ACP | 46.1 | 10845.3 | 106.0 | 324.1 | 83.6 | 1371.4 | 3253.2 | 1055.6 |
| CL-LNS | - | - | 892.6 | - | 406.3 | - | - | - |
| Gurobi | 39.7 | 252.7 | 119.6 | 349.0 | 64.7 | 183.1 | 103.8 | 319.7 |
| SCIP | 25238.2 | 533457.2 | 165.2 | 402.1 | 96.9 | 103.6 | 94.6 | 433463.8 |
| GNN&GBDT | 72.6 | 7349.2 | 147.2 | 678.8 | 100.4 | 1076.6 | * | * |
| Light-MILPOPT | 66.6 | 1223.3 | 118.5 | 305.6 | 79.1 | 239.4 | 124.2 | 1473.9 |
| LLM-LNS(Ours) | 31.7 | 231.2 | 131.9 | 266.7 | 68.9 | 94.7 | 105.9 | 1868.3 |

particularly in $SC_2$ and $MIKS_2$, with SD values exceeding 298,000 and 242,000, respectively, which suggests that SCIP struggles with certain large-scale MILPs. This instability in SCIP could be due to its aggressive strategies or solver configurations being less suited to these specific problem instances. Light-MILPopt again demonstrates relatively stable performance in most instances, although its SD increases significantly in some cases, such as $MIKS_2$. GNN&GBDT and ACP also show considerable fluctuations, with ACP having an SD of 6338.3 in $SC_2$, further highlighting its instability in large-scale settings.

In summary, LLM-LNS not only consistently outperforms other methods in terms of objective values but also maintains strong stability across a wide range of instances, particularly when compared to methods like Random-LNS, ACP, and SCIP. This robustness makes LLM-LNS a strong candidate for solving large-scale MILP problems effectively and consistently.

### D.3 COMPARISON OF ERROR BAR

The error bar comparison across different methods using Gurobi and SCIP as optimizers provides insights into the variability and confidence in solutions across large-scale MILP instances. Error bars quantify the uncertainty or inconsistency in the results, with smaller values indicating more reliable and consistent performance.

As shown in Table 9, for methods using Gurobi, LLM-LNS again demonstrates strong reliability with relatively small error bars across most instances. For example, in $SC_1$, $MVC_2$, and $MIKS_2$, LLM-LNS has error bars of 27.9, 289.8, and 259.2, respectively. These values are noticeably smaller than those for methods like Random-LNS and ACP, which exhibit much larger error bars, reflecting greater instability. Light-MILPopt also shows excellent performance with particularly low error bars in $SC_1$ (1.4) and $MIKS_1$ (63.3), but its error increases significantly in some other instances. Notably, SCIP exhibits extremely large error bars in several instances, such as $SC_2$ and $MIKS_2$, where the error bars exceed 533,000 and 433,000, respectively, indicating significant inconsistency in its performance on these large-scale problems. GNN&GBDT also shows high error bars, suggesting that its performance is less reliable across different runs.

As shown in Table 10, when using SCIP as the optimizer, LLM-LNS continues to demonstrate relatively low error bars, particularly in $SC_1$, $MVC_2$, and $MIKS_1$, where the values are 31.7, 266.7, and 105.9, respectively. These results are significantly more stable compared to methods like ACP and GNN&GBDT, which show very high error bars in instances like $SC_2$ (error bar of 10845.3 for ACP) and $MIKS_2$. SCIP itself again shows extremely high error bars for instances such as $SC_2$ and $MIKS_2$, further highlighting its instability in handling large-scale problems. Light-MILPopt

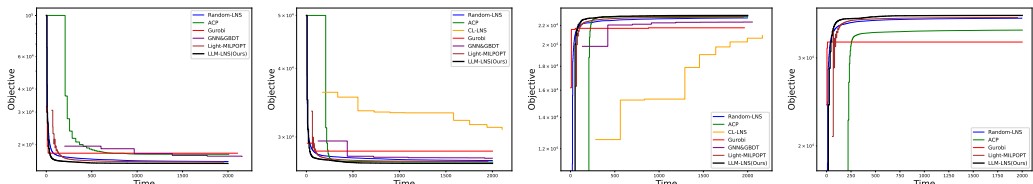

Figure 6: Time-objective value graphs of medium-scale problems using Gurobi: $SC_1$, $MVC_1$, $IS_1$, and $MIKS_1$.

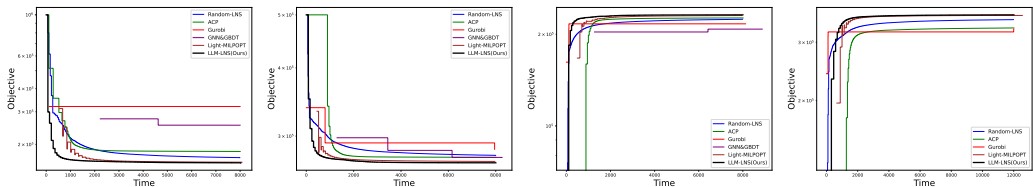

Figure 7: Time-objective value graphs of large-scale problems using Gurobi: $SC_2$, $MVC_2$, $IS_2$, and $MIKS_2$.

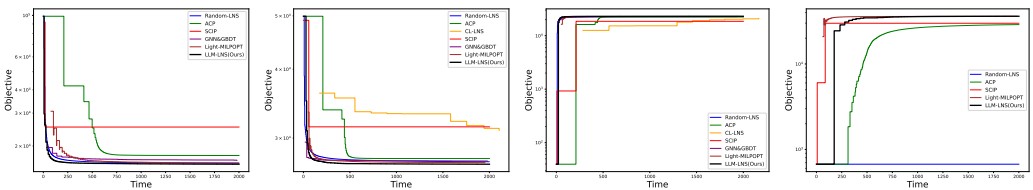

Figure 8: Time-objective value graphs of medium-scale problems using SCIP: $SC_1$, $MVC_1$, $IS_1$, and $MIKS_1$.

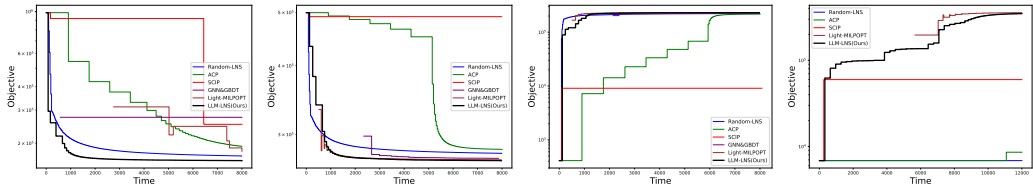

Figure 9: Time-objective value graphs of large-scale problems using SCIP: $SC_2$, $MVC_2$, $IS_2$, and $MIKS_2$.

performs well in some instances but also shows considerable variation in others, with error bars as high as 1473.9 in $MIKS_2$.

Overall, LLM-LNS consistently demonstrates lower error bars across both optimizers, Gurobi and SCIP, indicating that it provides more reliable and consistent solutions for large-scale MILP problems. This makes it a strong candidate for scenarios where both solution quality and stability are critical.

### D.4 CONVERGENCE ANALYSIS

In this section, we analyze the convergence performance of our proposed approach, our proposed LLM-LNS, in comparison to several baseline methods for solving large-scale MILP problems, including Random-LNS, ACP, Gurobi, GNN&GBDT, and Light-MILPOPT. The experimental results are shown in Figures 6 through 9, which include instances of four different problem types: Set Covering (SC), Maximum Vertex Covering (MVC), Independent Set (IS), and Mixed Integer Knap-

sack Set (MIKS). We evaluate both medium-scale and large-scale instances using two solvers as sub-optimizer, Gurobi and SCIP.

The analysis of the convergence curves reveals several important observations:

- **Faster Initial Convergence**: For nearly all problem instances, the **LLM-LNS** approach demonstrates a significantly faster initial convergence compared to the baseline methods. The objective value drops sharply within the first few time steps, indicating that our method can quickly identify high-quality solutions. In contrast, methods like **Random-LNS** and **ACP** exhibit slower initial convergence, requiring more time to achieve similar reductions in the objective value.

- **Superior Final Objective Value**: Across both medium- and large-scale problem instances, our proposed LLM-LNS consistently achieves lower final objective values compared to the other methods. This is particularly evident in the large-scale instances (e.g., $SC_2$, $MVC_2$, $IS_2$, and $MIKS_2$), where the superiority of our method becomes more pronounced. While methods such as Random-LNS and ACP plateau early, often with suboptimal solutions, our proposed LLM-LNS continues to improve the solution even after other methods have stagnated.

- **Stable Convergence Behavior**: The convergence curves of our proposed LLM-LNS exhibit smooth and gradual decreases in the objective value, indicating stable optimization behavior. In contrast, some of the baseline methods, especially Random-LNS and GNN&GBDT, show more erratic convergence patterns, characterized by large and sudden jumps in the objective value. This suggests that our method is more robust and avoids the instability that can arise in heuristic-based search strategies.

- **Scalability**: The performance gap between our proposed LLM-LNS and the baseline methods becomes even more pronounced in large-scale problem instances. For example, in the large-scale $MIKS_2$ and $SC_2$ instances, our proposed LLM-LNS outperforms all other methods by a significant margin, converging to a much lower objective value within a shorter time frame. This demonstrates the scalability of our method, as it remains effective even as the problem size increases, whereas the performance of other methods, such as Light-MILPOPT and ACP, degrades considerably.

- **Comparison with Exact Solvers**: When compared to the exact solver Gurobi, our proposed LLM-LNS shows comparable or even superior performance, particularly in terms of convergence speed. While Gurobi tends to find solutions that improve gradually over time, our proposed LLM-LNS reaches competitive solutions much faster, which is crucial in time-constrained scenarios. This highlights the practical advantage of our method in scenarios where computational resources or time are limited.

In summary, the experimental results demonstrate that **LLM-LNS** has clear advantages in terms of convergence speed, final solution quality, and robustness compared to both heuristic-based and exact optimization methods. Our approach is particularly well-suited for large-scale MILP problems, where it consistently outperforms the baseline methods by a significant margin.

## D.5 BASELINE COMPARISONS WITH ADDITIONAL LNS METHODS

To evaluate the effectiveness of the proposed LLM-LNS framework, we conducted comprehensive comparisons with several LNS methods that utilize different heuristic scoring functions. Specifically, we incorporated **Least-Integral** (Berthold, 2006), **Most-Integral** (Nair et al., 2020), and **RINS** (Danna et al., 2005), which are classical scoring functions commonly used in LNS frameworks, alongside the state-of-the-art methods **ACP** (Ye et al., 2023a) and classic method **Random-LNS** (Song et al., 2020).

The results, summarized in Table 11, demonstrate that our proposed LLM-LNS consistently outperforms all baseline methods across a variety of MILP tasks, including Set Covering (SC), Maximum Vertex Cover (MVC), Maximum Independent Set (MIS), and Mixed Integer Knapsack Set (MIKS). This advantage highlights the superior ability of LLM-LNS to balance exploration diversity and solution convergence.

Table 11: Performance comparison of LLM-LNS with additional LNS methods on MILP tasks. Results are reported as objective values (lower is better).

| Method | $SC_1$ | $SC_2$ | $MVC_1$ | $MVC_2$ | $MIS_1$ | $MIS_2$ | $MIKS_1$ | $MIKS_2$ |
|---|---|---|---|---|---|---|---|---|
| Random-LNS | 16140.6 | 169417.5 | 27031.4 | 276467.5 | 22892.9 | 223748.6 | 36011.0 | 351964.2 |
| ACP | 17672.1 | 182359.4 | 26877.2 | 274013.3 | 23058.0 | 226498.2 | 34190.8 | 332235.6 |
| Least-Integral | 22825.3 | 228188.0 | 29818.0 | 306567.1 | 20106.9 | 195782.2 | 27196.9 | 241663.4 |
| Most-Integral | 50818.2 | 519685.5 | 35340.5 | 327742.4 | 14584.4 | 157686.5 | 31235.3 | 314621.6 |
| RINS | 26116.2 | 261176.3 | 26851.3 | 306215.6 | 23069.7 | 201178.1 | 30049.1 | 299953.4 |
| LLM-LNS (Ours) | **15802.7** | **158878.9** | **26725.3** | **268033.7** | **23169.3** | **231636.9** | **36479.8** | **363749.5** |

From the results in Table 11, several key observations can be made. Among the classical LNS methods, RINS generally achieves better results compared to Least-Integral and Most-Integral, as it leverages neighborhood-based improvements combined with partial solutions. However, these methods still fall significantly behind ACP, Random-LNS, and our proposed LLM-LNS, particularly on larger problem instances such as $SC_2$, $MVC_2$, and $MIKS_2$. For example, on the $SC_2$ problem, RINS achieves an objective value of 261176.3, compared to 158878.9 for LLM-LNS, highlighting the limitations of traditional scoring functions in handling large-scale MILP problems.

ACP and Random-LNS perform much better than the classical scoring-based methods due to their adaptiveness and ability to leverage heuristic diversity. However, even these state-of-the-art baselines are consistently outperformed by LLM-LNS across all tasks. For instance:

- On the $SC_1$ problem, LLM-LNS achieves an objective value of 15802.7, compared to 16140.6 for Random-LNS and 17672.1 for ACP.

- On the $MIS_2$ problem, LLM-LNS achieves 231636.9, compared to 223748.6 for Random-LNS and 226498.2 for ACP.

The clear performance advantage of LLM-LNS can be attributed to its dual-layer architecture, which combines prompt evolution and heuristic strategy optimization to balance search diversity and convergence. The outer layer generates diverse prompts that broaden the search space, avoiding premature convergence to suboptimal solutions. Meanwhile, the inner layer refines heuristic strategies and accelerates convergence by leveraging the evolved prompts. This interaction ensures that LLM-LNS adapts effectively to different problem scales and complexities.

Moreover, the results highlight the scalability of LLM-LNS. While ACP and Random-LNS demonstrate reasonable performance on smaller tasks, their effectiveness diminishes as the problem size increases. In contrast, LLM-LNS maintains its performance advantage across both small-scale (e.g., $SC_1$) and large-scale (e.g., $SC_2$) tasks, showcasing its robustness and adaptability. This scalability is directly enabled by the dynamic feedback loop between the two layers, ensuring continuous refinement of both the search space (via prompt evolution) and the solution strategies (via heuristic evolution).

In summary, the experimental results validate the effectiveness of LLM-LNS over both classical and state-of-the-art LNS baselines. Its dual-layer mechanism provides superior generalization and adaptability, making it a powerful framework for solving diverse and large-scale MILP problems.

# E   ABLATION STUDY OF THE DUAL-LAYER SELF-EVOLUTIONARY LLM AGENT

This section presents the results of the ablation study conducted to analyze the contributions of the dual-layer framework components: **Prompt Evolution** (outer layer) and **Directional Evolution** (inner layer). The study evaluates the effects of removing or isolating each component on the overall performance of the framework. Specifically, we compare the following variations:

- **Base (EOH):** The baseline Evolution of Heuristic (EOH) method without any modifications.

- **Base + Dual Layer:** The EOH method with the dual-layer structure (**Prompt Evolution** in the outer layer).

Table 12: Ablation study results on various datasets. The table compares the baseline (EOH), the addition of the dual-layer structure (Prompt Evolution, outer layer), the addition of the differential evolution mechanism (Directional Evolution, inner layer), and the complete method (Ours). The best results for each dataset are highlighted in bold.

| | 1k_C100 | 5k_C100 | 10k_C100 | 1k_C500 | 5k_C500 | 10k_C500 |
|---|---|---|---|---|---|---|
| Base (EOH) | 4.48% | 0.88% | 0.83% | 4.32% | 1.06% | 0.97% |
| Base + Dual Layer | 3.78% | 0.93% | **0.40%** | 3.91% | 0.92% | **0.39%** |
| Base + Differential | **2.64%** | 0.94% | 0.69% | **2.54%** | 0.94% | 0.70% |
| Ours | 3.58% | **0.85%** | 0.41% | 3.67% | **0.82%** | 0.42% |

- **Base + Differential:** The EOH method with the **Directional Evolution** mechanism (inner layer).

- **Ours:** The complete dual-layer framework incorporating both **Prompt Evolution** and **Directional Evolution**.

We evaluate these variations on datasets of different scales to observe their impact on both small-scale and large-scale problems. The datasets include **1k_C100**, **5k_C100**, **10k_C100**, **1k_C500**, **5k_C500**, and **10k_C500**, representing combinatorial optimization instances of varying sizes. The results are summarized in Table 12.

**Key Observations:**

- **Impact of Prompt Evolution:** Adding the dual-layer structure (Base + Dual Layer) significantly improves performance on large-scale problems, as the outer layer enhances the diversity of the search process through prompt optimization. This is particularly evident in the **10k_C500** dataset, where the error rate decreases from **0.97%** (Base) to **0.39%**.

- **Impact of Directional Evolution:** Incorporating the differential evolution mechanism (Base + Differential) improves performance on small-scale problems by accelerating convergence through more effective crossover and mutation strategies. For example, on the **1k_C100** dataset, the error rate decreases from **4.48%** (Base) to **2.64%**.

- **Synergy of Both Components:** The complete dual-layer framework (Ours) achieves the most balanced improvements across datasets, particularly for larger-scale problems. However, on small-scale datasets like **1k_C100**, the additional exploration introduced by Prompt Evolution can slightly increase the error rate compared to Base + Differential (from **2.64%** to **3.58%**).

These results validate the complementary roles of **Prompt Evolution** and **Directional Evolution** in enhancing both diversity and convergence, demonstrating the effectiveness of the dual-layer framework for solving combinatorial optimization problems of varying scales.

# F ADDITIONAL VALIDATION EXPERIMENTS

## F.1 STABILITY EVALUATION OF MULTIPLE RUNS

To evaluate the stability and consistency of our proposed method, we conducted repeated experiments on the **Bin Packing** task. Specifically, we ran three independent trials for both EoH and our method, and the results are summarized in Table 13.

In our proposed method, the seed heuristic strategies are not hand-crafted. Instead, they are automatically generated by the large language model , which introduces some degree of randomness between runs. Despite this randomness, our method consistently outperforms EoH in both effectiveness and stability. For example, on the **1k_C100**, **10k_C100**, and **10k_C500** test sets, the variance in our results is small, and the average performance is consistently better than EoH.

These results demonstrate that our dual-layer framework, combined with the differential evolution mechanism, effectively enhances both consistency and generalization. Moreover, the smaller variance in our method's results highlights its robustness against the randomness introduced by the seed generation process.

Table 13: Stability evaluation of multiple runs on Bin Packing tasks. Results are reported as error rates (%).

| Method | 1k_C100 | 5k_C100 | 10k_C100 | 1k_C500 | 5k_C500 | 10k_C500 | Avg |
|---|---|---|---|---|---|---|---|
| EOH Run 1 | 4.48% | 0.88% | 0.83% | 4.32% | 1.06% | 0.97% | 2.09% |
| EOH Run 2 | 7.56% | 3.33% | 2.62% | 7.22% | 3.19% | 2.50% | 4.07% |
| EOH Run 3 | 4.18% | 3.24% | 3.35% | 3.79% | 3.12% | 3.21% | 3.48% |
| EOH Avg | **5.41%** | **2.48%** | **2.27%** | **5.11%** | **2.46%** | **2.23%** | **3.33%** |
| Ours Run 1 | 3.58% | 0.85% | 0.41% | 3.67% | 0.82% | 0.42% | 1.63% |
| Ours Run 2 | 2.69% | 0.86% | 0.54% | 2.54% | 0.87% | 0.52% | 1.34% |
| Ours Run 3 | 2.64% | 0.94% | 0.69% | 2.54% | 0.94% | 0.70% | 1.41% |
| Ours Avg | **2.97%↑** | **0.88%↑** | **0.55%↑** | **2.92%↑** | **0.88%↑** | **0.55%↑** | **1.46%↑** |

Table 14: Impact of population size on Bin Packing tasks. Results are reported as error rates (%).

| Method | 1k_C100 | 5k_C100 | 10k_C100 | 1k_C500 | 5k_C500 | 10k_C500 | Avg |
|---|---|---|---|---|---|---|---|
| EOH (20) | 4.48% | 0.88% | 0.83% | 4.32% | 1.06% | 0.97% | 2.09% |
| Ours (4) | 3.23% | 0.80% | 0.43% | 3.96% | 1.27% | 0.89% | 1.76%↑ |
| Ours (20) | 3.58% | 0.85% | 0.41% | 3.67% | 0.82% | 0.42% | **1.63%↑** |

## F.2 IMPACT OF POPULATION SIZE ON EXPERIMENTAL OUTCOMES

To further analyze the impact of population size on experimental outcomes, we conducted additional experiments on the **Bin Packing** task, testing our method with a reduced population size of 4. The results are summarized in Table 14.

As shown in the table, although the average performance slightly decreases with a smaller population size, our method still outperforms EoH (population size 20). For example, the average error of our method with a population size of 4 is **1.76%**, which is better than EoH's **2.09%**. This demonstrates that our dual-layer framework and differential evolution mechanism exhibit significant robustness and effectiveness, maintaining superior performance even with smaller populations.

## F.3 PERFORMANCE COMPARISON WITH EOH ON LNS TASKS

In this subsection, we present a comparison between the proposed LLM-LNS framework and existing method EoH, on large-scale combinatorial optimization tasks. While EoH focus on discovering strategies for combinatorial optimization problems, our LLM-LNS framework is specifically designed to address the challenges of large-scale MILP problems through its dual-layer self-evolutionary mechanism.

We evaluated the methods on the Set Covering (SC) problem, a minimization task, using two large-scale datasets:

- **SC$_1$**: Instances with 200,000 decision variables and constraints.
- **SC$_2$**: Instances with 2,000,000 decision variables and constraints.

EOH-LNS was selected as the primary baseline for comparison because it generally outperforms FunSearch on combinatorial optimization tasks, as reported in prior literature. This ensures a fair and representative evaluation of our framework. The experimental results are summarized in Tables 15 and 16, where LLM-LNS consistently outperforms EOH-LNS across all test instances. Specifically:

- On the **SC$_1$** dataset (200,000 variables and constraints), LLM-LNS achieves an average improvement of **1.67%** over EOH-LNS.
- On the **SC$_2$** dataset (2,000,000 variables and constraints), the improvement is more pronounced, reaching **9.20%** on average.

These results demonstrate the superior capability of LLM-LNS in solving large-scale optimization tasks, particularly in terms of solution quality. The improvements can be attributed to the dual-layer architecture, which effectively balances search diversity and solution convergence.

The results demonstrate that LLM-LNS consistently outperforms EOH-LNS, particularly on large-scale instances. This improvement is enabled by the dual-layer self-evolutionary mechanism, which dynamically balances exploration and exploitation:

Table 15: Performance comparison on the $SC_1$ dataset (200,000 variables and constraints). Results are reported as objective values (lower is better).

| Method | Instance$_1$ | Instance$_2$ | Instance$_3$ | Instance$_4$ | Avg |
|---|---|---|---|---|---|
| EOH-LNS | 16114.27 | 16073.72 | 16046.83 | 16074.26 | 16070.15 |
| LLM-LNS (Ours) | **15830.61↑** | **15801.19↑** | **15800.17↑** | **15800.17↑** | **15802.68↑** |

Table 16: Performance comparison on the $SC_2$ dataset (2,000,000 variables and constraints). Results are reported as objective values (lower is better).

| Method | Instance$_1$ | Instance$_2$ | Instance$_3$ | Instance$_4$ | Avg |
|---|---|---|---|---|---|
| EOH-LNS | 175358.59 | 174339.78 | 174782.76 | 174026.33 | 174978.20 |
| LLM-LNS (Ours) | **158901.57↑** | **158953.57↑** | **158712.64↑** | **158759.90↑** | **158831.42↑** |

- The **outer layer** generates diverse prompts to broaden search space coverage, preventing premature convergence to suboptimal solutions.

- The **inner layer** refines heuristic strategies and accelerates convergence by leveraging the evolved prompts, ensuring high-quality solutions.

This collaborative interaction between the two layers forms a dynamic feedback loop, enabling continuous learning and adaptation. While EOH-LNS demonstrates strong performance on small-scale combinatorial optimization tasks, its inability to balance exploration and convergence limits its scalability to larger and more complex problems, as evidenced by the significant performance gap on $SC_2$.

### F.4 COMPREHENSIVE EVALUATION ON TSPLIB INSTANCES

We evaluated our method on all 87 instances from the TSPLib benchmark to comprehensively assess its performance. As shown in Table 17, our method achieves better results than the EOH baseline on 43 instances, matches EOH on 39 instances, and performs slightly worse on only 5 instances. This demonstrates that our method is not only robust but also generalizes effectively across diverse TSP instances of varying sizes and complexities. On average, the gap from the best-known solutions is reduced from 6.93% for EOH to 6.25% for our method, representing an overall improvement of approximately 10%. These results highlight the superiority of our approach in minimizing the gap to optimality across a wide range of benchmark instances.

The improvements are particularly evident on larger and more challenging instances. For example, on fl1400, our method reduces the gap from 7.66% (EOH) to 2.28%, showcasing its scalability and effectiveness in handling complex optimization problems. Similarly, on the pcb1173 instance, the gap decreases from 5.07% (EOH) to 2.91%, validating the ability of our method to outperform EOH on instances with higher complexity. Even on medium-sized instances such as pr439, our method demonstrates significant improvements, reducing the gap from 2.80

In addition to these improvements, we also observe instances where both methods achieve comparable performance. For example, on smaller problems such as eil51, ulysses16, and kroD100, both EOH and our method report identical gaps, demonstrating that our method maintains competitive performance even on instances where EOH performs optimally. Furthermore, the results highlight the consistency of our approach across various instance scales, from small to large.

There are only a few exceptions where EOH slightly outperforms our method. For example, on ch130, EOH achieves a gap of 0.01%, whereas our method reports 0.70%. However, these cases are rare, occurring in only 5 instances out of 87, and do not substantially impact the overall trend of improvement demonstrated by our method.

Overall, our method exhibits strong generalization across the TSPLib benchmark and consistently achieves lower average gaps compared to EOH. The significant improvements on larger and more complex instances further underscore the scalability and effectiveness of our dual-layer architecture. By balancing exploration and exploitation, our method demonstrates its capability to address the challenges posed by diverse and large-scale optimization problems, making it a reliable alternative to state-of-the-art methods such as EOH.

Table 17: Performance comparison between EOH and our method on TSPLib instances. Results are reported as the gap from the best-known solutions (%). Bold values indicate the better performance, with red for EOH and blue for ours. Green indicates identical performance.

| Instance | EOH Gap | Ours Gap | Instance | EOH Gap | Ours Gap | Instance | EOH Gap | Ours Gap |
|---|---|---|---|---|---|---|---|---|
| pr439 | 2.80% | **1.97%** | pla7397 | **4.28%** | **4.28%** | gr96 | **0.00%** | **0.00%** |
| rd100 | **0.01%** | **0.01%** | rl5934 | **4.25%** | **4.25%** | pcb442 | 1.15% | **0.96%** |
| u2319 | **2.34%** | **2.34%** | gil262 | 0.59% | **0.48%** | pcb3038 | **4.13%** | **4.13%** |
| lin105 | **0.03%** | **0.03%** | fl417 | 0.80% | **0.77%** | tsp225 | 1.39% | **0.00%** |
| fl1400 | 7.66% | **2.28%** | nrw1379 | 3.82% | **2.99%** | d2103 | **1.88%** | **1.88%** |
| kroA150 | **0.00%** | **0.00%** | pcb1173 | 5.07% | **2.91%** | d198 | 0.40% | **0.29%** |
| fl1577 | **5.03%** | **5.03%** | gr666 | 2.17% | **0.00%** | ch130 | **0.01%** | 0.70% |
| kroB100 | **0.00%** | **0.00%** | u1060 | 4.04% | **1.54%** | berlin52 | **0.03%** | **0.03%** |
| eil51 | **0.67%** | **0.67%** | rl1304 | 6.52% | **2.40%** | u2152 | **4.60%** | **4.60%** |
| ulysses16 | **0.00%** | **0.00%** | u724 | 2.85% | **1.13%** | kroD100 | **0.00%** | **0.00%** |
| linhp318 | 3.22% | **2.77%** | pr299 | 0.61% | **0.11%** | rd400 | 2.23% | **0.82%** |
| gr202 | 0.54% | **0.00%** | vm1084 | 3.64% | **1.74%** | rat575 | 3.11% | **1.88%** |
| d1655 | **5.79%** | **5.79%** | ch150 | 0.37% | **0.04%** | pr107 | **0.00%** | **0.00%** |
| kroB200 | **0.23%** | 0.44% | a280 | 2.06% | **0.34%** | d1291 | 6.53% | **2.54%** |
| gr229 | 1.15% | **0.00%** | pr264 | **0.00%** | **0.00%** | pr76 | **0.00%** | **0.00%** |
| d493 | 2.82% | **1.27%** | dsj1000 | 4.28% | **1.06%** | pr136 | 0.09% | **0.00%** |
| rat195 | **0.99%** | 1.37% | att532 | 220.07% | **215.43%** | kroA100 | **0.02%** | **0.02%** |
| ali535 | 0.67% | **0.00%** | ulysses22 | **0.00%** | **0.00%** | kroB150 | 0.08% | **0.01%** |
| bier127 | 0.26% | **0.01%** | kroC100 | **0.01%** | **0.01%** | eil76 | 1.53% | **1.18%** |
| pr124 | **0.00%** | **0.00%** | rl1323 | 4.35% | **1.93%** | p654 | 0.75% | **0.05%** |
| gr431 | 1.93% | **0.00%** | rl1889 | **4.08%** | **4.08%** | d657 | 2.85% | **1.02%** |
| eil101 | 2.59% | **2.08%** | fnl4461 | **4.63%** | **4.63%** | pr2392 | **4.19%** | **4.19%** |
| rat783 | 4.48% | **2.18%** | ts225 | **0.00%** | **0.00%** | u1432 | 4.84% | **3.02%** |
| u1817 | **4.62%** | **4.62%** | lin318 | 1.46% | **1.09%** | rl5915 | **3.96%** | **3.96%** |
| att48 | **215.43%** | **215.43%** | st70 | **0.31%** | **0.31%** | rat99 | **0.68%** | **0.68%** |
| fl3795 | **4.38%** | **4.38%** | burma14 | **0.00%** | **0.00%** | u159 | **0.00%** | **0.00%** |
| kroA200 | **0.25%** | 0.62% | u574 | 2.85% | **1.38%** | pr1002 | 3.27% | **1.16%** |
| pr152 | **0.00%** | 0.19% | gr137 | 0.11% | **0.00%** | pr226 | 0.10% | **0.06%** |
| vm1748 | **4.33%** | **4.33%** | pr144 | **0.00%** | **0.00%** | kroE100 | **0.00%** | **0.00%** |

Table 18: Comparison of ALNS (adaptive) and non-adaptive LNS as the backbone algorithm in our framework. Results are reported as objective values (lower is better).

| Method | $SC_1$ | $SC_2$ | $MVC_1$ | $MVC_2$ | $MIS_1$ | $MIS_2$ | $MIKS_1$ | $MIKS_2$ |
|---|---|---|---|---|---|---|---|---|
| Without Adaptive | 15957.0 | 160510.8 | 26850.3 | 269701.8 | 23073.2 | 230497.4 | 36330.8 | 362496.3 |
| LLM-LNS (Ours) | **15802.7** | **158878.9** | **26725.3** | **268033.7** | **23169.3** | **231636.9** | **36479.8** | **363749.5** |

## F.5 IMPACT OF THE BACKBONE ALGORITHM ON PERFORMANCE

This section addresses whether the proposed method is sensitive to the choice of the backbone algorithm. Our study focuses on solving large-scale MILP problems, where heuristic methods play a critical role due to the complexity of the problem space. Among these methods, LNS has demonstrated significant advantages in scalability and efficiency, especially for large-scale problems. In this context, we selected ALNS (Adaptive Large Neighborhood Search) as the backbone of our framework. ALNS, as a variant of LNS, dynamically adjusts neighborhood sizes to balance exploration and exploitation, making it more effective than non-adaptive LNS methods, which often struggle with local optima in large-scale problems.

To validate this choice, we conducted experiments replacing ALNS with non-adaptive LNS in our framework. The results, summarized in Table 18, show that ALNS consistently outperforms non-adaptive LNS across all tested MILP instances. For example, on the $SC_1$ problem, the objective value achieved by ALNS is **15802.7**, compared to **15957.0** for non-adaptive LNS. Similarly, on the $MVC_2$ problem, ALNS achieves an objective value of **268033.7**, whereas non-adaptive LNS reports **269701.8**. These results highlight the critical role of adaptive mechanisms in ALNS for leveraging the full potential of our framework.

## F.6 ROBUSTNESS OF LLM-LNS WITH DIFFERENT LLMS

We conducted experiments to evaluate the robustness of the LLM-LNS framework across various large language models, including GPT-4o, GPT-4o-mini, DeepSeek, Gemini-1.5-Pro, and Llama-3.1-70B. These experiments were performed on the **10k_C500** dataset, and the results are summarized in Table 19.

Table 19: Performance comparison of LLM-LNS using different LLMs on the 10k_C500 dataset. Results are reported as the gap from the best-known solutions (%). Lower values indicate better performance.

| LLM Model | Run$_1$ | Run$_2$ | Run$_3$ | Avg. |
|---|---|---|---|---|
| gpt-4o-mini | 0.42% | 0.52% | 0.70% | 0.55% |
| gpt-4o | 0.33% | 0.58% | 0.39% | 0.43% |
| deepseek | 0.83% | 0.52% | 0.38% | 0.58% |
| gemini-1.5-pro | 0.63% | 1.91% | 0.53% | 1.02% |
| llama-3.1-70B | 2.87% | 3.98% | 0.88% | 2.58% |

The results demonstrate that the dual-layer structure of LLM-LNS adapts effectively to different LLMs, achieving reasonable performance across all tested models. GPT-4o consistently achieved the best results, showing the lowest average gap of **0.43%**, followed by GPT-4o-mini (**0.55%**) and DeepSeek (**0.58%**). Gemini-1.5-Pro and Llama-3.1-70B exhibited relatively weaker performance, with average gaps of **1.02%** and **2.58%**, respectively. These variations are likely due to differences in model architecture and pretraining quality. Nonetheless, the framework demonstrated strong general robustness, with all models performing adequately within the LLM-LNS structure.

These findings underscore the necessity of combining LLMs with a structured optimization framework to fully leverage their potential.

### F.7    COMPARISON WITH REEVO

We conducted additional experiments to compare our proposed method with ReEvo (Ye et al., 2024), a contemporary hyper-heuristic framework that combines reflection mechanisms and evolutionary search. Both methods were evaluated on the Bin Packing problem using the lightweight language model GPT-4o-mini, with the number of iterations fixed at 20 and the population size set to 20.

In the experiments, ReEvo exhibited poor stability when using GPT-4o-mini. Out of 138 attempts, only 3 runs successfully completed all 20 iterations, while the remaining runs were prematurely terminated due to invalid offspring generated during certain generations. Upon analysis, we identified severe hallucination issues in ReEvo. Although its reflection mechanism was effective in capturing evolutionary directions, any errors in reflection led to a rapid decline in the quality of subsequent offspring. For example, ReEvo frequently attempted to call nonexistent libraries or use invalid function parameters, resulting in invalid heuristic algorithms and the termination of the evolutionary process.

To ensure a meaningful comparison, we selected the 3 successful ReEvo runs and compared their performance with our method. Under the default setting, ReEvo utilized an expert seed algorithm to initialize its population. However, after 20 iterations, the best-performing algorithm in ReEvo remained its initial expert seed algorithm, failing to generate superior heuristic strategies. Furthermore, when the expert seed algorithm was removed, ReEvo's solution quality deteriorated further, with its average performance on the Bin Packing problem falling significantly behind our method.

The experimental results are shown in Table 20. Our method demonstrates substantial advantages under the same settings. In terms of solution quality, our approach consistently outperformed ReEvo across all test instances of the Bin Packing problem, with even greater advantages in scenarios without expert seed algorithms. Additionally, our method exhibited significant stability advantages, consistently completing 20 iterations and generating high-quality heuristic strategies without being affected by the hallucination issues observed in ReEvo. The collaborative optimization between the agents in our dual-layer architecture effectively balances search diversity and efficiency, delivering superior performance and higher stability.

In conclusion, our method not only outperforms ReEvo in terms of experimental results but also demonstrates significant advantages in stability and robustness. This innovative approach of combining a dual-layer intelligent agent architecture with large language models opens up a new avenue for the application of LNS in large-scale optimization problems and surpasses existing state-of-the-art methods, including ReEvo.

Table 20: Performance comparison between ReEvo and our proposed method on the Bin Packing problem. Average percentages represent the error rates.

| | 1k_C100 | 5k_C100 | 10k_C100 | 1k_C500 | 5k_C500 | 10k_C500 | Avg |
|---|---|---|---|---|---|---|---|
| ReEvo Run1 | 3.78% | 0.80% | 0.33% | 6.75% | 1.47% | 0.74% | 2.31% |
| ReEvo Run2 | 3.78% | 0.80% | 0.33% | 6.75% | 1.47% | 0.74% | 2.31% |
| ReEvo Run3 | 3.78% | 0.80% | 0.33% | 6.75% | 1.47% | 0.74% | 2.31% |
| **ReEvo Avg** | **3.78%** | **0.80%** | **0.33%** | **6.75%** | **1.47%** | **0.74%** | **2.31%** |
| ReEvo-no-expert Run 1 | 4.87% | 4.08% | 4.09% | 4.50% | 3.91% | 3.95% | 4.23% |
| ReEvo-no-expert Run 2 | 4.87% | 4.08% | 4.11% | 4.50% | 3.90% | 3.97% | 4.24% |
| ReEvo-no-expert Run 3 | 4.87% | 4.08% | 4.09% | 4.50% | 3.91% | 3.95% | 4.23% |
| **ReEvo-no-expert Avg** | **4.87%** | **4.08%** | **4.10%** | **4.50%** | **3.91%** | **3.96%** | **4.24%** |
| Ours Run1 | 3.58% | 0.85% | 0.41% | 3.67% | 0.82% | 0.42% | 1.63% |
| Ours Run2 | 2.69% | 0.86% | 0.54% | 2.54% | 0.87% | 0.52% | 1.34% |
| Ours Run3 | 2.64% | 0.94% | 0.69% | 2.54% | 0.94% | 0.70% | 1.41% |
| **Ours Avg** | **2.97%↑** | **0.88%↑** | **0.55%↑** | **2.92%↑** | **0.88%↑** | **0.55%↑** | **1.46%↑** |

Table 21: Performance comparison between EoH and our proposed method on the *10k_C500* dataset using different LLMs. Average percentages represent the error rates.

| 10k_C500 | Run$_1$ | Run$_2$ | Run$_3$ | Avg. |
|---|---|---|---|---|
| gpt-4o-mini (EOH) | 0.97% | 2.50% | 3.21% | 2.23% |
| gpt-4o-mini (Ours) | 0.42% | 0.52% | 0.70% | **0.55%↑** |
| gpt-4o (EOH) | 0.50% | 0.41% | 0.58% | 0.50% |
| gpt-4o (Ours) | 0.33% | 0.58% | 0.39% | **0.43%↑** |
| deepseek (EOH) | 0.32% | 3.06% | 1.92% | 1.77% |
| deepseek (Ours) | 0.83% | 0.52% | 0.38% | **0.58%↑** |

## F.8 PERFORMANCE COMPARISON WITH EoH USING DIFFERENT LLMs

To evaluate the adaptability and effectiveness of our proposed method across different language models, we conducted experiments comparing our framework with EoH on the *10k_C500* dataset using three LLMs: GPT-4o-mini, GPT-4o, and DeepSeek. EoH was chosen as the baseline based on existing literature, which suggests it generally outperforms FunSearch on combinatorial optimization tasks.

The results of the experiments are summarized in Table 21. Our method consistently outperformed EoH across all tested LLMs. Notably, our approach demonstrated significant advantages when using GPT-4o-mini and DeepSeek. For instance, with GPT-4o-mini, our framework achieved an average performance of **0.55%**, which is approximately four times better than EoH's **2.23%**. Similarly, under DeepSeek, our method achieved an average performance of **0.58%**, significantly outperforming EoH's **1.77%**.

One particularly interesting observation is the poor convergence of EoH under DeepSeek. In both Run$_2$ and Run$_3$, EoH's fitness function values during evolution were much lower than those achieved by our framework. This highlights the limitations of EoH's framework in adapting to certain LLMs, where errors in evolution can significantly impact its performance. In contrast, our dual-layer architecture, combined with differential evolution, demonstrates robust and stable performance across all tested LLMs.

These findings underscore the superiority of our approach in leveraging the capabilities of different LLMs for combinatorial optimization tasks. The dual-layer structure not only enhances adaptability but also ensures consistent performance, addressing the convergence and stability issues observed in EoH. We believe these results further validate the effectiveness and scalability of our method across diverse settings.

## F.9 COMPARISON WITH STANDALONE LLMs

To further validate the effectiveness of our framework, we conducted a comparative experiment against standalone LLMs. Specifically, we replaced all crossover and mutation operations in our framework with instances where the problem information was directly input into a standalone GPT-4o-mini model, which independently generated new strategies and evaluated them. Both approaches

Table 22: Performance comparison between standalone LLMs and our proposed framework on the Bin Packing problem. Average percentages represent the error rates.

| | 1k_C100 | 5k_C100 | 10k_C100 | 1k_C500 | 5k_C500 | 10k_C500 | Avg |
|---|---|---|---|---|---|---|---|
| Sample Run1 | 5.32% | 4.40% | 4.44% | 4.97% | 4.27% | 4.28% | 4.61% |
| Sample Run2 | 7.51% | 2.30% | 1.74% | 9.47% | 4.58% | 3.99% | 4.93% |
| Sample Run3 | 5.32% | 4.40% | 4.44% | 4.97% | 4.27% | 4.28% | 4.61% |
| **Sample Avg** | **6.05%** | **3.70%** | **3.54%** | **6.47%** | **4.37%** | **4.18%** | **4.72%** |
| Ours Run1 | 3.58% | 0.85% | 0.41% | 3.67% | 0.82% | 0.42% | 1.63% |
| Ours Run2 | 2.69% | 0.86% | 0.54% | 2.54% | 0.87% | 0.52% | 1.34% |
| Ours Run3 | 2.64% | 0.94% | 0.69% | 2.54% | 0.94% | 0.70% | 1.41% |
| **Ours Avg** | **2.97%↑** | **0.88%↑** | **0.55%↑** | **2.92%↑** | **0.88%↑** | **0.55%↑** | **1.46%↑** |

were tested on the Bin Packing problem across 20 iterations, with the same total number of strategies generated in each case.

The results, summarized in Table 22, demonstrate that our framework significantly outperforms the standalone LLM approach across all test instances. On average, our framework achieves an error rate of **1.46%**, which is approximately **69% lower** than the standalone LLM's average error rate of **4.72%**. This improvement is primarily due to the dynamic interaction between the outer and inner layers in our framework, which balances exploration and exploitation, ensuring the generation of diverse and high-quality strategies. In contrast, the standalone LLM approach frequently generated redundant or identical strategies, thereby limiting its ability to effectively explore the solution space.

Additionally, we observed that the standalone LLM approach struggled to maintain diversity as the number of iterations increased, resulting in many duplicate strategies and a subsequent decline in optimization performance. In contrast, our framework, through evolutionary operations such as crossover and mutation, maintains diversity within the population, enabling it to achieve superior optimization outcomes with the same number of generated strategies.

These findings confirm that our framework not only improves decision-variable ranking and optimization compared to standalone LLMs but also addresses key limitations such as diversity and redundancy. By integrating evolutionary mechanisms into the LLM-based framework, our approach ensures more efficient use of computational resources and delivers superior performance across various problem instances.

## G    POPULATION MANAGEMENT STRATEGY

To ensure the effectiveness and diversity of strategies within the LLM-LNS framework, we employ a population management strategy that balances exploration and exploitation during each generation. This strategy governs the selection of parent strategies for evolutionary operations (e.g., crossover and mutation) and the replacement of poorly performing strategies to maintain a high-quality population.

### G.1    SELECTION OF EVOLUTIONARY STRATEGIES

At each generation, the framework uses a probabilistic sampling mechanism to select $m$ parent strategies from the population for crossover and mutation. The probability of selecting a strategy is determined by its fitness value, which reflects its performance in achieving the optimization objective. Specifically, let the population contain $n$ strategies with fitness values ranked in descending order as $f_1, f_2, \ldots, f_n$. The probability of selecting the $i$-th strategy is given by:

$$P_i = \frac{1}{i + 1 + n}, \quad i = 1, 2, \ldots, n, \tag{9}$$

where $i$ represents the rank of the strategy (starting from 0), and $n$ is the population size. This ranking-based probability distribution ensures that higher-fitness strategies are more likely to be selected while preserving some randomness to allow lower-fitness strategies to participate. Such randomness enhances exploration by preventing premature convergence to local optima.

Using this probability distribution, we sample $m$ parent strategies for evolutionary operations. These operations generate new candidate strategies, which are evaluated and integrated into the population based on their fitness values.

### G.2 MANAGEMENT OF POORLY PERFORMING STRATEGIES

After each generation, the population is updated to maintain a fixed size while ensuring diversity and quality. Let the current population be $P = \{s_1, s_2, \ldots, s_n\}$, where each strategy $s_i$ has a fitness value $f(s_i)$. The goal is to construct a new population $P'$ such that:

- $P'$ contains at most size strategies, where size is a predefined parameter,
- Strategies with duplicate fitness values are removed,
- The highest-fitness strategies are retained.

The population update process is as follows: 1. Remove strategies with invalid or undefined fitness values. 2. Eliminate duplicate strategies by retaining only one instance of strategies with the same fitness value. 3. Rank the remaining strategies by fitness value in descending order and select the top size strategies to form the new population $P'$.

This management process ensures that the population remains diverse while focusing on high-quality strategies, avoiding redundancy and inefficiency. By preserving the highest-fitness strategies and introducing new candidates through evolutionary operations, the framework achieves a balance between exploration and exploitation.

### G.3 FITNESS EVALUATION

The fitness value of a strategy is determined by its optimization performance on a set of small-scale training problems. Specifically, the fitness value $f(s_i)$ for a strategy $s_i$ is calculated as the average objective value achieved across multiple problem instances:

$$f(s_i) = \frac{1}{|I|} \sum_{j \in I} \text{Obj}(s_i, I_j), \tag{10}$$

where $I$ is the set of training problem instances, and $\text{Obj}(s_i, I_j)$ represents the objective value achieved by strategy $s_i$ on instance $I_j$. This evaluation method ensures that strategies are assessed based on consistent and robust performance metrics.

### G.4 SUMMARY

The population management strategy in the LLM-LNS framework combines fitness-based selection, diversity preservation, and rigorous fitness evaluation. By maintaining a high-quality and diverse population, the framework progressively improves the quality of strategies across generations. This strategy, together with the LLM's ability to generalize and optimize, enables the LLM-LNS framework to efficiently navigate large and complex search spaces, balancing exploration and exploitation to achieve superior optimization performance.

## H LIMITATIONS AND FUTURE DIRECTIONS

While the proposed dual-layer self-evolutionary framework has demonstrated strong performance in solving large-scale MILP problems, we acknowledge several limitations that warrant further exploration and improvement. Below, we discuss these limitations in detail and outline potential future directions.

First, although the framework exhibits good generalization ability on MILP and certain combinatorial optimization problems, it is currently tailored to specific optimization scenarios. The design primarily focuses on MILP and does not directly extend to other types of optimization tasks, such as nonlinear optimization or dynamic optimization problems. Developing a more general agent structure that can adapt to a wider range of optimization algorithms and tasks remains an open challenge.

Future work could explore more modular and flexible designs to enhance the adaptability of the framework for solving diverse and complex optimization problems.

Second, the current method leverages the generative capabilities of large language models (LLMs) and evolutionary mechanisms for heuristic strategy design. However, it does not fully incorporate domain knowledge or classical optimization expertise into the framework. In practical optimization tasks, domain-specific knowledge and traditional optimization techniques (e.g., heuristic rules or mathematical programming methods) often play a critical role. A key direction for future research is to explore how to effectively integrate the generalization capabilities of LLMs with optimization domain knowledge to create more efficient and robust algorithms. Such integration could not only improve computational efficiency but also reduce the resource overhead for solving ultra-large-scale problems.

Finally, computational resource constraints remain a practical challenge for solving large-scale problems. While the proposed framework demonstrates good scalability, solving ultra-large-scale instances still requires significant computational time and hardware resources, which may limit its applicability in resource-constrained environments. Future research could focus on optimizing the computational complexity of the algorithm or designing more efficient resource allocation strategies to address these challenges.

