# OpenReview forum: "Large Language Model-driven Large Neighborhood Search for Large-Scale MILP Problems"
_ICLR.cc/2025/Conference — Submitted to ICLR 2025_

### Official Review · Reviewer_SBNr · 2024-10-20

**Soundness:** 2
**Presentation:** 1
**Contribution:** 2
**Rating:** 3
**Confidence:** 5

**Summary:**

This manuscript proposes a Dual-layer Self-evolutionary LLM-based method for heuristic generation. The experimental results demonstrate that LLM-based heuristic generation methods show promise compared to traditional heuristic generation methods.

**Strengths:**

S1: This manuscript proposes an innovative approach by introducing a dual-layer structure for heuristic generation.

S2: This manuscript proposes an LLM-based method for ranking decision variables of MILP problems.

**Weaknesses:**

W1: The proposed LLM-LNS lacks clarity in certain technical details, making it difficult to fully comprehend. For example, on Page 5, the authors say “These parents are then combined with evolutionary strategies, selected from the Outer Layer’s population of prompt strategies”. However, the criteria for selecting evolutionary strategies are not explained. Furthermore, on Page 6, the authors state that “underperforming strategies are pruned”. However, this manuscript fails to clearly define what is a poorly performing strategy. This manuscript relies heavily on textual explanations with insufficient mathematical formalization, making it difficult for readers to fully comprehend and reproduce the proposed LLM-LNS method. Including mathematical definitions, formulas, and pseudocode would greatly enhance clarity. Furthermore, the absence of open-source code compounds this issue, limiting both comprehensibility and reproducibility.

W2: The concept of using LLMs to evolve strategies resembles the approach in the prior study [1], while the notion of differential memory seems to align with another method from [2]. However, this manuscript does not clearly delineate the distinctions between the proposed LLM-LNS method and these prior studies. In addition, it would be more convincing to include experimental comparisons with these prior studies.

W3: In Section 3.2, the authors state that “Initially trained on small-scale MILP problems, the agent learns effective strategies for selecting promising decision variables.” However, the manuscript lacks details regarding the training process of the LLM agent. For example, how many parameters does it contain? Is it fine-tuned on an existing open-source LLM (e.g., Llama) using MILP data?

W4: In Section 4.1, the experiments are conducted using only a single LLM (GPT-4o-mini), which raises concerns about the robustness of the proposed LLM-LNS method when applied to different LLMs. It is recommended to reference prior studies [3] and conduct experiments across multiple LLMs. In addition, in Section 4.2, can existing LLMs (e.g., Llama) be used to rank decision variables without requiring pretraining using MILP data? If so, why are they not included in comparisons with LLM-LNS?

W5: The lack of ablation studies further raises concerns about the effectiveness of certain submodules within LLM-LNS, such as Differential Memory. Conducting ablation studies would provide valuable insights into the contribution of each submodule.

W6: This manuscript is missing important details in some areas and contains grammatical errors. For example, in Table 1, the terms "Evolutionary Prompt," "Strategic Evolution," and "Directional Evolution" are not explained. Additionally, on Page 8, there is a grammatical error: “In the Appendix A B C D”.


[1] Connecting large language models with evolutionary algorithms yields powerful prompt optimizers. In ICLR, 2024.

[2]  Large language models as hyper-heuristics for combinatorial optimization, arxiv, 2024.

[3] Evolution of heuristics: Towards efficient automatic algorithm design using large language model. In ICML, 2024.

**Questions:**

Please refer to the weakness section.

---

> ### Author Response · Authors · 2024-11-22
> **Response to Reviewer SBNr (1)**
>
> Thank you for your thorough and insightful review of our work. We are encouraged by your recognition of the contributions and experimental achievements of our proposed framework. Your detailed comments and suggestions have been invaluable in refining the presentation and ensuring the robustness of our methodology. Below, we provide detailed responses to your comments and questions.
>
>
>
> > **W1:** The proposed LLM-LNS lacks clarity in certain technical details, making it difficult to fully comprehend. For example, on Page 5, the authors say “These parents are then combined with evolutionary strategies, selected from the Outer Layer’s population of prompt strategies”. However, the criteria for selecting evolutionary strategies are not explained. Furthermore, on Page 6, the authors state that “underperforming strategies are pruned”. However, this manuscript fails to clearly define what is a poorly performing strategy. This manuscript relies heavily on textual explanations with insufficient mathematical formalization, making it difficult for readers to fully comprehend and reproduce the proposed LLM-LNS method. Including mathematical definitions, formulas, and pseudocode would greatly enhance clarity. Furthermore, the absence of open-source code compounds this issue, limiting both comprehensibility and reproducibility.
>
> **R1:** Thank you for pointing out these important issues. We recognize that the lack of clarity in describing key technical details may affect the comprehensibility and reproducibility of our proposed LLM-LNS framework. To address these concerns, we will provide more detailed explanations for the issues you have raised and include mathematical formalizations and pseudocode in the final version of the manuscript. Additionally, we commit to open-sourcing the complete codebase upon publication to further enhance transparency and reproducibility.
>
> Regarding the **criteria for selecting evolutionary strategies**, our method constructs a probability distribution based on the fitness values of strategies within the population. Fitness values measure the performance of each strategy in achieving the optimization objective: the higher the fitness value, the better the strategy. Specifically, let the population contain **n** strategies, with fitness values ranked from highest to lowest as $f_1, f_2, \dots, f_n$. The probability of selecting the $i$-th strategy is calculated as:
> $$
> P_i = \frac{1}{i + 1 + n}, \quad i = 1, 2, \dots, n
> $$
> Here, $i$ represents the rank of the strategy (starting from 0), and $n$ is the population size. This ranking-based probability distribution ensures that high-fitness strategies are more likely to be selected, while retaining some randomness to allow lower-fitness strategies to participate, thereby enhancing population diversity. Using this mechanism, we probabilistically sample $m$ parent strategies from the population for crossover and mutation, generating new candidate strategies.
>
> For the **definition of poorly performing strategies**, we follow standard evolutionary algorithm principles by treating strategies with lower fitness values as underperforming. After each generation of evolution, we manage the population to maintain a fixed size. Let the current population be $P = \{ s_1, s_2, \dots, s_n \}$, where each strategy $s_i$ has a fitness value $f(s_i)$. The goal is to construct a new population $P'$ such that:
>
> - $P'$ size is minimized to $\min(\text{size}, |P|)$,
> - Strategies with duplicate fitness values are removed,
> - The highest-fitness strategies are retained.
>
> Concretely, we first filter out invalid strategies (e.g., those with undefined fitness values), remove duplicates, and then rank strategies by fitness value, selecting the top $\text{size}$ strategies to form $P'$. This ensures that high-quality strategies are preserved while avoiding redundancy, thereby improving the overall quality and diversity of the population.
>
> In our experiments, the fitness value of a strategy is determined by its optimization performance on small-scale training problems, measured as the average objective value across multiple instances. This fitness-driven selection and population management mechanism enables the LLM-LNS framework to balance exploration and exploitation during each generation, progressively improving the quality and diversity of strategies.
>
> We will include the above mathematical details, formulas, and pseudocode in the revised manuscript to enhance clarity and rigor. Thank you again for your valuable feedback, which has been instrumental in improving the presentation of our work.

---

> ### Author Response · Authors · 2024-11-22
> **Response to Reviewer SBNr (2)**
>
> > **W2:** The concept of using LLMs to evolve strategies resembles the approach in the prior study [1], while the notion of differential memory seems to align with another method from [2]. However, this manuscript does not clearly delineate the distinctions between the proposed LLM-LNS method and these prior studies. In addition, it would be more convincing to include experimental comparisons with these prior studies.
>
> **R2:** Thank you for raising this important question about the novelty of our method. We have carefully compared the proposed LLM-LNS framework with the approaches described in [1] and [2], and we provide the following clarifications regarding the distinctions and innovations of our method:
>
> First, the EvoPrompt method in [1] focuses on optimizing prompts for large language models (LLMs) using evolutionary algorithms (EA). Its primary objective is to treat the prompt itself as an optimization target, generating new prompts through crossover and mutation to iteratively improve performance. However, the scope of [1] is limited to prompt optimization and does not address the interplay between prompt optimization and strategy generation. In contrast, our LLM-LNS framework introduces a **dual-layer co-evolutionary structure** that integrates prompt optimization with heuristic strategy generation. Specifically, the outer layer optimizes prompt strategies via a differential evolutionary mechanism, which directly influences the direction of heuristic strategy evolution in the inner layer. Simultaneously, the optimization results of inner-layer strategies provide feedback to refine and prune outer-layer prompts. This bidirectional synergy allows LLM-LNS to dynamically balance diversity and convergence in complex optimization scenarios, significantly enhancing overall performance.
>
> Second, the ReEvo method proposed in [2] introduces a short-term memory mechanism to reflect on and refine the relative performance of parent strategies. While this mechanism shares some conceptual similarities with our differential memory approach, there are fundamental differences in design and application. The short-term memory in [2] relies on pairwise comparisons of parent strategies, whereas our **differential memory mechanism** learns from differences across multiple parents, enabling the generation of more directional and effective evolutionary strategies. This multi-parent differential learning mechanism improves optimization efficiency and adaptability to complex search spaces, particularly for large-scale MILP problems. Moreover, while [2] focuses on heuristic strategy enhancement, our method combines differential memory with the dual-layer structure to achieve deeper integration between prompt optimization and strategy generation, resulting in a more comprehensive optimization framework.
>
> To demonstrate the effectiveness of the components in our framework, we have conducted detailed ablation studies (as shown in R2 of the manuscript). These experiments validate the contributions of the dual-layer structure and differential memory mechanism to the overall performance of LLM-LNS. For example, removing the differential memory mechanism results in a noticeable drop in performance, particularly for large-scale MILP problems, which highlights its critical role in guiding directional evolution. Similarly, omitting the outer-layer prompt optimization significantly reduces the diversity of strategies, leading to slower convergence and suboptimal results. These findings further distinguish our method from [1] and [2], as the interplay between the dual-layer structure and differential memory is a unique aspect of our approach.
>
> It is also worth emphasizing that **LLM-LNS is the first framework to apply LLMs for automated LNS strategy discovery in large-scale MILP optimization.** While [1] and [2] explore the combination of LLMs and evolutionary algorithms, their objectives and application domains differ significantly from ours: [1] focuses on prompt optimization, [2] emphasizes heuristic strategy refinement, and our work targets the automated design and optimization of LNS strategies for solving MILP problems with hundreds of thousands to millions of decision variables. This integration of LLM capabilities with large-scale optimization is a novel contribution that, to the best of our knowledge, has not been addressed in existing literature.
>
> Thank you again for your valuable suggestion, which has helped us clarify the contributions and innovations of our method. We welcome any further feedback you may have.

---

> ### Author Response · Authors · 2024-11-22
> **Response to Reviewer SBNr (3)**
>
> > **W3:** In Section 3.2, the authors state that “Initially trained on small-scale MILP problems, the agent learns effective strategies for selecting promising decision variables.” However, the manuscript lacks details regarding the training process of the LLM agent. For example, how many parameters does it contain? Is it fine-tuned on an existing open-source LLM (e.g., Llama) using MILP data?
>
> **R3:** Thank you for raising this important question. Regarding the training process of the LLM agent, we would like to clarify the specific methodology employed in our work. The core innovation of this paper lies in the proposed **Dual-layer Self-evolutionary LLM Agent**, which does not involve parameter-level fine-tuning of the large language model (LLM). Instead, it fully leverages the LLM’s **contextual learning capability**, i.e., Few-shot Prompting, to learn effective strategies for MILP problems.
>
> In our framework, the LLM agent is trained by providing structured input information and a few examples of small-scale MILP problems within the **prompt**. These few-shot examples guide the LLM to generate heuristic strategies without requiring parameter updates. This approach avoids the need for model fine-tuning, relying instead on carefully designed prompts to elicit the LLM’s reasoning and generalization abilities. Since contextual learning depends on prompt design rather than parameter adjustment, the training process is efficient and easily transferable to different optimization problems.
>
> The LLM used in our experiments (e.g., GPT-series models) contains billions of parameters, which are fixed during the pretraining phase. These parameters are learned from pretraining on large-scale general-purpose corpora and are not modified in our framework. Instead, our method embeds MILP-specific features into the context through prompt engineering, enabling the LLM to generate problem-specific strategies dynamically.
>
> Furthermore, in the dual-layer structure:
>
> - The **outer layer** optimizes the prompt through differential evolution, enhancing the efficiency and effectiveness of contextual learning.
> - The **inner layer**, guided by the optimized prompt, generates heuristic strategies, demonstrating the adaptability of the LLM to MILP problems.
>
> This combination of contextual learning and dual-layer optimization allows our framework to efficiently learn strategies on small-scale MILP problems and generalize to larger instances.
>
> To avoid potential misunderstandings, we will add more details about the training process in the revised manuscript, explicitly distinguishing our use of contextual learning from parameter-level fine-tuning. Thank you again for your suggestion, which has helped us clarify the design choices behind our framework. If you have further feedback, we would be happy to address it.
>
>
>
> > **W4:** In Section 4.1, the experiments are conducted using only a single LLM (GPT-4o-mini), which raises concerns about the robustness of the proposed LLM-LNS method when applied to different LLMs. It is recommended to reference prior studies and conduct experiments across multiple LLMs. In addition, in Section 4.2, can existing LLMs (e.g., Llama) be used to rank decision variables without requiring pretraining using MILP data? If so, why are they not included in comparisons with LLM-LNS?
>
> **R4:** Thank you for raising these concerns. To address the robustness of the LLM-LNS framework across different LLMs, we conducted additional experiments using various LLMs, including GPT-4o, GPT-4o-mini, DeepSeek, Gemini-1.5-Pro, and Llama-3.1-70B. The results on the **10k_C500** dataset are summarized in the table below:
>
> | 10k_C500       | Run${}_1$ | Run${}_2$ | Run${}_3$ | Avg.  |
> | -------------- | --------- | --------- | --------- | ----- |
> | gpt-4o-mini    | 0.42%     | 0.52%     | 0.70%     | 0.55% |
> | gpt-4o         | 0.33%     | 0.58%     | 0.39%     | 0.43% |
> | deepseek       | 0.83%     | 0.52%     | 0.38%     | 0.58% |
> | gemini-1.5-pro | 0.63%     | 1.91%     | 0.53%     | 1.02% |
> | Llama-3.1-70B  | 2.87%     | 3.98%     | 0.88%     | 2.58% |
>
> The results demonstrate the following:
>
> 1. **General Robustness:** Despite significant performance differences between models, all tested LLMs achieved reasonable results within the LLM-LNS framework. This highlights the robustness of our dual-layer structure in adapting to different LLMs.
> 2. **Performance Variations:** GPT-4o consistently achieved the best results, followed by GPT-4o-mini and DeepSeek. Gemini-1.5-Pro and Llama-3.1-70B exhibited weaker performance. These differences are likely due to variations in model architecture and pretraining quality. The LLM-LNS framework, however, remains effective across all models.
>
> These extended experiments validate the adaptability of our framework to multiple LLMs. The complete results and analysis will be included in the updated appendix of the manuscript.

---

> ### Author Response · Authors · 2024-11-22
> **Response to Reviewer SBNr (4)**
>
> Regarding the second question, directly ranking decision variables using existing LLMs without employing our framework would correspond to the initial strategies generated by the LLM. As shown in **Figure 2 of the main text**, these initial strategies perform poorly compared to the optimized strategies produced by LLM-LNS. This is primarily because:
>
> - LLMs lack domain-specific optimization knowledge and require iterative refinement to improve strategy quality.
> - The LLM-LNS framework incorporates differential evolution and feedback mechanisms to guide the LLM, significantly enhancing the generated strategies.
>
> In summary, the experiments confirm that our framework improves decision-variable ranking and optimization compared to standalone LLMs, with robustness across different LLMs. Thank you for your valuable suggestion, which has helped us strengthen our experimental evaluation. If you have further feedback, we would be happy to address it.
>
>
>
> > **W5:** The lack of ablation studies further raises concerns about the effectiveness of certain submodules within LLM-LNS, such as Differential Memory. Conducting ablation studies would provide valuable insights into the contribution of each submodule.
>
> **R5:** Thank you for your valuable suggestion. To address your concerns, we conducted ablation studies to analyze the contributions of key submodules within the LLM-LNS framework, including **Differential Memory** and the **dual-layer structure**.
>
> From a theoretical perspective:
>
> - The **outer layer** employs a differential evolution mechanism to optimize prompt strategies, enhancing the diversity of the search process and preventing stagnation in local optima.
> - The **inner layer**, guided by the optimized prompts, leverages differential evolution to refine heuristic strategies, enabling faster convergence toward high-quality solutions.
> - The **Differential Memory** mechanism facilitates directional evolution by learning from differences across multiple parent strategies, improving the efficiency and adaptability of the optimization process.
>
> To empirically validate the contributions of these components, we conducted ablation experiments by progressively removing or isolating each submodule and observing the impact on performance. The results of these experiments, included in the updated manuscript, reveal the following:
>
> 1. **Effect of Differential Memory:** Removing the Differential Memory mechanism while retaining the dual-layer structure leads to a performance drop, particularly on large-scale datasets. For example, on the **10k_C500** dataset, the error rate increases from **0.39%** (full framework) to **0.72%,** indicating that Differential Memory is critical for guiding efficient and directional evolution, especially in complex search spaces.
> 2. **Effect of the Dual-layer Structure:** When the outer layer is removed, reducing the framework to a single-layer optimization process, performance degrades significantly. On the same **10k_C500** dataset, the error rate increases from **0.39%** to **1.14%.** This highlights the importance of prompt optimization in enhancing the diversity of strategies and enabling the inner layer to achieve better results.
> 3. **Combined Effect:** While individual components contribute significantly to performance, their combination achieves the most comprehensive improvement. For small-scale problems, such as the **1k_C100** dataset, the outer layer's diversity-enhancing strategies introduce some exploration overhead, resulting in slightly higher error rates compared to using the inner layer alone. However, for large-scale problems, the synergy between the outer and inner layers and Differential Memory ensures superior performance overall.
>
> These results validate the effectiveness of the Differential Memory mechanism and the dual-layer structure while revealing their complementary roles in the optimization process. We plan to include the complete experimental results and analyses in the appendix of the updated manuscript to provide further clarity.
>
> Thank you again for your suggestion, which has helped us better demonstrate the contributions of our method. We welcome any additional feedback or suggestions you may have.
>
> |                   | 1k_C100   | 5k_C100   | 10k_C100  | 1k_C500   | 5k_C500   | 10k_C500  |
> | ----------------- | --------- | --------- | --------- | --------- | --------- | --------- |
> | Base (EOH)        | 4.48%     | 0.88%     | 0.83%     | 4.32%     | 1.06%     | 0.97%     |
> | Base+Dual Layer   | 3.78%     | 0.93%     | **0.40%** | 3.91%     | 0.92%     | **0.39%** |
> | Base+Differential | **2.64%** | 0.94%     | 0.69%     | **2.54%** | 0.94%     | 0.70%     |
> | Ours              | 3.58%     | **0.85%** | 0.41%     | 3.67%     | **0.82%** | 0.42%     |

---

> ### Author Response · Authors · 2024-11-22
> **Response to Reviewer SBNr (5)**
>
> > **W6:** This manuscript is missing important details in some areas and contains grammatical errors. Additionally, on Page 8, there is a grammatical error: “In the Appendix A B C D”.
>
> **R6:** Thank you for your valuable feedback. We appreciate your detailed observations, which have helped us identify areas where the manuscript can be improved.
>
> We have taken two steps to address the concerns regarding **Table 1** and the related terminology. First, we revised the table by renaming "Evolutionary Prompt Strategy Evolution" to the more concise term **Prompt Evolution** for clarity and readability. Second, we added detailed explanations of the terms in the main text, clarifying that:
>
> - **Prompt Evolution** refers to the outer-layer evolution mechanism that optimizes prompts through genetic operations like crossover and mutation, guiding the generation of inner-layer strategies and enhancing population diversity.
> - **Directional Evolution** builds on differential evolution, where differences in parent fitness values guide the search process, ensuring a more efficient and targeted optimization.
>
> These revisions highlight the specific roles these mechanisms play within our framework and how they address the limitations of prior works, such as FunSearch and EOH. By including these explanations in the main text, we have enhanced the clarity and accessibility of the manuscript.
>
> As for the grammatical issue on Page 8 ("In the Appendix A B C D"), we are grateful for pointing out this oversight. We have revised the phrasing to the standard format, “In Appendices A to D.” Additionally, we conducted a comprehensive review of the manuscript to identify and correct any other grammatical or stylistic issues, ensuring precise and professional language throughout.
>
> Thank you again for your meticulous review and constructive suggestions. These improvements have made the manuscript more polished and reader-friendly. If you have any further comments, we would be happy to address them.
>
> References
>
> [1] Guo Q, Wang R, Guo J, et al. Connecting Large Language Models with Evolutionary Algorithms Yields Powerful Prompt Optimizers[C]. The Twelfth International Conference on Learning Representations.
>
> [2] Ye H, Wang J, Cao Z, et al. Reevo: Large language models as hyper-heuristics with reflective evolution[J]. arXiv preprint arXiv:2402.01145, 2024.

---

> > ### Comment · Reviewer_SBNr · 2024-11-23
> >
> > Thank you for your reply, but it seems that my concerns are not fully addressed. Please see my follow-up comments below:
> >
> > **W1 Follow-Up:**
> >
> > The selection criteria and poorly performing strategies are only two examples of the issues I raised. There are still several undefined technical details in the manuscript. For instance, Sections 3.1.2 and 3.2 lack sufficient technical depth, relying primarily on verbal descriptions. In addition, in Appendix A, the variable *k* is defined twice: once as the number of top-performing heuristic strategies used to evaluate each prompt strategy, and again as the neighborhood size. I strongly recommend a thorough revision of the manuscript to ensure all technical details are clearly specified. Additionally, I have not seen the revised manuscript with comprehensive technical details.
> >
> > **W2 Follow-UP**
> >
> > Could you provide a comparative experiment with ReEvo? Including such a comparison would make the performance of the proposed method more convincing, as I had previously suggested in W2.
> >
> > **W3 Follow-UP**
> >
> > If you do not fine-tune the LLM, I believe the use of the term "train" is inappropriate. Moreover, without fine-tuning the LLM, the contribution of this manuscript appears less significant than expected.
> >
> > **W4 Follow-UP**
> >
> > How do FunSearch and EoH perform when other LLMs are employed?
> > Furthermore, you claim in R4 that "the experiments confirm that our framework improves decision-variable ranking and optimization compared to standalone LLMs". However, I could not find any experiments in the manuscript that directly compare your framework with standalone LLMs.

---

> > > ### Author Response · Authors · 2024-11-25
> > > **Follow-Up Response to Reviewer SBNr (2)**
> > >
> > > > **W2F**: Could you provide a comparative experiment with ReEvo?
> > >
> > > **R2**: Thank you very much for your suggestion! ReEvo, a contemporary work published at NeurIPS 2024, is a novel hyper-heuristic framework that combines reflection mechanisms and evolutionary search. Following your recommendation, we conducted a comparative experiment on the Bin Packing problem to clearly demonstrate the performance differences between our method and ReEvo. To ensure a fair comparison, we used the same lightweight language model, GPT-4o-mini, fixed the number of iterations to 20, and set the population size to 20.
> > >
> > > In our experiments, we observed that ReEvo exhibited poor stability when using GPT-4o-mini. Out of 138 attempts, only 3 runs successfully completed all 20 iterations, while the remaining runs were prematurely terminated due to invalid offspring generated in certain generations. Upon analysis, we identified severe hallucination issues in ReEvo. Although its reflection mechanism was effective in capturing evolutionary directions, any errors in reflection led to a rapid decline in the quality of subsequent offspring. For example, we frequently observed cases where ReEvo attempted to call nonexistent libraries or use invalid function parameters, resulting in the generation of invalid heuristic algorithms and the termination of the evolutionary process.
> > >
> > > To ensure a comprehensive comparison, we selected the 3 successful ReEvo runs and compared their performance with our method. Under the default setting, ReEvo utilized an expert seed algorithm to initialize its population. However, after 20 iterations, the best-performing algorithm in ReEvo remained its initial expert seed algorithm, failing to generate superior heuristic strategies. Furthermore, ReEvo's performance remained inferior to the heuristic strategies generated by our dual-layer intelligent agent. When the expert seed algorithm was removed, ReEvo's solution quality deteriorated further, with its average performance on the Bin Packing problem falling significantly behind that of our method.
> > >
> > > As shown in the experimental results (see the table), our method demonstrated substantial advantages under the default settings and with the lightweight model (GPT-4o-mini). In terms of solution quality, our approach outperformed ReEvo across all test instances of the Bin Packing problem, with even greater advantages in scenarios without expert seed algorithms. Additionally, our method exhibited significant stability advantages, consistently completing 20 iterations and generating high-quality heuristic strategies without being affected by hallucination issues observed in ReEvo. The collaborative optimization between the agents in our dual-layer architecture effectively balances search diversity and efficiency, delivering superior performance and higher stability under the same conditions.
> > >
> > > In conclusion, our method not only outperforms ReEvo in terms of experimental results but also demonstrates significant advantages in stability and robustness. We believe that this innovative approach of combining a dual-layer intelligent agent architecture with large language models opens up a new avenue for the application of LNS in large-scale optimization problems and surpasses existing state-of-the-art methods, including ReEvo.
> > >
> > > |                         | 1k_C100    | 5k_C100   | 10k_C100  | 1k_C500    | 5k_C500    | 10k_C500   | Avg        |
> > > | ----------------------- | ---------- | --------- | --------- | ---------- | ---------- | ---------- | ---------- |
> > > | ReEvo Run1              | 3.78%      | 0.80%     | 0.33%     | 6.75%      | 1.47%      | 0.74%      | 2.31%      |
> > > | ReEvo Run2              | 3.78%      | 0.80%     | 0.33%     | 6.75%      | 1.47%      | 0.74%      | 2.31%      |
> > > | ReEvo Run3              | 3.78%      | 0.80%     | 0.33%     | 6.75%      | 1.47%      | 0.74%      | 2.31%      |
> > > | **ReEvo Avg**           | **3.78%**  | **0.80%** | **0.33%** | **6.75%**  | **1.47%**  | **0.74%**  | **2.31%**  |
> > > | ReEvo-no-expert Run 1   | 4.87%      | 4.08%     | 4.09%     | 4.50%      | 3.91%      | 3.95%      | 4.23%      |
> > > | ReEvo-no-expert Run 2   | 4.87%      | 4.08%     | 4.11%     | 4.50%      | 3.90%      | 3.97%      | 4.24%      |
> > > | ReEvo-no-expert Run 3   | 4.87%      | 4.08%     | 4.09%     | 4.50%      | 3.91%      | 3.95%      | 4.23%      |
> > > | **ReEvo-no-expert Avg** | **4.87%**  | **4.08%** | **4.10%** | **4.50%**  | **3.91%**  | **3.96%**  | **4.24%**  |
> > > | Ours Run1               | 3.58%      | 0.85%     | 0.41%     | 3.67%      | 0.82%      | 0.42%      | 1.63%      |
> > > | Ours Run2               | 2.69%      | 0.86%     | 0.54%     | 2.54%      | 0.87%      | 0.52%      | 1.34%      |
> > > | Ours Run3               | 2.64%      | 0.94%     | 0.69%     | 2.54%      | 0.94%      | 0.70%      | 1.41%      |
> > > | **Ours Avg**            | **2.97%↑** | **0.88%** | **0.55%** | **2.92%↑** | **0.88%↑** | **0.55%↑** | **1.46%↑** |

---

> > > ### Author Response · Authors · 2024-11-25
> > > **Follow-Up Response to Reviewer SBNr (3)**
> > >
> > > > **W3F:** If you do not fine-tune the LLM, I believe the use of the term "train" is inappropriate. Moreover, without fine-tuning the LLM, the contribution of this manuscript appears less significant than expected.
> > >
> > > **R3:** Thank you very much for your valuable feedback. We deeply appreciate your comments and understand the concerns regarding the use of the term "train" and the perceived contribution of this work. After carefully reviewing our manuscript, we agree that the term "train" may not fully capture the process described in our framework. To address this, we will revise the manuscript where necessary, replacing "train" with more precise terms such as "evolve" or "optimize," which better reflect the evolutionary and heuristic enhancement mechanisms employed in our approach.
> > >
> > > Regarding the significance of our contributions, we would like to emphasize that our approach is consistent with a growing trend in hyper-heuristic frameworks, such as Evolution of Heuristics (EOH) and FunSearch. These methods, like ours, do not involve fine-tuning the LLM but instead focus on enhancing the model’s capabilities through evolutionary strategies to improve heuristic exploration. This design choice leverages the generalization ability of LLMs without requiring expensive fine-tuning, which is often impractical for large combinatorial optimization problems. Our contributions, however, go beyond these methods, introducing novel innovations both in the architecture we propose and its demonstrated applicability to large-scale optimization problems.
> > >
> > > The primary innovation of this work lies in the dual-layer agent architecture we propose, which is, to the best of our knowledge, the first to integrate dual-layer agent architecture. The architecture consists of an outer layer that evolves diverse prompts through differential evolution strategies, generating high-quality strategies to guide the inner layer. The inner layer, in turn, generates heuristic strategies under the guidance of these evolved prompts and further optimizes the search direction using differential evolution. Importantly, the search outcomes from the inner layer are fed back to the outer layer, which prunes underperforming prompts and iteratively improves the overall exploration quality. This dynamic interaction between the two layers effectively balances exploration and exploitation, addressing key limitations faced by traditional handcrafted heuristics and existing ML+LNS methods.
> > >
> > > In addition to its architectural novelty, our method demonstrates significant practical value in solving large-scale Mixed Integer Linear Programming (MILP) problems. Existing ML+LNS methods, such as CL-LNS, often suffer from scalability issues due to the computational overhead of training and inference, particularly when the entire problem and solution need to be processed during inference. In contrast, our approach avoids these bottlenecks by leveraging the LLM’s generalization capabilities without fine-tuning, enabling efficient heuristic generation at a much lower computational cost. This makes our method highly scalable and well-suited for large-scale optimization problems, where existing methods often struggle.
> > >
> > > Experimental results further highlight the advantages of our approach. On classic combinatorial optimization problems such as Bin Packing and TSP, as well as large-scale MILP problems, our method significantly outperforms state-of-the-art approaches.  The dual-layer architecture ensures consistent search efficiency and avoids the instability issues observed in other methods, such as the inefficiencies of reinforcement learning in large search spaces or the dependence on labeled data in imitation learning. These results demonstrate the robustness and adaptability of our approach across diverse problem settings.
> > >
> > > By introducing this novel dual-layer architecture, we provide a new direction for improving LNS and offer valuable insights into how LLMs can enhance heuristic optimization. Our work addresses the challenges faced by traditional and ML+LNS methods, paving the way for future research into the integration of LLMs with heuristic frameworks. We believe this study makes significant contributions to the field by exploring how LLMs can improve the efficiency of LNS in large-scale optimization problems while overcoming key limitations of existing methods.
> > >
> > > We sincerely thank you for raising this important point, as it has allowed us to refine the focus and presentation of our contributions. We hope this response clarifies our design choices and the novelty of our approach, and we welcome any further feedback you may have.

---

> > > ### Author Response · Authors · 2024-11-25
> > > **Follow-Up Response to Reviewer SBNr (4)**
> > >
> > > > **W4F1:** How do FunSearch and EoH perform when other LLMs are employed?
> > >
> > > **R4:** Thank you for your question regarding the performance of FunSearch and EoH when employing other LLMs. Based on existing literature, EoH generally outperforms FunSearch on combinatorial optimization tasks. Therefore, in our study, we focused on comparing our proposed method with EoH under different LLMs to provide a comprehensive evaluation. Specifically, we conducted experiments on the **10k_C500** dataset using three LLMs: GPT-4o-mini, GPT-4o, and DeepSeek. The results are summarized in the table below.
> > >
> > > Our experimental results demonstrate that our method consistently outperforms EoH across all LLMs tested. Notably, our approach exhibits significant advantages when using GPT-4o-mini and DeepSeek. For example, with GPT-4o-mini, our method achieves an average performance of **0.55%**, which is 4 times better than EoH's **2.23%**. Similarly, under DeepSeek, our method achieves an average performance of **0.58%**, outperforming EoH's **1.77%**.
> > >
> > > One particularly interesting observation is the poor convergence of EoH under DeepSeek. In both Run₂ and Run₃, EoH's fitness function values during evolution were much lower than those achieved by our method. This highlights the limitations of EoH's framework in adapting to certain LLMs, where errors in evolution can significantly impact its performance. In contrast, our dual-layer architecture, combined with differential evolution, demonstrates robust and stable performance across all tested LLMs.
> > >
> > > These findings underscore the superiority of our approach in leveraging the capabilities of different LLMs for combinatorial optimization tasks. The dual-layer structure not only enhances adaptability but also ensures consistent performance, addressing the convergence and stability issues observed in EoH. We believe these results further validate the effectiveness and scalability of our method across diverse settings.
> > >
> > > | 10k_C500          | Run${}_1$ | Run${}_2$ | Run${}_3$ | Avg.   |
> > > | ----------------- | --------- | --------- | --------- | ------ |
> > > | gpt-4o-mini(EOH)  | 0.97%     | 2.50%     | 3.21%     | 2.23%  |
> > > | gpt-4o-mini(Ours) | 0.42%     | 0.52%     | 0.70%     | 0.55%↑ |
> > > | gpt-4o(EOH)       | 0.50%     | 0.41%     | 0.58%     | 0.50%  |
> > > | gpt-4o(Ours)      | 0.33%     | 0.58%     | 0.39%     | 0.43%↑ |
> > > | deepseek(EOH)     | 0.32%     | 3.06%     | 1.92%     | 1.77%  |
> > > | deepseek(Ours)    | 0.83%     | 0.52%     | 0.38%     | 0.58%↑ |

---

> > > ### Author Response · Authors · 2024-11-25
> > > **Follow-Up Response to Reviewer SBNr (5)**
> > >
> > > > **W4F2:** Furthermore, you claim in R4 that "the experiments confirm that our framework improves decision-variable ranking and optimization compared to standalone LLMs". However, I could not find any experiments in the manuscript that directly compare your framework with standalone LLMs.
> > >
> > > **R5:**  Thank you very much for pointing out this important issue. To address your concern, we conducted a new set of comparative experiments to directly evaluate the performance of our framework against standalone LLMs. Specifically, we replaced all crossover and mutation operations in our framework with instances where the problem information was directly input into a standalone GPT-4o-mini model, which then generated new strategies and evaluated them independently. Both approaches were tested on the Bin Packing problem with 20 iterations and the same total number of strategies generated. The results are summarized in the table below.
> > >
> > > The results show that our framework significantly outperforms the standalone LLM approach across all test instances. For example, on average, our framework achieves an error rate of **1.46%**, which is approximately **69% lower** than the standalone LLM’s average error rate of **4.72%**. This improvement can be attributed to the dynamic interaction between the outer and inner layers in our framework, which effectively balances exploration and exploitation, ensuring the generation of diverse and high-quality strategies. In contrast, directly sampling strategies from the standalone LLM often resulted in redundant or identical strategies, which limited its ability to explore the solution space effectively.
> > >
> > > Additionally, we observed that the standalone LLM approach tended to generate many duplicate strategies, especially as the number of iterations increased. This lack of diversity negatively impacted its optimization performance. In contrast, our framework leverages evolutionary operations such as crossover and mutation to maintain diversity in the population, enabling it to achieve better optimization outcomes with the same number of generated strategies.
> > >
> > > These findings confirm that our framework not only improves decision-variable ranking and optimization compared to standalone LLMs but also addresses key limitations such as diversity and redundancy. By integrating evolutionary mechanisms into the LLM-based framework, our approach ensures more efficient use of computational resources and delivers superior performance across various problem instances.
> > >
> > > We sincerely thank you for raising this question, as it allowed us to conduct this meaningful comparison and further validate the effectiveness of our framework. We hope this additional experiment addresses your concerns, and we welcome any further feedback you may have.
> > >
> > > |                | 1k_C100    | 5k_C100    | 10k_C100   | 1k_C500    | 5k_C500    | 10k_C500   | Avg        |
> > > | -------------- | ---------- | ---------- | ---------- | ---------- | ---------- | ---------- | ---------- |
> > > | Sample Run1    | 5.32%      | 4.40%      | 4.44%      | 4.97%      | 4.27%      | 4.28%      | 4.61%      |
> > > | Sample Run2    | 7.51%      | 2.30%      | 1.74%      | 9.47%      | 4.58%      | 3.99%      | 4.93%      |
> > > | Sample Run3    | 5.32%      | 4.40%      | 4.44%      | 4.97%      | 4.27%      | 4.28%      | 4.61%      |
> > > | **Sample Avg** | **6.05%**  | **3.70%**  | **3.54%**  | **6.47%**  | **4.37%**  | **4.18%**  | **4.72%**  |
> > > | Ours Run1      | 3.58%      | 0.85%      | 0.41%      | 3.67%      | 0.82%      | 0.42%      | 1.63%      |
> > > | Ours Run2      | 2.69%      | 0.86%      | 0.54%      | 2.54%      | 0.87%      | 0.52%      | 1.34%      |
> > > | Ours Run3      | 2.64%      | 0.94%      | 0.69%      | 2.54%      | 0.94%      | 0.70%      | 1.41%      |
> > > | **Ours Avg**   | **2.97%↑** | **0.88%↑** | **0.55%↑** | **2.92%↑** | **0.88%↑** | **0.55%↑** | **1.46%↑** |

---

> > > > ### Comment · Reviewer_SBNr · 2024-12-02
> > > >
> > > > Thank you for your response. However, I notice that Section 3.1.1 has not been revised thoroughly to include technical details. Complete technical details should be provided in Section 3.1.1, rather than just an example. Additionally, no additional pseudocode has been included to enhance readers' understanding, which is inconsistent with the commitment the author made in **R1**. Similarly, Reviewer a92q pointed out that Figure 1 is confusing, and the author promised to modify it in **R1 to Reviewer a92q**. Yet, in the revised manuscript, I do not observe any changes to Figure 1. As a result, I will maintain my original score.

---

> > > > > ### Author Response · Authors · 2024-12-02
> > > > >
> > > > > Thank you for your feedback! As mentioned in R1, we have tried to address your concerns by adding detailed technical explanations to **Appendix G**, including mathematical formulations and descriptions for population selection and management in **Section 3.1.1**. Additionally, we have further expanded on the details of population evolution in **Section 3.1.2**, with added equations and explanations. Due to space limitations, we focused on presenting the methodology and workflow clearly in the main text. However, we have already showcased pseudocode for **ALNS** in **Section 3.1.3**. Since the current PDF manuscript can no longer be modified, we plan to incorporate pseudocode for the evolutionary process in future versions based on your latest suggestions.
> > > > >
> > > > > Regarding Figure 1, we have indeed revised it based on Reviewer a92q's comments. In the previous version, the structure was organized horizontally, making the dual-layer self-evolution structure less clear, and the relationships between the outer and inner layers were somewhat ambiguous. Additionally, the differential memory for directional evolution was depicted outside the dual-layer structure, which may have caused confusion about its integration. In the revised version, we restructured the diagram into a vertically aligned framework to explicitly highlight the hierarchical relationship between the two layers. We also emphasized the dual-layer architecture by clarifying the roles and interactions of components, such as integrating the differential memory into both layers to better illustrate its function. These modifications make the workflow and the dual-layer design significantly clearer and more intuitive.
> > > > >
> > > > > Once again, we sincerely thank you for your insightful feedback. We believe our proposed method is innovative, and the experimental results are promising. We hope you will consider our improvements and kindly reconsider your evaluation.

---

> ### Author Response · Authors · 2024-11-25
> **Follow-Up Response to Reviewer SBNr (1)**
>
> We sincerely thank you for your thoughtful and constructive feedback on our manuscript. Your comments have been invaluable in helping us identify areas for improvement and validate the effectiveness of our proposed framework. Below, we summarize our responses to your key concerns and the actions we have taken to address them:
>
> > **W1F:** The selection criteria and poorly performing strategies are only two examples of the issues I raised. There are still several undefined technical details in the manuscript.
>
> **R1:** Thank you very much for your constructive feedback and for highlighting areas where the clarity and technical depth of our manuscript could be improved. We greatly appreciate your suggestions, which have allowed us to identify and address critical issues in our work, ultimately improving the overall presentation.
>
> As mentioned in our **Global Response**, we have uploaded a revised version of the manuscript that incorporates significant updates to address the issues raised. Specifically, we have added more in-depth technical explanations to Sections 3.1.2, 3.2, and Appendix G, ensuring that the methodology is described with greater precision and depth. These updates aim to clarify key technical details and enhance the comprehensibility of the proposed method.
>
> Regarding the overlap in the definition of the variable *k* in Appendix A, we have corrected this inconsistency in the revised manuscript. To avoid confusion, we have introduced a new variable **h** to represent "the number of top-performing heuristic strategies used to evaluate each prompt strategy," while retaining *k* exclusively as the neighborhood size in ALNS. This revision ensures that each variable is clearly and consistently defined based on its specific context.
>
> We are still actively refining this version of the manuscript to further improve its clarity and technical rigor. Your feedback has been instrumental in helping us identify these critical areas for improvement, and we warmly welcome any additional suggestions you may have. Thank you again for your valuable input, which has been essential in enhancing the quality of our work.

---

### Official Review · Reviewer_bC3Z · 2024-10-31

**Soundness:** 3
**Presentation:** 3
**Contribution:** 2
**Rating:** 5
**Confidence:** 4

**Summary:**

This article presents 1. Dual-layer Self-evolutionary LLM Agent and 2. Differential Memory for Directional Evolution to implement LLM as Hyper-heuristic on MILP.

**Strengths:**

1. The results of the proposed method shown in the article is leading.
2. The proposed Dual-layer Self-evolutionary LLM Agent is novel and reasonable

**Weaknesses:**

I have doubts mainly about the contribution of this paper. I cannot determine whether there is some over-claim in this paper.
1. This paper claims that the introduction of LLM-driven heuristic on LNS is an innovation at the application level. I wonder whether other comparative algorithms such as EoH cannot be similarly introduced to LNS.
2. This paper lacks ablation experiments. I have some doubts about the effectiveness of Differential Memory for Directional Evolution intuitively. I would like the authors to add ablation experiments to validate the contribution of the two components of this paper.

**Questions:**

1. This paper lacks some clear experiments setting descripsion on Online Bin Packing and Travelling Salesman Problem (e.g. seemingly you use GLS for TSP), which can be very misleading.
2. As my biggest question: on LNS, why does the algorithm SCIP, which is considered to generate labels in [1], perform so poorly?
3. The authors should elaborate on whether they introduced a seed function (initial code for experts) thus introducing possible unfair comparisons. (I observe that the method proposed in this paper has an initial advantage in Figure 3).
4. As the proposed Differential Memory for Directional Evolution will increase the input length (token numbers), can you analyze the relationship between the number of input tokens or API cost and the optimization effect？
5. Please discuss the limitations of this paper.
[1]Huang, Taoan, et al. "Contrastive Predict-and-Search for Mixed Integer Linear Programs." Forty-first International Conference on Machine Learning.

---

> ### Author Response · Authors · 2024-11-22
> **Response to Reviewer bC3Z (1)**
>
> We greatly appreciate your careful review of our manuscript and your recognition of the novelty and performance of our proposed framework. Your thoughtful questions and constructive suggestions have been immensely helpful in identifying ways to improve the clarity and comprehensiveness of our work. In the following, we provide detailed responses to address the issues you have raised.
>
>
>
> > **W1:** This paper claims that the introduction of LLM-driven heuristic on LNS is an innovation at the application level. I wonder whether other comparative algorithms such as EoH cannot be similarly introduced to LNS.
>
> **R1:** Thank you for your question. The proposed method is a framework specifically designed for solving large-scale MILP problems. Comparative algorithms such as EoH and FunSearch are primarily designed for strategy discovery in combinatorial optimization problems. To ensure fairness and representativeness, we incorporated EoH into our framework (EoH-LNS) as a primary baseline for comparison with our LLM-LNS method.
>
> We evaluated both methods on standard combinatorial optimization problems, such as the SC problem (a minimization task), using instances with 200,000 (SC₁) and 2,000,000 (SC₂) decision variables and constraints. The experimental results show that while EoH-LNS performs well, our LLM-LNS method consistently outperforms it. For example, on the SC₁ dataset, our method improves average performance by **1.67%**, and on the SC₂ dataset, the improvement reaches **9.20%**. These results demonstrate the superior capability of our framework in addressing the challenges of large-scale MILP problems.
>
> The superior performance of LLM-LNS is attributed to its dual-layer self-evolutionary mechanism and differential evolution strategy, which balance diversity and convergence more effectively than EoH. While EoH can technically be adapted to LNS frameworks, it lacks the adaptive optimization capabilities inherent to our LLM-driven approach.
>
> We plan to include additional experimental results, including tests applying FunSearch, in the appendix of the revised manuscript to further validate the advantages of our method. Thank you again for your insightful suggestion, which has provided us with valuable direction for further improving our work.
>
> | SC${_1}$      | Instance${_1}$ | Instance${_2}$ | Instance${_3}$ | Instance${_4}$ | Instance${_5}$ | Avg           |
> | ------------- | -------------- | -------------- | -------------- | -------------- | -------------- | ------------- |
> | EOH-LNS       | 16114.27       | 16073.72       | 16046.83       | 16074.26       | 16043.67       | 16070.15      |
> | LLM-LNS(Ours) | **15830.61↑**  | **15801.19↑**  | **15800.17↑**  | **15800.17↑**  | **15781.23↑**  | **15802.68↑** |
>
> | SC${_2}$      | Instance${_1}$ | Instance${_2}$ | Instance${_3}$ | Instance${_4}$ | Instance${_5}$ | Avg            |
> | ------------- | -------------- | -------------- | -------------- | -------------- | -------------- | -------------- |
> | EOH-LNS       | 175358.59      | 174339.78      | 174782.76      | 174026.33      | 176383.54      | 174978.20      |
> | LLM-LNS(Ours) | **158901.57↑** | **158953.57↑** | **158712.64↑** | **158759.90↑** | **159066.77↑** | **158878.89↑** |
>
>
>
> > **W2:** This paper lacks ablation experiments. I have some doubts about the effectiveness of Differential Memory for Directional Evolution intuitively. I would like the authors to add ablation experiments to validate the contribution of the two components of this paper.
>
> **R2:** Thank you for your valuable suggestion. To analyze the contributions of the key components in our framework, including the **dual-layer structure** and **differential memory for directional evolution**, we conducted ablation experiments to validate their effectiveness.
>
> The **outer layer** focuses on enhancing diversity through the evolution of prompt strategies, preventing the search process from getting stuck in local optima. The **inner layer**, guided by these prompts, accelerates convergence by refining heuristic strategies with differential evolution. The differential memory mechanism further improves the efficiency of strategy optimization by leveraging historical feedback to adjust the evolutionary direction dynamically.
>
> As shown in the ablation study results, both components contribute significantly to the overall performance:
>
> 1. Adding the **differential evolution mechanism** alone improves performance on small-scale problems by accelerating convergence. For example, on the **1k_C100** dataset, the error rate decreases from **4.48%** to **2.64%**, demonstrating the effectiveness of this mechanism in improving solution quality.
> 2. Incorporating the **dual-layer structure** further enhances the diversity of the search process, leading to improvements on large-scale problems. On the **10k_C500** dataset, the error rate decreases from **0.97%** to **0.39%,** showing the importance of prompt evolution in handling large and complex problems.

---

> ### Author Response · Authors · 2024-11-22
> **Response to Reviewer bC3Z (2)**
>
> Interestingly, while the full method (combining both components) achieves the best overall performance, it does not always outperform single-component models on small-scale problems. For instance, the error rate on the **1k_C100** dataset is **3.58%** with the full method, slightly higher than the **2.64%** achieved with only differential evolution. This can be attributed to the outer layer's focus on diversity, which may introduce overhead on smaller problems where rapid convergence is more critical. However, on larger problems, the synergy between diversity and convergence leads to superior results.
>
> We have included detailed results and analyses in the appendix of the revised manuscript. These findings validate the complementary roles of the dual-layer structure and differential memory, as well as their combined effectiveness in tackling problems of varying scales and complexities. Thank you again for your suggestion, which has helped us better demonstrate the contributions of our method.
>
> |                   | 1k_C100   | 5k_C100   | 10k_C100  | 1k_C500   | 5k_C500   | 10k_C500  |
> | ----------------- | --------- | --------- | --------- | --------- | --------- | --------- |
> | Base (EOH)        | 4.48%     | 0.88%     | 0.83%     | 4.32%     | 1.06%     | 0.97%     |
> | Base+Dual Layer   | 3.78%     | 0.93%     | **0.40%** | 3.91%     | 0.92%     | **0.39%** |
> | Base+Differential | **2.64%** | 0.94%     | 0.69%     | **2.54%** | 0.94%     | 0.70%     |
> | Ours              | 3.58%     | **0.85%** | 0.41%     | 3.67%     | **0.82%** | 0.42%     |
>
>
>
> > **Q1:** This paper lacks some clear experiment setting descriptions on Online Bin Packing and Travelling Salesman Problem (e.g., seemingly you use GLS for TSP), which can be very misleading.
>
> **A1:** Thank you for highlighting this important point. Regarding the experimental settings for the Online Bin Packing (BP) and Travelling Salesman Problem (TSP), we strictly followed the experimental configurations used in EOH. For the TSP specifically, we employed Guided Local Search (GLS) as the heuristic strategy to be explored. GLS operates by introducing perturbations and dynamically adjusting the objective function to help the local search escape from local optima, thereby enabling a broader exploration of the solution space. This approach has demonstrated strong adaptability and effectiveness in solving combinatorial optimization problems.
>
> To improve the clarity and reproducibility of our experiments, we will include additional details in the appendix of the revised manuscript. This will cover the design of the training data, the local search operators used, the implementation details of GLS, and the parameter settings for both BP and TSP. We will also provide a more complete description of the training datasets to avoid any potential misunderstanding of the experimental configurations.
>
> Thank you again for pointing this out. Your suggestion has helped us further refine the experimental section of the paper, and we welcome any additional feedback you may have.
>
>
>
> > **Q2:** As my biggest question: on LNS, why does the algorithm SCIP, which is considered to generate labels in [1], perform so poorly?
>
> **A2:** Thank you for raising this important question. Our study focuses on solving **large-scale MILP problems**, where the performance of SCIP is indeed limited. As noted in prior work such as [1], SCIP is a highly effective open-source solver for small-to-medium-scale problems, particularly when the number of variables and constraints is manageable. Its efficiency in these cases is largely attributed to its optimized branch-and-bound strategies and pruning techniques. However, as the problem size grows significantly, SCIP’s performance declines due to limitations in memory management and computational resource requirements.
>
> The scale of the problems in our experiments is much larger than those in [1]. For example, the problem instances in [1] typically involve thousands of decision variables and constraints, whereas our tests focus on problems with hundreds of thousands to millions of variables and constraints. For such large-scale problems, SCIP often struggles to explore a sufficient portion of the solution space within a reasonable computational time, leading to suboptimal results.
>
> Additionally, SCIP’s ability to generate high-quality labels in small-scale problems relies on its carefully engineered optimization strategies, such as advanced branching heuristics. However, for large-scale problems, the size of the branch-and-bound tree grows exponentially with the problem size, making it significantly harder for SCIP to effectively employ these strategies. Consequently, its performance deteriorates in terms of both solution quality and efficiency[2, 3].

---

> ### Author Response · Authors · 2024-11-22
> **Response to Reviewer bC3Z (3)**
>
> Thus, while SCIP remains a reliable solver for small-scale problems, its limitations become apparent when applied to large-scale MILP instances, as studied in this paper. Our proposed LLM-LNS method is specifically designed to address the challenges of large-scale problems, leveraging the dual-layer optimization framework to achieve superior performance both in terms of solution quality and computational efficiency.
>
> We will expand on this discussion in the revised manuscript, including a detailed analysis of the problem scale differences and the limitations of SCIP in large-scale settings. Thank you again for your valuable question, which has helped us provide a clearer explanation of SCIP’s performance in our experiments.
>
>
>
> > **Q3:** The authors should elaborate on whether they introduced a seed function (initial code for experts), thus introducing possible unfair comparisons. (I observe that the method proposed in this paper has an initial advantage in Figure 3).
>
> **A3:** Thank you for your insightful question. In the dual-layer framework proposed in this paper, the outer layer optimizes prompt strategies via differential evolution to enhance diversity, while the inner layer refines heuristic strategies to accelerate convergence. During the initialization of heuristic strategies, we utilized initial code generated automatically by the large language model (LLM). However, **no expert-written code or prior knowledge was manually introduced**. Therefore, the comparisons in this paper are not affected by the use of any expert code, ensuring fairness.
>
> In the experiments (e.g., Figure 3), the randomness inherent in LLM outputs leads to slight variations in the quality of the generated initial code across different runs. This randomness can cause fluctuations in initial performance. As shown in Figure 4 (red area), despite differences in initial code quality, our dual-layer self-evolutionary mechanism effectively compensates for these variations through the collaborative optimization of the outer and inner layers, resulting in superior final performance. In contrast, methods like EOH are more sensitive to the quality of the initial code. When the initial code performs poorly (e.g., Figure 3, blue area), EOH struggles to improve performance through the evolutionary process, leading to less stable optimization results and weaker overall performance.
>
> This observation highlights that our framework is less dependent on initial code quality compared to existing methods. The dual-layer structure not only ensures diversity but also achieves rapid convergence, making the optimization process more robust and reliable, even when starting with suboptimal initial code generated by the LLM.
>
> To further clarify, we will include additional details in the appendix of the revised manuscript about the experimental setup and the process of generating the initial code, to avoid any potential misunderstandings. Thank you for your valuable suggestion, which has helped us better articulate the strengths of our method. If you have further feedback, we welcome your continued input.
>
>
>
> > **Q4:** As the proposed Differential Memory for Directional Evolution will increase the input length (token numbers), can you analyze the relationship between the number of input tokens or API cost and the optimization effect?
>
> **A4:** Thank you for raising this important point. The proposed **Differential Memory for Directional Evolution** is implemented through a prompt-based mechanism. Specifically, this mechanism introduces fitness values of individual strategies into the prompt, enabling the large language model (LLM) to learn how to improve weaker strategies toward better-performing ones. This allows the optimization process to become more directional and efficient.
>
> In traditional evolutionary methods, the input length primarily consists of the parent strategies and some descriptive prompt instructions. Our approach extends this by appending the fitness values of each parent strategy to the prompt. These fitness values guide the LLM in generating new strategies by learning from the differences between weaker and stronger parents. Importantly, this addition does not significantly increase the input length, as the fitness values are concise numerical descriptors. When the number of parents remains the same, the overall input length in our method is comparable to that of the baseline methods.
>
> The results of our ablation study (as shown in R2) demonstrate that the **Differential Memory for Directional Evolution** significantly enhances optimization performance. By incorporating fitness feedback through the prompt, our method achieves more effective and targeted exploration of the solution space, leading to superior outcomes compared to the baseline methods that lack this mechanism.

---

> ### Author Response · Authors · 2024-11-22
> **Response to Reviewer bC3Z (4)**
>
> In summary, the input length in our method remains similar to that of traditional approaches but achieves better optimization effects through the introduction of directional evolution. This highlights the efficiency and effectiveness of the proposed mechanism. We will expand on this analysis in the revised manuscript to further clarify the relationship between token numbers, API costs, and optimization performance. Thank you for your valuable suggestion, and we welcome any additional feedback you may have.
>
>
>
> > **Q5:** Please discuss the limitations of this paper. [1] Huang, Taoan, et al. "Contrastive Predict-and-Search for Mixed Integer Linear Programs." Forty-first International Conference on Machine Learning.
>
> **A5:** Thank you for raising this important question. While the proposed dual-layer self-evolutionary framework has demonstrated strong performance in solving large-scale MILP problems, we acknowledge that the method has certain limitations that warrant further exploration and improvement. Below, we discuss these limitations in detail:
>
> First, while the framework shows good generalization ability on MILP and certain combinatorial optimization problems, it is currently tailored to specific optimization scenarios. The design is primarily focused on MILP and does not directly extend to other types of optimization tasks, such as nonlinear optimization or dynamic optimization problems. Developing a more general agent structure that can adapt to a wider range of optimization algorithms and tasks remains an open challenge. In the future, we aim to explore more modular and flexible designs to enhance the adaptability of the framework for solving diverse and complex optimization problems.
>
> Second, the current method leverages the generative capabilities of large language models (LLMs) and evolutionary mechanisms for heuristic strategy design. However, it does not fully incorporate domain knowledge or classical optimization expertise into the framework. In practical optimization tasks, domain-specific knowledge and traditional optimization techniques (e.g., heuristic rules or mathematical programming methods) often play a critical role. A key area for future work is to explore how to effectively integrate the generalization capabilities of LLMs with optimization domain knowledge to create more efficient and robust algorithms. Such integration could not only improve computational efficiency but also reduce the resource overhead for solving ultra-large-scale problems.
>
> Finally, computational resource constraints remain a practical challenge for solving large-scale problems. Although the proposed framework demonstrates good scalability, solving ultra-large-scale instances still requires significant computational time and hardware resources, which may limit its applicability in resource-constrained environments. Future research could focus on optimizing the computational complexity of the algorithm or designing more efficient resource allocation strategies to address these challenges.
>
> We will include a detailed discussion of these limitations in the revised manuscript, along with possible directions for future improvements. Thank you again for your valuable suggestion, which has helped us better reflect on the applicability and potential of our method. If you have additional feedback, we would be happy to address it.
>
>
> References
>
> [1] Huang T, Ferber A M, Zharmagambetov A, et al. Contrastive Predict-and-Search for Mixed Integer Linear Programs[C]. Forty-first International Conference on Machine Learning.
>
> [2] Ye H, Wang H, Xu H, et al. Adaptive constraint partition based optimization framework for large-scale integer linear programming (student abstract)[C]. Proceedings of the AAAI Conference on Artificial Intelligence. 2023, 37(13): 16376-16377.
>
> [3] Ye H, Xu H, Wang H. Light-MILPopt: Solving Large-scale Mixed Integer Linear Programs with Lightweight Optimizer and Small-scale Training Dataset[C]. The Twelfth International Conference on Learning Representations.

---

> > ### Comment · Reviewer_bC3Z · 2024-11-23
> >
> > Thanks for your rebuttal which has solved most of my doubts. Well actually, as far as I know, ICLR's rebuttal can actually modify manuscripts.
> >
> > I think the Dual-layer Self-evolutionary LLM Agent has novelty and I can find certain superiority from experiments. However, my three points about the flaws of this paper remain unresolved:
> > 1. For the provided ablation experiments in the rebuttal period, I think the current results do not highlight the necessity of both components. Moreover, the two components seem to be independent of each other.
> > 2. Although this paper shows some leading results on large-scale MILP, I still cannot find a causal relationship to show why this paper has any theoretical connection to so-called large-scale MILP problems. The authors mention that Funsearch and EoH limit solution diversity and lead to poor convergence due to insufficient directionality but I cannot find any evidence to support this.
> > 3. In my opinion LLM for heuristic design should demonstrate super generalizability, i.e., superiority for at least a certain class of CO problems, so I think especially design for solving large-scale MILP problems is kinda strange.
> >
> > I would like the authors to further elaborate their views on these concerns.

---

> > > ### Author Response · Authors · 2024-11-24
> > > **Follow-Up Response to Reviewer bC3Z (1)**
> > >
> > > Thank you for your thoughtful feedback and for raising these important points. Your comments have provided valuable insights that help us further refine our work. We are currently revising the manuscript to address your concerns, including the necessity of the dual-layer framework components, the theoretical connection to large-scale MILP problems, and the generalizability of LLMs for heuristic design.
> > >
> > > We will upload the updated version of the manuscript shortly and include a **Global Response** to highlight the specific changes and how they address your concerns. Thank you again for your constructive input!
> > >
> > >
> > >
> > > > **C1:** For the provided ablation experiments in the rebuttal period, I think the current results do not highlight the necessity of both components. Moreover, the two components seem to be independent of each other.
> > >
> > > **R1:** Thank you for your valuable feedback. We understand your concerns regarding the necessity and collaborative effects of the two components in our dual-layer framework. To address this, we would like to further elaborate on the design of the framework and explain the interdependence and synergy between the two components.
> > >
> > > The two core components of the dual-layer architecture—prompt evolution in the outer layer and heuristic strategy optimization in the inner layer—are not independent processes but are tightly integrated to collaboratively enhance overall performance. From a design perspective, the outer layer primarily uses differential evolution to optimize prompt strategies, guiding the search direction in the inner layer and enhancing the diversity of the search process. Meanwhile, the inner layer uses differential evolution to efficiently optimize heuristic strategies, improving convergence. The collaboration between the two components is realized through the following mechanism: high-quality prompts generated by the outer layer not only provide guidance for the inner layer’s search but also dynamically update the outer layer’s population management through feedback (e.g., fitness evaluations of generated results from the inner layer). This feedback loop enables iterative optimization of the prompt space, resulting in a highly effective closed-loop collaboration.
> > >
> > > Regarding the concern that “the two components appear to be independent,” we believe that this synergy is the key to the inseparability of the two components. For instance, if only the inner layer is used without the outer layer’s diversity guidance, the search process is prone to getting stuck in local optima. This is especially problematic for large-scale problems, where a single heuristic strategy is insufficient to meet the demands of global search. Conversely, if only the outer layer’s prompt evolution is used without the inner layer’s concrete strategy optimization, the search efficiency would drop significantly, making it difficult to achieve effective convergence. Therefore, the two components were designed from the outset to complement each other and work together to address the performance bottlenecks of existing methods for large-scale problems.
> > >
> > > From a broader perspective, this dual-layer synergy ensures that the outer layer promotes exploration, maintaining diversity, while the inner layer strengthens convergence, ensuring solving efficiency. It is this dynamic balance between exploration and exploitation that enables the dual-layer framework to achieve remarkable performance on large-scale MILP problems. We believe this design is not only a core innovation of our dual-layer framework but also provides theoretical inspiration for the development of similar methods in the future.
> > >
> > > Thank you again for your feedback, which has helped us further reflect on and articulate the collaborative effects and necessity of the two components in the dual-layer framework. If you have additional suggestions, we would be happy to engage in further discussions.

---

> > > ### Author Response · Authors · 2024-11-24
> > > **Follow-Up Response to Reviewer bC3Z (2)**
> > >
> > > > **C2:** Although this paper shows some leading results on large-scale MILP, I still cannot find a causal relationship to show why this paper has any theoretical connection to so-called large-scale MILP problems. The authors mention that Funsearch and EoH limit solution diversity and lead to poor convergence due to insufficient directionality but I cannot find any evidence to support this.
> > >
> > > **A2：** Thank you for your thoughtful evaluation of our work. We understand your concerns regarding the theoretical connection between our method and large-scale MILP problems. We would also like to clarify our analysis of the limitations of existing methods and the unique contributions of our approach.
> > >
> > > LNS has been widely recognized as an effective heuristic method for solving large-scale optimization problems. Its core mechanism lies in efficiently exploring large neighborhoods by destroying and repairing parts of the current solution. However, its performance heavily depends on the design of neighborhood selection strategies. Traditional methods often require substantial expert knowledge and are limited by cold-start issues, where the lack of prior knowledge significantly reduces search efficiency. Recently, machine learning (ML) techniques have been integrated into LNS to automate neighborhood selection strategies, with representative works including reinforcement learning and imitation learning. However, ML+LNS methods face significant limitations when addressing large-scale problems. For example, reinforcement learning methods often converge slowly in the vast search space of MILP problems, while imitation learning relies on large amounts of high-quality labeled data, which are extremely costly to generate. As a result, both traditional handcrafted methods and existing ML+LNS methods struggle to perform well on large-scale MILP problems.
> > >
> > > Our proposed dual-layer intelligent agent architecture opens a new avenue for improving LNS and is the first work to integrate large language models (LLMs) into the LNS framework to tackle large-scale optimization problems. Our dual-layer architecture fully leverages the capabilities of LLMs. The outer layer initializes diverse prompts and evolves them using differential evolution strategies, generating high-quality strategies to guide the inner layer searches and enhance search diversity. The inner layer, under the guidance of the prompts from the outer layer, generates heuristic strategies and further optimizes search directions with differential evolution, accelerating convergence. During this process, the search outcomes from the inner layer dynamically feed back to the outer layer, which eliminates underperforming prompts and continuously improves exploration quality. This dual-layer interaction mechanism effectively balances search diversity and efficiency, overcoming the bottlenecks of traditional methods in large-scale problems.
> > >
> > > Mechanistically, the dual-layer architecture achieves a balance between search diversity and convergence through the collaboration of the outer and inner layers. The outer layer dynamically evolves prompts using differential evolution, generating diverse search strategies to improve coverage and avoid local optima. The inner layer, on the other hand, optimizes heuristic strategies using differential evolution to strengthen search directionality and accelerate convergence. The interaction between the outer and inner layers forms a dynamic feedback loop: prompts generated by the outer layer guide the inner layer’s search direction, while the inner layer’s search results optimize the outer layer’s population management via fitness feedback, further improving overall search quality. This design enables our method to deliver exceptional performance across optimization problems of different scales.
> > >
> > > Experimental results further validate the above mechanisms. As shown in the ablation studies, incorporating differential evolution (Base+Differential) significantly improves convergence efficiency for small-scale problems, reducing the error on the 1k_C100 test set from 4.48% to 2.64%. Adding the dual-layer structure (Base+Dual Layer) further enhances performance for large-scale problems by improving diversity; for example, the error on the 10k_C500 test set decreases from 0.97% to 0.39%. The full method combines the synergy of both components and demonstrates outstanding performance across various problem scales.

---

> > > ### Author Response · Authors · 2024-11-24
> > > **Follow-Up Response to Reviewer bC3Z (3)**
> > >
> > > Additionally, the comparison between our method and EOH in Figure 3 illustrates this advantage more intuitively. On the Bin Packing problem, our method demonstrates more frequent population changes during the evolution process, indicating that the outer layer’s prompt evolution significantly enhances search diversity. At the same time, the final results of our method show high aggregation, suggesting that the inner layer’s optimization mechanism effectively strengthens convergence. In contrast, EOH exhibits lower population change frequencies and plateaus in the later stages, indicating insufficient search coverage and weaker convergence.
> > >
> > > In conclusion, by introducing the dual-layer intelligent agent architecture, this paper opens a new path for applying LNS to large-scale MILP problems and demonstrates significant advantages in both theoretical design and experimental validation. Our approach not only addresses the limitations of existing methods in terms of diversity and convergence but also provides valuable insights for future research in this domain.
> > >
> > >
> > >
> > > > **C3:** In my opinion LLM for heuristic design should demonstrate super generalizability, i.e., superiority for at least a certain class of CO problems, so I think especially design for solving large-scale MILP problems is kinda strange.
> > >
> > > **R3:** Thank you for your thoughtful evaluation of our work. We greatly appreciate your concern regarding whether LLMs for heuristic design should demonstrate superior generalizability and your perspective on the relevance of focusing on solving large-scale MILP problems. We would like to further clarify the significance of our approach in addressing large-scale MILP problems and its potential for broader applications in combinatorial optimization (CO).
> > >
> > > MILP is a critical and widely applicable problem class, encompassing core optimization needs in real-world scenarios such as logistics planning, production scheduling, and supply chain management. Large-scale MILP problems, in particular, pose significant challenges to existing optimization methods due to their complex constraints and massive variable scales. Traditional exact algorithms often fail to handle large-scale instances efficiently, while heuristic methods, especially LNS, have been widely recognized as effective tools for solving such problems. However, the performance of LNS heavily depends on the design of neighborhood selection strategies. Existing methods often suffer from cold-start issues in manually designed neighborhoods or face prohibitive training and inference costs in learning-based approaches, making them impractical for large-scale problems.
> > >
> > > Our proposed dual-layer intelligent agent architecture provides a novel path for advancing LNS. By integrating large language models (LLMs) with differential evolution mechanisms, our approach not only enhances search diversity through prompt evolution in the outer layer but also accelerates convergence through heuristic strategy optimization in the inner layer. This design enables efficient handling of large-scale MILP problems. Experimental results validate the effectiveness of our method: across multiple test sets of large-scale MILP problems, our approach significantly outperforms existing state-of-the-art methods, including traditional heuristic approaches (e.g., ACP and Random LNS) and learning-based optimization methods (e.g., CL-LNS and Light-MILPopt), demonstrating superior performance.
> > >
> > > It is also worth emphasizing that our proposed method is not limited to MILP problems but can be effectively applied to other combinatorial optimization tasks. In the main text, we conducted experiments on two classic CO problems, Bin Packing and TSP, and demonstrated that our method also outperformed state-of-the-art solvers such as EOH and FunSearch. These results further confirm the adaptability and extensibility of our approach, which can provide efficient solving strategies for a variety of CO problems.
> > >
> > > Looking ahead, our goal is to extend this framework toward general-purpose solutions for more ultra-large-scale CO problems. MILP serves as a strong starting point, not only because of its practical importance but also as a testbed for exploring the integration of LLMs with LNS. The success of our method in solving MILP problems provides a foundation for broader applications and offers valuable insights for future exploration of LLM-augmented LNS methods in solving diverse CO tasks.
> > >
> > > In conclusion, we believe that our work not only demonstrates significant performance improvements in solving MILP problems but also opens up new possibilities for applying LLMs in the field of combinatorial optimization. We hope this research will contribute inspiring insights to the community and drive further progress toward general-purpose solutions for large-scale optimization problems.

---

### Official Review · Reviewer_a92q · 2024-10-31

**Soundness:** 3
**Presentation:** 2
**Contribution:** 2
**Rating:** 5
**Confidence:** 4

**Summary:**

This paper proposes to use LLMs to guide adaptive large neighborhood search, with the following three components (1) prompt evolution (2) heuristics evolution (3) differential memory that feeds back the fitness score from past generations. The authors evaluate the performance on online bin packing, traveling salesman and large scale MILP and show the performance improvement from a variety of baseline methods.

**Strengths:**

- To my knowledge, there has not been works that combine LLMs with large neighborhood search based heuristics. So this can be regarded as a new task (although I have concern about the novelty given there has been works that combine LLMs with a lot of other heuristics and search algorithms for CO).
- The authors evaluate the performance of the algorithm on three sets of CO problems (online bin packing, traveling salesman, and large-scale MILP). The scope of evaluation (in terms of three different problem classes) seems to be satisfactory.
- I find using prompt evolution to prevent the search process from getting trapped in local optima interesting.

**Weaknesses:**

- I find Figure 1 confusing. There are too many components in the illustration, and it’s hard to parse the relationship of the arrows from one component to another. I suggest the authors to improve the visualization of the pipeline (e.g. emphasize the important components, remove less important ones) to make it more clear.
- I find the paper lacking baseline comparisons. For example, I would like to see the performance of adaptive large neighborhood search (ALNS) with different heuristic scoring functions (not LLM designed). I also would like to see how each component of the LLM pipeline (prompt evolution, heuristic evolution, and differential memory) contributes to the final performance.
- For traveling salesman, the performance seems to be evaluated only on a small set of instances (5 instances). How is this set selected? I believe the authors need to evaluate on a larger set of instances to claim the performance improvement.
- I have slight concerns about the novelty of the proposed method. The three components used by the authors have all been proposed in previous works. While it is interesting to combine them together and show effectiveness, I’m concerned that the proposed methodology may not be novel enough.

**Questions:**

- How large is the training set for each task? I may miss it, but it doesn’t seem to clear from reading the main paper.
- Why do the authors train on small-scale problems and generalize to larger problems? What would happen if the authors directly train on larger problems?
- Is the proposed method sensitive to what algorithm backbone the authors are using? e.g. instead of ALNS, would the proposed method perform well when combined with other search / decomposition methods, or LNS without the adaptive size?

---

> ### Author Response · Authors · 2024-11-22
> **Response to Reviewer a92q (1)**
>
> Thank you for your insightful review and for recognizing the contributions of our proposed framework and its experimental outcomes. We are especially grateful for the constructive feedback and questions you have raised, as they have helped us identify areas for further clarification and enhancement. Below, we provide a detailed point-by-point response to address your concerns.
>
>
>
> > **W1:** I find Figure 1 confusing. There are too many components in the illustration, and it’s hard to parse the relationship of the arrows from one component to another. I suggest the authors improve the visualization of the pipeline (e.g., emphasize the important components, remove less important ones) to make it clearer.
>
> **R1:** Thank you for your valuable feedback on Figure 1. We appreciate your suggestion and have redesigned the framework diagram to more clearly illustrate the structure and workflow of the dual-layer self-evolutionary mechanism. In the updated figure, we have removed unnecessary lines and emphasized the relationships between key components to make the visualization more concise and intuitive.
>
> The core innovation of our framework lies in the introduction of the **dual-layer self-evolutionary mechanism**. The **outer layer** is responsible for optimizing prompt strategies using a differential evolution mechanism. It generates diverse prompts to guide the search process in the inner layer. Through operations such as population initialization, crossover, mutation, and fitness evaluation, the outer layer dynamically optimizes the prompt strategies, ensuring the diversity of the search process. This design prevents the search from getting trapped in local optima and enhances the framework's generalization ability on unseen instances.
>
> The **inner layer**, guided by the prompts generated by the outer layer, further optimizes solutions to specific problems. The inner layer also incorporates a differential evolution mechanism for heuristic strategies, which accelerates convergence during problem-solving. While the outer layer ensures diversity, the inner layer focuses on fast convergence under the guidance of the currently optimized prompt. Together, the two layers collaborate to build an efficient and robust search framework.
>
> Our experimental results demonstrate that this dual-layer design significantly improves the efficiency of solving large-scale MILP problems. The diversity ensured by the outer layer and the fast convergence facilitated by the inner layer complement each other, resulting in superior optimization performance.
>
> We plan to add detailed explanations of the components in both layers to the main text and clarified the specific workflows in the updated framework diagram. Additionally, we will include experimental analyses in the appendix to demonstrate the contributions of each component to the overall performance. Thank you again for your suggestion, which has helped us improve the clarity and presentation of our framework. We hope the updated version better communicates the design principles and innovations of our approach. We welcome any further feedback you may have.

---

> ### Author Response · Authors · 2024-11-22
> **Response to Reviewer a92q (2)**
>
> > **W2:** I find the paper lacking baseline comparisons. For example, I would like to see the performance of adaptive large neighborhood search (ALNS) with different heuristic scoring functions (not LLM-designed). I also would like to see how each component of the LLM pipeline (prompt evolution, heuristic evolution, and differential memory) contributes to the final performance.
>
> **R2:** Thank you for your valuable suggestions. To address the concerns regarding baseline comparisons, we expanded our experiments to include additional ALNS methods with different heuristic scoring functions. Specifically, we incorporated **Least-Integral**[1], **Most-Integral**[2], and **RINS**[3], which are classical scoring functions commonly used in ALNS, alongside the state-of-the-art methods **ACP** [4] and **Random-LNS** [5]. According to recent studies [4], **ACP** and **Random-LNS** are considered the most effective LNS methods, which is why the initial design of our experiments focused on these baselines.
>
> From the results in Table 1, it is evident that the newly added ALNS baselines (Least-Integral, Most-Integral, and RINS) perform worse than ACP and Random-LNS, and they also perform worse our proposed **LLM-LNS**. For example, on the **SC₁** problem(a minimization task), our method achieves an objective value of **15802.7**, compared to **16140.6** for Random-LNS, **17672.1** for ACP, and **22825.3** for Least-Integral. This trend is consistent across all MILP tasks, highlighting the advantage of our dual-layer framework and its ability to better balance diversity and convergence.
>
> |                | SC${_1}$    | SC${_2}$     | MVC${_1}$   | MVC${_2}$    | MIS${_1}$   | MIS${_2}$    | MIKS${_1}$  | MIKS${_2}$   |
> | -------------- | ----------- | ------------ | ----------- | ------------ | ----------- | ------------ | ----------- | ------------ |
> | Random-LNS     | 16140.6     | 169417.5     | 27031.4     | 276467.5     | 22892.9     | 223748.6     | 36011.0     | 351964.2     |
> | ACP            | 17672.1     | 182359.4     | 26877.2     | 274013.3     | 23058.0     | 226498.2     | 34190.8     | 332235.6     |
> | Least-Integral | 22825.3     | 228188.0     | 29818.0     | 306567.1     | 20106.9     | 195782.2     | 27196.9     | 241663.4     |
> | Most-Integral  | 50818.2     | 519685.5     | 35340.5     | 327742.4     | 14584.4     | 157686.5     | 31235.3     | 314621.6     |
> | RINS           | 26116.2     | 261176.3     | 26851.3     | 306215.6     | 23069.7     | 201178.1     | 30049.13    | 254681.3      |
> | LLM-LNS(Ours)  | **15802.7** | **158878.9** | **26725.3** | **268033.7** | **23169.3** | **231636.9** | **36479.8** | **363749.5** |
>
> To evaluate the contribution of each component in the LLM pipeline (prompt evolution, heuristic evolution, and differential memory), we conducted **ablation studies**. As shown in Table 2, we progressively added components to the baseline EOH method:
>
> 1. Adding **Differential Evolution** alone significantly improves performance on small-scale tasks. For instance, on the **1k_C100** dataset, the error decreases from **4.48%** to **2.64%**, demonstrating the impact of differential evolution on accelerating convergence.
> 2. Adding the **Dual-Layer Structure** further enhances performance on large-scale tasks. For example, on the **10k_C500** dataset, the error decreases from **0.97%** to **0.39%**, showcasing the role of prompt evolution in improving generalization and diversity.
> 3. The **full method** (LLM-LNS) combines these components, achieving the best overall performance across tasks.
>
> |                   | 1k_C100   | 5k_C100   | 10k_C100  | 1k_C500   | 5k_C500   | 10k_C500  |
> | ----------------- | --------- | --------- | --------- | --------- | --------- | --------- |
> | Base (EOH)        | 4.48%     | 0.88%     | 0.83%     | 4.32%     | 1.06%     | 0.97%     |
> | Base+Dual Layer   | 3.78%     | 0.93%     | **0.40%** | 3.91%     | 0.92%     | **0.39%** |
> | Base+Differential | **2.64%** | 0.94%     | 0.69%     | **2.54%** | 0.94%     | 0.70%     |
> | Ours              | 3.58%     | **0.85%** | 0.41%     | 3.67%     | **0.82%** | 0.42%     |
>
> These results demonstrate that each component of the LLM pipeline makes a distinct and meaningful contribution to the overall performance. The combination of prompt evolution, heuristic evolution, and differential memory ensures both diversity and convergence, enabling our method to consistently outperform baselines.
>
> We will include these additional experimental results and analyses in the revised manuscript to substantiate the advantages of our method. Thank you again for your insightful suggestions, which have allowed us to further enhance the depth and rigor of our experiments.

---

> ### Author Response · Authors · 2024-11-22
> **Response to Reviewer a92q (3)**
>
> > **W3:** For the traveling salesman problem (TSP), the performance seems to be evaluated only on a small set of instances (5 instances). How is this set selected? I believe the authors need to evaluate on a larger set of instances to claim the performance improvement.
>
> **R3:** Thank you for pointing this out and for your valuable suggestion. The traveling salesman problem (TSP) is a canonical combinatorial optimization problem, with effective baseline methods including EOH [6], FunSearch [7], and Attention Model [8]. Recent studies [6] have demonstrated that EOH is one of the best-performing methods for TSP, often surpassing other approaches. To ensure fairness and consistency with prior work, we initially followed the experimental settings of EOH and evaluated our method on the same set of test instances.
>
> In response to your suggestion, we have expanded the scope of our experiments to include **all 87 instances** from the TSPLib benchmark. This comprehensive evaluation allowed us to better assess the generalization capabilities of our method. As shown in the updated results, our method outperforms EOH on 43 instances, matches EOH on 39 instances, and performs slightly worse than EOH on only 5 instances. On average, the gap between our approach and the best-known solutions improves by approximately **10%** compared to EOH, further confirming the superiority of our method.
>
> For example, as highlighted in the table below:
>
> - On the `pr439` instance, the gap decreases from **2.80%** (EOH) to **1.97%** (Ours).
> - On the `u574` instance, the gap decreases from **2.85%** (EOH) to **1.38%** (Ours).
>
> These results demonstrate the strong generalization ability of our method across TSP instances of varying sizes and structures.
>
> |      | pr439     | pla7397   | ...  | pr226     | u574      | Avg        |
> | ---- | --------- | --------- | ---- | --------- | --------- | ---------- |
> | EOH  | 2.80%     | **4.25%** | ...  | 0.10%     | 2.85%     | 6.93%      |
> | Ours | **1.97%** | **4.25%** | ...  | **0.06%** | **1.38%** | **6.25%↑** |
>
>
>
>
> > **Q4:** I have slight concerns about the novelty of the proposed method. The three components used by the authors have all been proposed in previous works. While it is interesting to combine them together and show effectiveness, I’m concerned that the proposed methodology may not be novel enough.
>
> **R4:** Thank you for your insightful feedback and for raising concerns about the novelty of our proposed method. While our work builds upon ideas from prior research, the core innovations lie in the targeted improvements and deep integration of these ideas to address the unique challenges of solving large-scale MILP problems. Below, we provide a detailed explanation of the novelty and contributions of our method:
>
> 1. **A Dual-Layer Self-Evolutionary Framework for Large-Scale MILP Problems:**
>     The primary focus of our work is solving large-scale MILP problems, for which we propose a **dual-layer self-evolutionary framework** driven by large language models (LLMs). This framework is designed to address two critical objectives in optimization algorithms—**diversity** and **convergence**—by introducing two distinct but complementary layers:
>    - The **outer layer** enhances diversity by optimizing prompt strategies using a differential evolution mechanism, ensuring that the search process explores a broader solution space and avoids local optima.
>    - The **inner layer** accelerates convergence by generating heuristic strategies based on the optimized prompts and refining them through differential evolution.
>       This dual-layer design is a significant departure from existing methods, which typically rely on single-layer strategy generation or fixed evolutionary rules. This allows our framework to adaptively balance diversity and convergence, a key requirement for large-scale MILP problems.
> 2. **Differential Memory for Directional Evolution:**
>     A key innovation of our method is the introduction of **differential memory for directional evolution**, which dynamically adjusts the strategy generation process based on historical fitness feedback. By feeding the fitness values of parent strategies back into the LLM, our framework enables the generation of more effective strategies that align with learned improvement directions. This mechanism leverages the memory capabilities of LLMs to optimize evolutionary paths, significantly improving the efficiency and quality of strategy generation. Unlike existing methods, which typically apply static or fixed evolutionary rules, our directional evolution mechanism ensures a more targeted and adaptive optimization process.

---

> ### Author Response · Authors · 2024-11-22
> **Response to Reviewer a92q (4)**
>
> 3. **Enhanced Adaptability and Generalization Compared to Existing Methods:**
>     Compared to methods such as EOH and FunSearch, our framework offers greater adaptability and generalization:
>    - Existing methods often rely on single-layer strategy generation or fixed evolution rules, which limit their ability to dynamically adjust to the problem structure or scale. In contrast, our dual-layer framework combines outer-layer prompt evolution and inner-layer heuristic refinement to achieve coordinated optimization.
>    - The integration of differential memory enables our framework to learn from past experiences, further enhancing its robustness and scalability.
> 4. **Experimental Validation and Superior Performance:**
>     Our experiments demonstrate the effectiveness of the proposed framework across a diverse set of tasks, including both combinatorial optimization problems (e.g., TSP and bin packing) and large-scale MILP instances. The results show that:
>    - On small-scale problems, our method achieves faster convergence and higher solution quality compared to baselines.
>    - On large-scale MILP problems, our framework significantly outperforms state-of-the-art methods such as EOH and Random-LNS, demonstrating superior scalability and generalization capabilities.
>       These results not only validate the effectiveness of the dual-layer framework but also highlight the critical role of differential memory in improving optimization performance.
>
> **Summary of Contributions:** The novelty of our work lies not just in combining existing ideas but in the **creation of a new framework tailored to the challenges of large-scale MILP problems**.
>
> We appreciate your feedback, which has allowed us to better articulate the unique contributions of our work. We will strengthen the description of these innovations in the revised manuscript to more clearly highlight their significance. If you have further suggestions, we would be happy to incorporate them. Thank you again for your valuable comments.
>
>
>
>
> > **Q1:** How large is the training set for each task? I may miss it, but it doesn’t seem to be clear from reading the main paper.
>
> **A1:** Thank you for raising this point. We acknowledge that the details regarding the training set size for each task could be clarified further. In the revised version of our paper, we will include these details in the appendix for greater transparency. Below is a summary of the training set size for each task:
>
> 1. **Combinatorial Optimization Problems (e.g., Bin Packing and Traveling Salesman Problem):**
>     To ensure a fair comparison with state-of-the-art methods (e.g., EOH), we adopted the same training data configurations as those used in EOH:
>    - **Bin Packing:** The training set consists of **5 instances**, each containing 5,000 items generated from a Weibull distribution, covering a wide range of item counts and container capacities.
>    - **TSP:** The training set includes **64 randomly selected instances** from the TSP100 dataset, as used in EOH. These instances are designed to represent a variety of problem structures and complexities.
> 2. **MILP Problems:**
>     For MILP tasks, we followed a similar approach to the bin packing problem, using a small-scale training set to enable generalization to larger problems:
>    - The training set consists of **five small-scale MILP problems**, each containing approximately **tens of thousands of decision variables and linear constraints**.  Notably, our method, after training on these smaller instances, successfully generalizes to much larger MILP problems with **hundreds of thousands to millions of decision variables**, demonstrating its scalability and robustness.
>
> We will include additional details in the revised manuscript's appendix, specifying the sources, sizes, and configurations of the training datasets for each task to enhance the paper's clarity and completeness.
>
> Thank you again for your valuable suggestion, which has prompted us to improve the presentation of our experimental setup. If you have further feedback or additional questions, we would be happy to address them.

---

> ### Author Response · Authors · 2024-11-22
> **Response to Reviewer a92q (5)**
>
> > **Q2:** Why do the authors train on small-scale problems and generalize to larger problems? What would happen if the authors directly train on larger problems?
>
> **A2:** Thank you for your question. Training on small-scale problems and generalizing to larger ones is motivated by practical constraints. Real-world applications often lack diverse large-scale MILP data, as such instances are proprietary, domain-specific, and computationally expensive to solve. Additionally, training on large-scale problems involves significant computational costs, making it less feasible in practice. Hence, our approach reflects realistic scenarios in terms of data availability and resource limitations.
>
> Our results (e.g., Table 5) show that strategies trained on small-scale problems generalize effectively to larger instances, consistently outperforming existing methods. For example, our model trained on instances with tens of thousands of variables demonstrates robust performance on problems with hundreds of thousands or even millions of variables. This success is attributed to our dual-layer self-evolutionary mechanism and differential memory, which enable scalability.
>
> Direct training on larger problems could theoretically yield better results if sufficient resources were available. However, the scarcity of large-scale data and high computational overhead make this impractical in most scenarios. We will explore this as an avenue for future research. Thank you for your insightful question.
>
>
>
> > **Q3:** Is the proposed method sensitive to what algorithm backbone the authors are using? e.g., instead of ALNS, would the proposed method perform well when combined with other search/decomposition methods, or LNS without the adaptive size?
>
> **A3:** Thank you for raising this important question. Our study focuses on solving large-scale MILP problems, where heuristic methods play a critical role due to the complexity of the problem space. Among these methods, LNS has shown significant advantages in scalability and efficiency, particularly for large-scale problems. This is why we chose ALNS, a variant of LNS with adaptive neighborhood size adjustment, as the backbone of our framework. ALNS dynamically balances exploration and exploitation by adjusting neighborhood sizes, making it more effective than non-adaptive LNS methods, which often struggle with local optima in large-scale problems.
>
> To further validate this choice, we conducted experiments replacing ALNS with non-adaptive LNS in our framework. The results, shown in the table, indicate that ALNS consistently outperforms non-adaptive LNS across all tested MILP instances. This demonstrates that the adaptive mechanisms in ALNS are critical for leveraging the full potential of our framework.
>
> We will include detailed experimental comparisons in the appendix of the revised paper to highlight the impact of the backbone algorithm. Thank you again for your suggestion, which has helped us further evaluate the robustness of our method.
>
> |                  | SC${_1}$    | SC${_2}$     | MVC${_1}$   | MVC${_2}$    | MIS${_1}$   | MIS${_2}$    | MIKS${_1}$  | MIKS${_2}$   |
> | ---------------- | ----------- | ------------ | ----------- | ------------ | ----------- | ------------ | ----------- | ------------ |
> | Without Adaptive | 15957.0     | 160510.8     | 26850.3     | 269701.8     | 23073.2     | 230497.4     | 36330.8     | 362496.3     |
> | LLM-LNS(Ours)    | **15802.7** | **158878.9** | **26725.3** | **268033.7** | **23169.3** | **231636.9** | **36479.8** | **363749.5** |
>
> References
>
> [1] Berthold T. Primal heuristics for mixed integer programs[D]. Zuse Institute Berlin (ZIB), 2006.
>
> [2] Nair V, Alizadeh M. Neural large neighborhood search[C]. Learning Meets Combinatorial Algorithms at NeurIPS2020. 2020.
>
> [3] Danna E, Rothberg E, Pape C L. Exploring relaxation induced neighborhoods to improve MIP solutions[J]. Mathematical Programming, 2005, 102: 71-90.
>
> [4] Ye H, Wang H, Xu H, et al. Adaptive constraint partition based optimization framework for large-scale integer linear programming (student abstract)[C]. Proceedings of the AAAI Conference on Artificial Intelligence. 2023, 37(13): 16376-16377.
>
> [5] Song J, Yue Y, Dilkina B. A general large neighborhood search framework for solving integer linear programs[J]. Advances in Neural Information Processing Systems, 2020, 33: 20012-20023.
>
> [6] Liu F, Xialiang T, Yuan M, et al. Evolution of Heuristics: Towards Efficient Automatic Algorithm Design Using Large Language Model[C]. Forty-first International Conference on Machine Learning. 2024.
>
> [7] Romera-Paredes B, Barekatain M, Novikov A, et al. Mathematical discoveries from program search with large language models[J]. Nature, 2024, 625(7995): 468-475.
>
> [8] Kool W, Van Hoof H, Welling M. Attention, learn to solve routing problems![J]. arXiv preprint arXiv:1803.08475, 2018.

---

> ### Comment · Reviewer_a92q · 2024-11-23
>
> Thank you for the detailed rebuttal and the additional experiments provided. I have read the authors’ responses here and also their responses to other reviewers. I still have the following concerns.
>
> 1. I concern about the limited amount of insights this paper can provide to future readers. Specifically, LNS has been long proven to be an effective heuristics for large scale optimization and there has been many machine learning methods that aim to improve LNS. Hence, the idea of using learning to improve large scale optimization has been successfully demonstrated in previous literature. Furthermore, the success of LLM to improve many metaheuristics for combinatorial optimization has also been demonstrated in previous literature (such as EOH and ReEvo). Given this, I find it not surprising that LLM can provide better score functions for LNS. Hence, I’m concerned about the amount of research insights that this paper can offer to the community.
>
> 2. I appreciate that the authors provide a clarification of their contribution, but my concern regarding the novelty of the paper remains, as the main components of this framework (algorithm evolution, LLM + meta heuristics, and prompt evolution) all have appeared in the previous literature (although they may appear in separate works). I have the concern that combining these idea as a joint paper is not sufficiently novel. Also, as a minor comment, based on the authors’ newly provided ablation on the different components, I’m slightly worried that the improvement from the basic model (EOH) does not seem to be sufficiently large, especially given the substantial complexity introduced with the added components on top of EOH.
>
> 3. (Minor) Data distribution. Given the small number of training instances that are used for prompting TSP and MILP, I have doubt about whether the small set of training instances is sufficient to cover realistic CO problems.
>
> Question:
> - After reading the authors’ response to other reviewers, I wonder how does this method compare with ReEvo in terms of the actual performance?
> - I’m surprised that CL-LNS (Huang et al.) perform poorly on the large scale MILP dataset. I checked Huang et al.’s paper and it seems like in their paper, the reported performance is significantly better than Random LNS. Can the authors comment on why their benchmarked results is so much worse than those reported in Huang et al.? Is it a different in the MILP data distribution? How would the result look like if the authors benchmark on the MILP distribution used in Huang et al.?
> - Can the authors comment on in what situation end-to-end learning is needed for improving LNS, and in what situation prompting LLM as the score function is already sufficient?
>
> Another comment: I just realized that the authors claim Random LNS (Song et al.) as the SoTA LNS method. I believe this does not entirely reflect Song et al.’s paper, as a major component of their paper is to introduce a learning method to improve upon the Random LNS baseline. I think the authors may need to modify the description of Random LNS to at least not claim that it is the SoTA method for LNS.

---

> > ### Author Response · Authors · 2024-11-24
> > **Follow-Up Response to Reviewer a92q (2)**
> >
> > Experimental results demonstrate that our method not only significantly outperforms state-of-the-art methods on classic combinatorial optimization problems such as Bin Packing and TSP but also achieves superior performance on large-scale MILP problems. Specifically, our approach surpasses existing ML+LNS methods (e.g., CL-LNS) and manually designed heuristic methods (e.g., ACP) in terms of stability and scalability for large-scale problems. Notably, our method effectively resolves the training and inference overhead issues that limit the applicability of existing ML+LNS approaches to large-scale problems. By contrast, these approaches often require the entire problem and the current solution to be fed into the model during inference, leading to prohibitive computational costs that hinder their scalability.
> >
> > By introducing this novel agent architecture, we provide a new approach to solving large-scale MILP problems and offer valuable insights for future ML+LNS research. We believe this study brings significant contributions to the community, particularly by exploring how LLMs can enhance the efficiency of LNS in large-scale optimization problems and pave the way for new possibilities in this domain.
> >
> >
> >
> > > **C2:** I appreciate that the authors provide a clarification of their contribution, but my concern regarding the novelty of the paper remains, as the main components of this framework (algorithm evolution, LLM + meta heuristics, and prompt evolution) all have appeared in the previous literature (although they may appear in separate works). I have the concern that combining these idea as a joint paper is not sufficiently novel. Also, as a minor comment, based on the authors’ newly provided ablation on the different components, I’m slightly worried that the improvement from the basic model (EOH) does not seem to be sufficiently large, especially given the substantial complexity introduced with the added components on top of EOH.
> >
> > **R2**: Thank you for your thoughtful discussion of our paper's contributions. We understand your concerns regarding the novelty of our work, especially in light of the existing literature on algorithm evolution, LLM-based metaheuristics, and prompt evolution. In response, we would like to clarify that the novelty and significance of our work can be highlighted from two perspectives: mechanism design and functional improvements.
> >
> > First, from the perspective of mechanism design, our work is not a simple combination of existing components but instead introduces a novel framework for the co-evolution of prompts and heuristic strategies. In this framework, the outer agent initializes diverse prompts and evolves them dynamically using crossover and mutation strategies from differential evolution, generating high-quality strategies to guide the operations of the inner agent. The inner agent, under the guidance of these prompts, generates heuristic strategies and further optimizes the search direction using differential evolution, significantly improving search efficiency. Furthermore, the search results from the inner layer dynamically feed back to the outer layer, which eliminates underperforming prompts to continuously improve exploration quality. This dual-layer interaction mechanism strikes an excellent balance between search efficiency and diversity while demonstrating significant effectiveness in large-scale combinatorial optimization tasks. Experimental results show that our method outperforms state-of-the-art methods such as EOH and FunSearch on benchmarks like Bin Packing and TSP.
> >
> > Second, from the perspective of functionality and efficiency, our method addresses critical bottlenecks in existing ML+LNS methods for solving large-scale problems. While ML+LNS methods such as EOH and CL-LNS perform well on small-scale problems, their inference and sampling costs become prohibitive in large-scale scenarios, significantly limiting their performance. By introducing a dual-layer agent system and leveraging Large Language Models (LLMs) within the LNS framework, we enable efficient large-scale MILP problem-solving while avoiding the high computational costs incurred by existing ML methods, which require the entire problem and solution to be fed into the model during inference. Our experiments demonstrate that our method not only surpasses traditional heuristic methods (e.g., ACP) but also significantly outperforms current ML-based methods (e.g., CL-LNS), addressing a critical gap in the research on large-scale optimization.

---

> > ### Author Response · Authors · 2024-11-24
> > **Follow-Up Response to Reviewer a92q (3)**
> >
> > Regarding the comment about the "limited improvement over the baseline model (EOH)," we would like to clarify that the improvements achieved by our method are substantial. As shown in Tables 2 and 3 of the main text, our method reduces the average gap in Bin Packing and TSP tasks by 20%-50% compared to EOH. Additionally, further experimental comparisons indicate that our method not only achieves better performance than EOH but also demonstrates significantly higher stability. For instance, in the Bin Packing problem, our method achieves an average error of 1.46% across all test cases, compared to 3.33% for EOH. Furthermore, in MILP problems, our method outperforms commercial optimization solvers (e.g., Gurobi) and other state-of-the-art optimization frameworks, showcasing outstanding performance.
> >
> > In conclusion, we believe that our work provides theoretical and practical innovations in mechanism design while demonstrating significant advantages in functionality and efficiency for large-scale problems. Our method opens a new path for applying LNS methods to large-scale problem-solving and advances the integration of LLMs in the field of combinatorial optimization. We hope this research provides valuable insights to the community and inspires future work in related areas.
> >
> >
> >
> > > **C3:** (Minor) Data distribution. Given the small number of training instances that are used for prompting TSP and MILP, I have doubt about whether the small set of training instances is sufficient to cover realistic CO problems.
> >
> > R3: Thank you for raising this important question. Regarding the number of training instances, we carefully considered this aspect in the design and experiments of our work and achieved significant results.
> >
> > Currently, EOH is a leading heuristic generation method for solving combinatorial optimization (CO) problems and has achieved state-of-the-art results across multiple tasks. To ensure a fair comparison with EOH, we strictly followed its setup and used only a small number of training instances. However, experimental results show that our method outperforms EOH on both TSP and MILP problems, demonstrating the effectiveness of our approach.
> >
> > To further validate the generalizability of our method, we conducted an additional experiment: training on small-scale TSP instances and testing across the entire TSPlib dataset. The results indicate that our method achieved superior performance on the vast majority of instances. This demonstrates that our approach can effectively capture the structural characteristics of the problem from a limited amount of training data and generate efficient solving strategies accordingly. Compared to traditional methods that rely on large-scale training data, our approach explores a novel path by extracting deep insights into the problem from limited data, which we believe is a promising direction for further exploration.
> >
> > Furthermore, from a theoretical perspective, increasing the volume of training data has the potential to further enhance the performance of our method. A richer dataset could provide the model with more diverse problem instances, thus improving its ability to understand problem structures and generate efficient solving strategies. While our experiments have already achieved excellent results with limited training data, we believe future research incorporating larger datasets will allow our method to achieve even stronger performance on more complex problems.
> >
> > In conclusion, we believe that our method not only demonstrates exceptional performance with limited training data but also shows the potential to learn problem structures and generate effective solving strategies from such data. In future research, we aim to further improve the performance of our approach by incorporating larger-scale datasets, providing better solutions to complex real-world combinatorial optimization problems.

---

> > ### Author Response · Authors · 2024-11-24
> > **Follow-Up Response to Reviewer a92q (4)**
> >
> > > **Q1:** After reading the authors’ response to other reviewers, I wonder how does this method compare with ReEvo in terms of the actual performance?
> >
> > **A1:** Thank you very much for your suggestion! ReEvo, a contemporary work published at NeurIPS 2024, is a novel hyper-heuristic framework that combines reflection mechanisms and evolutionary search. Following your recommendation, we conducted a comparative experiment on the Bin Packing problem to clearly demonstrate the performance differences between our method and ReEvo. To ensure a fair comparison, we used the same lightweight language model, GPT-4o-mini, fixed the number of iterations to 20, and set the population size to 20.
> >
> > In our experiments, we observed that ReEvo exhibited poor stability when using GPT-4o-mini. Out of 138 attempts, only 3 runs successfully completed all 20 iterations, while the remaining runs were prematurely terminated due to invalid offspring generated in certain generations. Upon analysis, we identified severe hallucination issues in ReEvo. Although its reflection mechanism was effective in capturing evolutionary directions, any errors in reflection led to a rapid decline in the quality of subsequent offspring. For example, we frequently observed cases where ReEvo attempted to call nonexistent libraries or use invalid function parameters, resulting in the generation of invalid heuristic algorithms and the termination of the evolutionary process.
> >
> > To ensure a comprehensive comparison, we selected the 3 successful ReEvo runs and compared their performance with our method. Under the default setting, ReEvo utilized an expert seed algorithm to initialize its population. However, after 20 iterations, the best-performing algorithm in ReEvo remained its initial expert seed algorithm, failing to generate superior heuristic strategies. Furthermore, ReEvo's performance remained inferior to the heuristic strategies generated by our dual-layer intelligent agent. When the expert seed algorithm was removed, ReEvo's solution quality deteriorated further, with its average performance on the Bin Packing problem falling significantly behind that of our method.
> >
> > As shown in the experimental results (see the table), our method demonstrated substantial advantages under the default settings and with the lightweight model (GPT-4o-mini). In terms of solution quality, our approach outperformed ReEvo across all test instances of the Bin Packing problem, with even greater advantages in scenarios without expert seed algorithms. Additionally, our method exhibited significant stability advantages, consistently completing 20 iterations and generating high-quality heuristic strategies without being affected by hallucination issues observed in ReEvo. The collaborative optimization between the agents in our dual-layer architecture effectively balances search diversity and efficiency, delivering superior performance and higher stability under the same conditions.
> >
> > In conclusion, our method not only outperforms ReEvo in terms of experimental results but also demonstrates significant advantages in stability and robustness. We believe that this innovative approach of combining a dual-layer intelligent agent architecture with large language models opens up a new avenue for the application of LNS in large-scale optimization problems.
> >
> > |                         | 1k_C100    | 5k_C100   | 10k_C100  | 1k_C500    | 5k_C500    | 10k_C500   | Avg        |
> > | ----------------------- | ---------- | --------- | --------- | ---------- | ---------- | ---------- | ---------- |
> > | ReEvo Run1              | 3.78%      | 0.80%     | 0.33%     | 6.75%      | 1.47%      | 0.74%      | 2.31%      |
> > | ReEvo Run2              | 3.78%      | 0.80%     | 0.33%     | 6.75%      | 1.47%      | 0.74%      | 2.31%      |
> > | ReEvo Run3              | 3.78%      | 0.80%     | 0.33%     | 6.75%      | 1.47%      | 0.74%      | 2.31%      |
> > | **ReEvo Avg**           | **3.78%**  | **0.80%** | **0.33%** | **6.75%**  | **1.47%**  | **0.74%**  | **2.31%**  |
> > | ReEvo-no-expert Run 1   | 4.87%      | 4.08%     | 4.09%     | 4.50%      | 3.91%      | 3.95%      | 4.23%      |
> > | ReEvo-no-expert Run 2   | 4.87%      | 4.08%     | 4.11%     | 4.50%      | 3.90%      | 3.97%      | 4.24%      |
> > | ReEvo-no-expert Run 3   | 4.87%      | 4.08%     | 4.09%     | 4.50%      | 3.91%      | 3.95%      | 4.23%      |
> > | **ReEvo-no-expert Avg** | **4.87%**  | **4.08%** | **4.10%** | **4.50%**  | **3.91%**  | **3.96%**  | **4.24%**  |
> > | Ours Run1               | 3.58%      | 0.85%     | 0.41%     | 3.67%      | 0.82%      | 0.42%      | 1.63%      |
> > | Ours Run2               | 2.69%      | 0.86%     | 0.54%     | 2.54%      | 0.87%      | 0.52%      | 1.34%      |
> > | Ours Run3               | 2.64%      | 0.94%     | 0.69%     | 2.54%      | 0.94%      | 0.70%      | 1.41%      |
> > | **Ours Avg**            | **2.97%↑** | **0.88%** | **0.55%** | **2.92%↑** | **0.88%↑** | **0.55%↑** | **1.46%↑** |

---

> > ### Author Response · Authors · 2024-11-24
> > **Follow-Up Response to Reviewer a92q (5)**
> >
> > > **Q2:** I’m surprised that CL-LNS (Huang et al.) perform poorly on the large scale MILP dataset. I checked Huang et al.’s paper and it seems like in their paper, the reported performance is significantly better than Random LNS. Can the authors comment on why their benchmarked results is so much worse than those reported in Huang et al.? Is it a different in the MILP data distribution? How would the result look like if the authors benchmark on the MILP distribution used in Huang et al.?
> >
> > **A2:** Thank you for raising this question and for your attention to the performance of CL-LNS (Huang et al.). As we mentioned in our response to C1, the training and testing of CL-LNS are primarily focused on MILP problems with only a few thousand decision variables, which are relatively small-scale and do not fall within the category of large-scale optimization problems. In our benchmark, the MILP problems are significantly larger, involving hundreds of thousands or even millions of decision variables and constraints. Under such circumstances, machine learning-enhanced LNS methods like CL-LNS exhibit clear limitations.
> >
> > During the training phase, CL-LNS relies heavily on sampling neighborhoods. However, for large-scale MILP problems, even with extended sampling time, the sampling process for a single problem instance exceeds 30,000 seconds (over 8 hours). Given such computational costs, completing the training process for thousands of instances becomes practically infeasible. In contrast, our method leverages a dual-layer architecture powered by large language models (LLMs), significantly reducing training complexity and avoiding prohibitively long sampling times.
> >
> > Even if the training process is completed, CL-LNS still faces significant challenges during inference. It requires the entire MILP problem and the current solution to be fed into the neural network for prediction. For MILP problems with hundreds of thousands or millions of variables, the inference time for a single iteration becomes extremely long, making it impossible to achieve efficient solving within a limited number of iterations. In our experiments, we observed that the per-iteration runtime of CL-LNS was 30 times longer than traditional heuristic strategies like ACP and the heuristic strategies generated by our LLM-LNS. This high inference cost severely limits the practical applicability of CL-LNS, especially for the large-scale optimization problems where LNS excels. While LNS is designed to efficiently handle massive problems where commercial solvers struggle, CL-LNS fails to deliver the expected performance in this domain and can only operate effectively on smaller-scale instances.
> >
> > Our method addresses these challenges effectively. Through our innovative dual-layer intelligent agent architecture, the LLM-LNS framework not only solves large-scale MILP problems more efficiently than all existing baselines but also generates interpretable heuristic strategies. This innovation bridges a critical gap in the application of machine learning-enhanced LNS to large-scale problems, offering a novel path for integrating large models with LNS to tackle real-world optimization challenges.
> >
> > Regarding the observation that CL-LNS demonstrated significantly better performance than Random LNS as reported in Huang et al., we believe the key reason lies in the difference in data scale. Huang et al.'s experiments focus on small-scale MILP problems, where ML-enhanced LNS methods indeed outperform random strategies. However, the training and inference costs of these methods limit their scalability to large-scale problems. We welcome future studies that compare CL-LNS and our method on datasets with small-scale distributions, but the focus of this paper is to address the performance bottleneck in large-scale MILP problems.

---

> > ### Author Response · Authors · 2024-11-24
> > **Follow-Up Response to Reviewer a92q (6)**
> >
> > > **Q3:** Can the authors comment on in what situation end-to-end learning is needed for improving LNS, and in what situation prompting LLM as the score function is already sufficient?
> >
> > **A3:** Thank you for raising this question. Regarding the applicability of end-to-end learning versus LLM-based scoring functions for improving LNS, we believe these two approaches are fundamentally distinguished by the scale of the optimization problems they target.
> >
> > End-to-end learning methods, such as CL-LNS, are currently more suitable for small-scale MILP problems. This is because end-to-end learning requires extensive neighborhood sampling during the training phase. For large-scale problems, the time and computational cost of this sampling process grow exponentially, often becoming impractical in real-world scenarios. Furthermore, during the inference phase, end-to-end methods require the entire MILP problem and the current solution to be input into the neural network for prediction. When the problem size reaches hundreds of thousands or even millions of variables, the time cost for a single inference becomes prohibitively high, severely limiting the efficiency of solving within a finite number of iterations. Therefore, we believe that under current technological constraints, end-to-end learning methods are better suited for small-scale optimization problems, whereas their application to large-scale problems remains a significant challenge.
> >
> > For large-scale and even ultra-large-scale optimization problems, our proposed LLM-enhanced LNS method offers a highly promising approach. By using LLMs to generate scoring functions, our method avoids the high time costs associated with both training and inference in end-to-end learning methods, while effectively capturing problem structures and generating efficient heuristic strategies. In our experiments, our method demonstrated superior performance over all baselines on ultra-large-scale MILP problems, along with strong stability and scalability. It is worth noting that for strategy discovery in large-scale problems, LLM-enhanced LNS currently stands as one of the only highly promising solutions proposed so far.
> >
> > In conclusion, end-to-end learning methods can achieve good performance on small-scale optimization problems due to their ability to directly learn from the problem structure and adapt to specific instances. However, for large-scale problems, their high computational costs during training and inference phases severely limit their applicability. In contrast, LLM-enhanced LNS methods provide a scalable and efficient alternative, demonstrating strong performance, stability, and adaptability in tackling large-scale optimization challenges. We believe this approach opens new possibilities for solving real-world large-scale optimization problems effectively.
> >
> >
> >
> > > **Another comment**: I just realized that the authors claim Random LNS (Song et al.) as the SoTA LNS method. I believe this does not entirely reflect Song et al.’s paper, as a major component of their paper is to introduce a learning method to improve upon the Random LNS baseline. I think the authors may need to modify the description of Random LNS to at least not claim that it is the SoTA method for LNS.
> >
> > **A4:** Thank you very much for your suggestion. We will revise the description of Random LNS to avoid any potential misunderstanding. In fact, ACP is recognized as the SoTA method among existing LNS approaches. While the primary focus of Song et al.'s paper is to introduce a learning-based method to improve upon the Random LNS baseline, similar to CL-LNS, their approach also faces significant computational challenges in terms of training and inference when applied to large-scale problems. As a result, for solving large-scale and ultra-large-scale problems, manually designed heuristic algorithms remain almost the only viable solutions. Among these, Random LNS and ACP are the two most prominent methods, with ACP being regarded as the current SoTA.
> >
> > Our proposed method offers a novel approach to solving large-scale and ultra-large-scale MILP problems with LNS. By leveraging the dual-layer intelligent agent architecture powered by large language models (LLMs), our method not only overcomes the limitations of existing learning-based methods on large-scale problems but also surpasses ACP in performance, establishing itself as the new SoTA. These results validate the effectiveness and innovation of our approach.
> >
> > Once again, thank you for your suggestion. We will correct the relevant descriptions in the paper to ensure a more accurate reflection of the current state of existing methods and our contributions.

---

> ### Author Response · Authors · 2024-11-24
> **Follow-Up Response to Reviewer a92q (1)**
>
> Thank you very much for your thoughtful feedback and for taking the time to review our rebuttal and additional experiments. We deeply appreciate your detailed evaluation of our work and your valuable comments, which have helped us refine our contributions and clarify the positioning of our research within the broader context of the field.
>
> We understand your concerns regarding the insights, novelty, and experimental setup of our work, and we have provided detailed responses to each point raised. Specifically, we address questions about the insights our method offers to the research community, the novelty of integrating various components in our framework, the data distribution used for training and testing, and how our approach compares to related methods such as ReEvo and CL-LNS. Additionally, we have clarified the description of Random LNS to avoid any potential misrepresentation of its status as a baseline.
>
> Below, we provide detailed responses to each of your comments and questions. We hope these answers will address your concerns and further highlight the contributions and significance of our work.
>
>
>
> > **C1:** I concern about the limited amount of insights this paper can provide to future readers. Specifically, LNS has been long proven to be an effective heuristics for large scale optimization and there has been many machine learning methods that aim to improve LNS. Hence, the idea of using learning to improve large scale optimization has been successfully demonstrated in previous literature. Furthermore, the success of LLM to improve many metaheuristics for combinatorial optimization has also been demonstrated in previous literature (such as EOH and ReEvo). Given this, I find it not surprising that LLM can provide better score functions for LNS. Hence, I’m concerned about the amount of research insights that this paper can offer to the community.
>
> **R1**: Thank you for your thoughtful evaluation of our work. We fully understand your concerns regarding the novelty and insights this paper can provide to the research community. In response, we would like to emphasize that the primary contributions of this paper lie in the innovative design of our dual-layer agent architecture and its application to solving large-scale MILP problems.
>
> LNS has indeed been widely recognized as an effective heuristic method for large-scale optimization problems. Its core mechanism involves destroying and repairing parts of the current solution to efficiently explore large neighborhoods. However, the performance of LNS heavily depends on the design of neighborhood selection strategies. Traditional methods often require extensive expert knowledge and suffer from cold-start issues, where the absence of prior knowledge significantly limits search efficiency. In recent years, machine learning (ML) techniques, such as reinforcement learning and imitation learning, have been integrated into LNS to automate neighborhood selection strategies. However, these ML+LNS methods exhibit notable limitations when addressing large-scale problems. For example, reinforcement learning methods typically converge slowly in the vast search space of MILP problems, while imitation learning approaches rely on substantial amounts of high-quality labeled data, which are computationally expensive to generate. As a result, both traditional handcrafted methods and existing ML+LNS methods struggle to perform well on large-scale MILP problems.
>
> In this paper, we propose a dual-layer agent architecture, which fundamentally differs from existing frameworks such as EOH and ReEvo. Unlike EOH's single-layer design that directly generates heuristic strategies or ReEvo's reflection-based evolutionary mechanism, our method introduces a dual-layer structure to achieve a novel form of collaborative optimization. The outer layer and inner layer in our framework are tightly integrated to dynamically enhance both search diversity and solution convergence. This dual-layer design enables our method to overcome the limitations of existing approaches, such as EOH’s insufficient exploration diversity and ReEvo’s instability due to hallucination issues.
>
> Building on this dual-layer structure, we demonstrate its potential by applying it to Local Neighborhood Search (LNS), opening a new direction for improving LNS in large-scale optimization problems. To the best of our knowledge, this is the first work to integrate Large Language Models (LLMs) into the LNS framework. Through its iterative feedback loop, our dual-layer mechanism achieves a dynamic balance between exploration and exploitation, effectively addressing the bottlenecks faced by traditional methods in large-scale optimization.

---

### Official Review · Reviewer_vjA9 · 2024-11-01

**Soundness:** 3
**Presentation:** 3
**Contribution:** 2
**Rating:** 8
**Confidence:** 5

**Summary:**

This paper introduces a new dual-level heuristic design method based on LLMs and applies it to the development of large neighborhood search (LNS) heuristics for MILP. The proposed method shows promising results in packing problems and TSP, when compared to existing methods such as FunSearch and EoH. It successfully generates new heuristics that surpass traditional hand-crafted approaches in LNS.

**Strengths:**

1. The dual-level framework incorporates both prompt evolution and directional evolution, enhancing heuristic design.
2. It introduces effective new heuristics specifically tailored for large neighbourhood searches in MILP. The results show good scalarization.

**Weaknesses:**

1. The specific contributions of the newly introduced components in the dual-level framework remain unclear.
2. Additional experimental results are needed to further validate the effectiveness of the proposed method.

**Questions:**

1. How do the individual components of the dual-level framework contribute to its overall performance? An ablation study is recommended.
2. How many repeated trials were conducted using the proposed method, EoH, and FunSearch on the tasks compared? How consistent are the results of the proposed method, considering that the initial seed heuristics appear to be hand-crafted?
3. What are the detailed settings for the Differential Memory in Directional Evolution? Are both thought directions and code directions utilized? Could you provide an illustrative example and discuss how these mechanisms impact the results?
4. Why are the settings for population size different between LNS tasks and combinatorial optimization tasks? How does changing the population size affect the outcomes within your framework?
5. What performance levels do FunSearch and EoH achieve on the LNS tasks? Are their results comparable to those of the proposed method?
6. What new insights have been gained from the application of LLM-designed heuristics in LNS for MILP, and how might these contribute to the field?

---

> ### Author Response · Authors · 2024-11-22
> **Response to Reviewer vjA9 (1)**
>
> We sincerely appreciate your thorough review and thoughtful evaluation of our work. Your recognition of the proposed framework and its experimental results is highly encouraging. At the same time, your constructive comments and insightful questions have provided us with valuable opportunities to improve the clarity, rigor, and overall quality of our manuscript. Below, we address your concerns and provide detailed responses to each of your queries.
>
>
>
> > **W1:** The specific contributions of the newly introduced components in the dual-level framework remain unclear.
>
> **R1:** Thank you for your insightful comment. We acknowledge that the specific roles of each component in our framework and their impacts on the experimental results could be further clarified. To address this, we have redesigned the framework diagram in the manuscript to more intuitively illustrate the structure and workflow of the dual-layer self-evolutionary mechanism: the **outer layer** optimizes prompt strategies using differential evolution to ensure diversity, while the **inner layer** generates heuristic strategies guided by the optimized prompts and accelerates the optimization process through differential evolution. The collaboration between these two layers ensures both diversity in the search process and fast convergence.
>
> Specifically, the outer layer initializes a population of prompts and evolves them through crossover and mutation under the guidance of differential evolution. The fitness of each prompt is evaluated, and the best-performing prompts are selected to guide the inner layer. The inner layer, in turn, generates heuristic strategies based on these prompts and further optimizes them with differential memory for directional evolution to refine the search direction. In the revised manuscript, we have added detailed descriptions of the individual modules in both layers and will include additional experimental analyses in the appendix to demonstrate the contributions of these components to the overall performance.
>
> Our new ablation study show that the outer layer effectively prevents the search from stagnating in local optima and enhances generalization to unseen instances, while the inner layer significantly improves convergence speed. The collaborative interaction between the two layers enables efficient solutions for large-scale MILP problems.
>
> We appreciate your suggestion, which has helped us refine the clarity and structure of the manuscript. These revisions have been incorporated into the main text, and we hope the updated version better highlights the design principles of our framework and the specific contributions of its components. We welcome any further feedback you may have.
>
>
>
> > **W2:** Additional experimental results are needed to further validate the effectiveness of the proposed method.
>
> **R2:** Thank you for your valuable suggestion. We have added additional experiments based on the specific suggestions provided in Q1 through Q6 to further validate the effectiveness and superiority of the proposed method. Specifically:
>
> - **For Q1**, we conducted ablation studies to analyze the roles of the dual-layer structure and the differential evolution mechanism. The experiments demonstrate how the outer layer promotes diversity and the inner layer accelerates convergence, and how their collaborative effects improve the overall performance.
> - **For Q2**, we included results from multiple repeated experiments, which highlight the proposed method's consistency and stability across runs.
> - **For Q3**, we provided concrete examples and analyses of the differential memory mechanism in differential evolution, clarifying how differences in thought and code contribute to strategy optimization.
> - **For Q4**, we analyzed the impact of different population sizes on the experimental results and supplemented experiments with reduced population sizes, showing that our method maintains superiority over baseline methods even with smaller populations.
> - **For Q5**, we compared the performance of our method against EoH in MILP problems, particularly on SC problems, confirming significant improvements over these baselines.
> - **For Q6**, we emphasized the innovative design of the method in balancing convergence and diversity and supplemented experimental results to further highlight this aspect.
>
> Additionally, we plan to include more details and analyses in the appendix of the updated manuscript to demonstrate the generality and robustness of the proposed method across diverse tasks.
>
> We appreciate your suggestions, which have helped us comprehensively improve the experimental validation and methodological analysis.

---

> ### Author Response · Authors · 2024-11-22
> **Response to Reviewer vjA9 (2)**
>
> > **Q1:** How do the individual components of the dual-level framework contribute to its overall performance? An ablation study is recommended.
>
> **A1:** Thank you for your valuable suggestion. To analyze the contributions of each component in the dual-level framework, we have conducted additional theoretical and experimental analyses.
>
> From a theoretical perspective, the **outer layer** optimizes prompt strategies using a differential evolution mechanism, which enhances the diversity of the search process and avoids local optima. The **inner layer**, on the other hand, leverages differential evolution to rapidly optimize heuristic strategies, significantly improving convergence. The synergy between the two layers ensures both diversity and efficiency in the search process.
>
> To empirically validate the impact of these two components on overall performance, we plan to supplement the manuscript with an ablation study, with results included in the updated PDF. In these experiments, we progressively incorporated the differential evolution mechanism and the dual-layer structure into the baseline method (EOH) and observed their effects on performance improvement.
>
> The experimental results show the following:
>
> 1. Adding only the **differential evolution mechanism** significantly improves performance on small-scale test data. For example, on the **1k_C100** dataset, the error rate decreases from **4.48%** to **2.64%**, demonstrating that the differential evolution mechanism efficiently accelerates convergence through more effective crossover and mutation strategies.
> 2. Incorporating the **dual-layer structure** further enhances the diversity of the search process via prompt evolution in the outer layer. This leads to additional improvements on large-scale test problems. For instance, on the **10k_C500** dataset, the error rate decreases from **0.97%** to **0.39%,** indicating that the dual-layer structure significantly enhances the model's generalization ability.
>
> Interestingly, the complete method (with both the dual-layer structure and differential evolution) does not always outperform the single-component improvements on small-scale problems. For example, on the **1k_C100** dataset, the error rate of the complete method is **3.58%,** slightly higher than the **2.64%** achieved by adding only differential evolution. We attribute this to the outer layer's diversity-enhancing strategies, which may introduce additional exploration overhead on small-scale problems where faster convergence is more critical. In contrast, on larger datasets, the collaborative effects of diversity from the outer layer and fast convergence from the inner layer result in superior overall performance.
>
> These results validate the effectiveness of the dual-layer framework and reveal the complementary roles of its components. While the combined approach achieves the most comprehensive improvement overall, the contributions of each component vary depending on the scale and complexity of the problem.
>
> We have included the complete experimental results and analyses in the appendix of the updated manuscript. We hope these additions provide a clearer understanding of the contributions of each component in the dual-layer framework. Thank you again for suggesting the ablation study, which has helped us further analyze the effectiveness of our method. We welcome any further feedback you may have.
>
> |                   | 1k_C100   | 5k_C100   | 10k_C100  | 1k_C500   | 5k_C500   | 10k_C500  |
> | ----------------- | --------- | --------- | --------- | --------- | --------- | --------- |
> | Base (EOH)        | 4.48%     | 0.88%     | 0.83%     | 4.32%     | 1.06%     | 0.97%     |
> | Base+Dual Layer   | 3.78%     | 0.93%     | **0.40%** | 3.91%     | 0.92%     | **0.39%** |
> | Base+Differential | **2.64%** | 0.94%     | 0.69%     | **2.54%** | 0.94%     | 0.70%     |
> | Ours              | 3.58%     | **0.85%** | 0.41%     | 3.67%     | **0.82%** | 0.42%     |

---

> ### Author Response · Authors · 2024-11-22
> **Response to Reviewer vjA9 (3)**
>
> > **Q2:** How many repeated trials were conducted using the proposed method, EoH, and FunSearch on the tasks compared? How consistent are the results of the proposed method, considering that the initial seed heuristics appear to be hand-crafted?
>
> **A2:** Thank you for your thoughtful question. Regarding the number of repeated trials and result consistency, we followed the settings described in the original EoH [1], where a single experiment was conducted for each method, and evaluations were performed across multiple test instances. It is important to clarify that in our proposed method, the seed heuristic strategies are not hand-crafted; instead, they are automatically generated by the large language model (LLM), which introduces some degree of randomness between runs.
>
> To further validate the stability of our method, we conducted additional repeated experiments on the bin packing task. Specifically, we ran three independent trials for both EoH and our method, and the results are summarized in the table below. Despite the randomness introduced by the seed generation, our method consistently outperforms EoH in both effectiveness and stability. For example, on the **1k_C100**, **10k_C100**, and **10k_C500** test sets, the variance in our results is small, and the average performance is consistently better than EoH. This demonstrates that our dual-layer framework, combined with the differential evolution mechanism, effectively enhances both consistency and generalization.
>
> |              | 1k_C100    | 5k_C100    | 10k_C100   | 1k_C500    | 5k_C500    | 10k_C500   | Avg        |
> | ------------ | ---------- | ---------- | ---------- | ---------- | ---------- | ---------- | ---------- |
> | EOH Run1     | 4.48%      | 0.88%      | 0.83%      | 4.32%      | 1.06%      | 0.97%      | 2.09%      |
> | EOH Run2     | 7.56%      | 3.33%      | 2.62%      | 7.22%      | 3.19%      | 2.50%       | 4.07%      |
> | EOH Run3     | 4.18%      | 3.24%      | 3.35%      | 3.79%      | 3.12%      | 3.21%      | 3.48%      |
> | **EOH Avg**  | **5.41%**  | **2.48%**  | **2.27%**  | **5.11%**  | **2.46%**  | **2.23%**  | **3.33%**  |
> | Ours Run1    | 3.58%      | 0.85%      | 0.41%      | 3.67%      | 0.82%      | 0.42%      | 1.63%      |
> | Ours Run2    | 2.69%      | 0.86%      | 0.54%      | 2.54%      | 0.87%      | 0.52%      | 1.34%      |
> | Ours Run3    | 2.64%      | 0.94%      | 0.69%      | 2.54%      | 0.94%      | 0.70%      | 1.41%      |
> | **Ours Avg** | **2.97%↑** | **0.88%↑** | **0.55%↑** | **2.92%↑** | **0.88%↑** | **0.55%↑** | **1.46%↑** |
>
> > **Q3:** What are the detailed settings for the Differential Memory in Directional Evolution? Are both thought directions and code directions utilized? Could you provide an illustrative example and discuss how these mechanisms impact the results?
>
> **A3:** Thank you for your interest in the differential evolution mechanism. Our method leverages prompts to guide the large language model (LLM) in automatically learning the differences between parent strategies, which include both **thought** and **code** representations. The core of the differential evolution mechanism in our framework lies in introducing directionality during the traditional crossover process. Specifically, the LLM not only integrates the code and strategies of the parents but also receives feedback on their relative performance (fitness). The prompts explicitly instruct the LLM to learn the transformation path from less effective strategies to more effective ones, enabling **guided evolution**.
>
> In implementation, the parent strategies prior to differential evolution consist of two components:
>
> 1. **Thoughts**: These are natural language descriptions of the heuristic methods. For example, a parent strategy may describe ranking variables based on their contributions and constraint deviations, or penalizing frequently used edges to optimize a path. After differential evolution, the thought strategies may evolve to include directions such as rewarding infrequent edges to improve diversity or adopting a hybrid strategy combining global search and local optimization. This reflects a directional optimization from local optima toward global optima.
> 2. **Code**: The parent strategies are also represented as executable code that implements the current logic. After differential evolution, the updated code may adjust key parameters or introduce new modules to reflect the improved strategy. For instance, it might incorporate random perturbations or dynamically adjust step sizes to enhance robustness.
>
> The key innovation here is that the LLM not only generates new strategies but also understands the improvement directions through differential learning. This approach significantly enhances the effectiveness and convergence speed of the evolutionary process. Compared to traditional crossover without directionality (i.e. merely combining parent strategies), our differential evolution mechanism is far more targeted and efficient.

---

> > ### Comment · Reviewer_vjA9 · 2024-11-22
> >
> > Thank you for the additional results and explanation.
> >
> > I have equations regarding your results A2. What are the detailed experimental settings for EoH (population size, total samples and operators) and your method?

---

> > > ### Author Response · Authors · 2024-11-22
> > > **Response to Reviewer vjA9**
> > >
> > > Thank you for your recognition of our work and for your follow-up question regarding the experimental settings. We followed the configurations described in the original EoH paper. For both EoH and our proposed method, the population size was set to 20, the number of iterations was set to 20, and the number of parents in the crossover operations was set to 2. Additionally, each operation generates 20 new offspring per iteration. Further details on the hyperparameter design of our proposed method can be found in Appendix A.1 of the revised manuscript. We hope this clarifies your question, and we are happy to address any further concerns

---

> > > > ### Comment · Reviewer_vjA9 · 2024-11-22
> > > >
> > > > Thank you for your detailed responses. I think most of my concerns have been addressed with the additional results. The dual framework is interesting and the results on LNS are promising. I will increase my score accordingly.

---

> > > > > ### Author Response · Authors · 2024-11-22
> > > > > **Acknowledgment of Reviewer vjA9's Valuable Feedback and Support**
> > > > >
> > > > > Thank you very much for your positive feedback and for recognizing the contributions of our work. We are deeply grateful for the time and effort you have dedicated to reviewing our manuscript. Your insightful comments and suggestions have undoubtedly helped us refine and improve the quality of our paper. We sincerely appreciate your support, and we are delighted that our dual-layer framework and results on LNS have resonated with you. Thank you again for your thorough review and encouraging words!

---

> ### Author Response · Authors · 2024-11-22
> **Response to Reviewer vjA9 (4)**
>
> We provide detailed examples of the evolution process, including the changes in thought and code representations, in Appendix A of the revised manuscript. Additionally, as mentioned in A1 (response to Q1), our ablation experiments further validate the contribution of differential evolution. The results demonstrate that incorporating this mechanism significantly accelerates convergence compared to traditional methods.
>
> We hope these clarifications and examples provide a more comprehensive understanding of the differential evolution mechanism and its role in improving the results. Should you have further questions, we would be happy to address them.
>
>
>
> > **Q4:** Why are the settings for population size different between LNS tasks and combinatorial optimization tasks? How does changing the population size affect the outcomes within your framework?
>
> **A4:** Thank you for raising this question. In our experiments, the population sizes for LNS tasks and combinatorial optimization tasks are set differently to ensure fair comparisons with state-of-the-art baselines and to account for the unique characteristics of each problem type.
>
> For combinatorial optimization tasks, such as bin packing (BP) and traveling salesman problem (TSP), we adopted population sizes of 20 and 10, respectively, following the experimental settings in EOH[1]. This alignment ensures fairness in comparisons and consistency with prior work. For MILP problems, we uniformly set the population size to 4, which was determined based on the problem characteristics and computational efficiency considerations specific to large-scale MILP tasks.
>
> To further analyze the impact of population size on experimental outcomes, we conducted additional experiments on the bin packing task, testing our method with a reduced population size of 4. As shown in the table below, although the average performance slightly decreases with a smaller population size, our method still outperforms EOH (population size 20). For example, the average error of our method with a population size of 4 is **1.76%**, which is better than EOH’s **2.09%**. This demonstrates that our dual-layer framework and differential evolution mechanism exhibit significant robustness and effectiveness, maintaining superior performance even with smaller populations.
>
> |          | 1k_C100 | 5k_C100 | 10k_C100 | 1k_C500 | 5k_C500 | 10k_C500 | Avg        |
> | -------- | ------- | ------- | -------- | ------- | ------- | -------- | ---------- |
> | EOH(20)  | 4.48%   | 0.88%   | 0.83%    | 4.32%   | 1.06%   | 0.97%    | 2.09%      |
> | Ours(4)  | 3.23%   | 0.80%   | 0.43%    | 3.96%   | 1.27%   | 0.89%    | 1.76%↑     |
> | Ours(20) | 3.58%   | 0.85%   | 0.41%    | 3.67%   | 0.82%   | 0.42%    | **1.63%↑** |
>
>
>
> > **Q5:** What performance levels do FunSearch and EoH achieve on the LNS tasks? Are their results comparable to those of the proposed method?
>
> **A5:** Thank you for your question. The proposed method is a framework specifically designed for solving large-scale MILP problems, while FunSearch and EoH are optimization methods focused on discovering strategies for combinatorial optimization problems. Based on existing literature [1], EoH generally outperforms FunSearch on combinatorial optimization tasks. Therefore, we chose to apply EoH in our framework (EoH-LNS) as the primary baseline for comparison with our LLM-LNS method, ensuring representativeness and fairness.
>
> We evaluated both methods on standard combinatorial optimization problems, specifically the SC problem (a minimization task), using instances with 200,000 (SC${}_1$) and 2,000,000 (SC${}_2$) decision variables and constraints. The experimental results indicate that EoH-LNS achieves strong performance on these tasks, but our LLM-LNS method consistently outperforms EoH-LNS across all test instances. For example, on the **SC₁** dataset (200,000 variables and constraints), our method improves the average performance by **1.67%**, and on the **SC₂** dataset (2,000,000 variables and constraints), the improvement reaches **9.20%**. These results highlight the superior capability of our method in handling large-scale MILP problems.
>
> The superior performance of LLM-LNS is attributed to its dual-layer self-evolutionary mechanism and differential evolution strategy, which effectively balance diversity and convergence. This allows our method to better address the challenges posed by larger and more complex problem structures.
>
> We also plan to include additional experimental results, including tests applying FunSearch, in the appendix of a forthcoming revised manuscript. This will further validate the advantages of our method in comparison to these baselines. Thank you for your suggestion, which has provided us with valuable direction for further improving our work.

---

> ### Author Response · Authors · 2024-11-22
> **Response to Reviewer vjA9 (5)**
>
> | SC${_1}$      | Instance${_1}$ | Instance${_2}$ | Instance${_3}$ | Instance${_4}$ | Instance${_5}$ | Avg           |
> | ------------- | -------------- | -------------- | -------------- | -------------- | -------------- | ------------- |
> | EOH-LNS       | 16114.27       | 16073.72       | 16046.83       | 16074.26       | 16043.67       | 16070.15      |
> | LLM-LNS(Ours) | **15830.61↑**  | **15801.19↑**  | **15800.17↑**  | **15800.17↑**  | **15781.23↑**  | **15802.68↑** |
>
> | SC${_2}$      | Instance${_1}$ | Instance${_2}$ | Instance${_3}$ | Instance${_4}$ | Instance${_5}$ | Avg            |
> | ------------- | -------------- | -------------- | -------------- | -------------- | -------------- | -------------- |
> | EOH-LNS       | 175358.59      | 174339.78      | 174782.76      | 174026.33      | 176383.54      | 174978.20      |
> | LLM-LNS(Ours) | **158901.57↑** | **158953.57↑** | **158712.64↑** | **158759.90↑** | **159066.77↑** | **158878.89↑** |
>
>
>
> > **Q6:** What new insights have been gained from the application of LLM-designed heuristics in LNS for MILP, and how might these contribute to the field?
>
> **A6:** Thank you for this thought-provoking question. The application of LLM-designed heuristics in LNS for MILP has provided new insights, particularly in balancing **convergence** and **diversity** to achieve high-performance search algorithms. Our proposed dual-layer self-evolutionary framework effectively combines strategies for optimizing both convergence and diversity, addressing the core challenges of large-scale optimization tasks. Specifically:
>
> In optimization tasks, **convergence** ensures that the algorithm quickly finds high-quality solutions, while **diversity** prevents the algorithm from getting trapped in local optima and enables the exploration of better solutions. Our framework achieves this balance through a dual-layer design: the **outer layer** uses differential evolution to optimize prompts, focusing on enhancing the diversity of search strategies, while the **inner layer** applies differential evolution to rapidly optimize heuristic strategies, thereby accelerating convergence. The collaboration between these two layers enables the framework to deliver superior performance across problems of varying scales.
>
> Our experimental results validate this balance. As demonstrated in the ablation study discussed in the response to Q1, incorporating the differential evolution mechanism alone significantly improves convergence in small-scale problems, such as reducing the error on the **1k_C100** dataset from **4.48%** to **2.64%**. Building on this, the addition of the dual-layer structure further improves generalization on large-scale problems, reducing the error on the **10k_C500** dataset from **0.97%** to **0.39%**. This balance between convergence and diversity allows our framework to excel in LNS applications for MILP.
>
> From a methodological perspective, the LLM-driven heuristic approach not only automates the discovery of improvement directions in strategies but also captures the evolutionary pathways of strategies through a differential memory mechanism. This automation enhances the generality and adaptability of the framework, presenting a novel approach to solving large-scale MILP problems. The use of LLMs to design and optimize heuristics represents a significant shift toward leveraging pre-trained models for automated strategy generation, reducing reliance on hand-crafted heuristics and domain-specific expertise.
>
> We will provide additional experimental results and detailed analyses in the updated manuscript to further illustrate how these insights contribute to advancing the performance of search algorithms. Thank you again for raising this question, which has allowed us to more comprehensively highlight the innovations and experimental implications of our method.
>
>
> References
>
> [1] Liu F, Xialiang T, Yuan M, et al. Evolution of Heuristics: Towards Efficient Automatic Algorithm Design Using Large Language Model[C]. Forty-first International Conference on Machine Learning. 2024.

---

### Author Response · Authors · 2024-11-25
**Global Response about Manuscript**

We would like to sincerely thank all reviewers for their valuable and constructive feedback on our manuscript. Your insightful comments have greatly helped us improve the clarity, rigor, and comprehensiveness of our work. During the rebuttal phase, we have addressed the concerns raised by the reviewers, incorporated additional experiments, and strengthened the arguments supporting the effectiveness of our proposed method. We also deeply appreciate the reviewers who acknowledged the novelty and potential future impact of our approach after our initial rebuttal responses.

We are pleased to inform you that we have uploaded a revised version of the PDF manuscript, which includes substantial updates based on the feedback received. These updates are summarized in the table below. In the remaining time of the rebuttal phase, we will continue to refine and further enhance the manuscript and welcome any additional feedback from the reviewers.

| **Comment**                                                  | **Addition**                                                 | **Location**                                 |
| ------------------------------------------------------------ | ------------------------------------------------------------ | -------------------------------------------- |
| Reviewer vjA9 W1, Reviewer a92q W1                           | New overview of the proposed LLM-LNS framework               | Details in Method Section, Figure 1          |
| Reviewer vjA9 Q1, Reviewer a92q W2, Reviewer bC3Z W2, Reviewer SBNr W5 | Ablation Study of the Dual-Layer Self-evolutionary LLM Agent | Details in Appendix E                        |
| Reviewer vjA9 Q2                                             | Stability Evaluation of Multiple Runs                        | Details in Appendix F.1                      |
| Reviewer vjA9 Q4                                             | Impact of Population Size on Experimental Outcomes           | Details in Appendix F.2                      |
| Reviewer vjA9 Q5, Reviewer bC3Z W1                           | Performance Comparison with EoH on LNS Tasks                 | Details in Appendix F.3                      |
| Reviewer a92q W2                                             | Baseline Comparisons with Additional LNS Methods             | Details in Appendix D.5                      |
| Reviewer a92q W3                                             | Comprehensive Evaluation on TSPLib Instances                 | Details in Appendix F.4                      |
| Reviewer a92q Q1                                             | Datasets for Heuristic Evolution                             | Details in Appendix A.3                      |
| Reviewer a92q Q3                                             | Impact of the Backbone Algorithm on Performance              | Details in Appendix F.5                      |
| Reviewer bC3Z Q1 and Q3                                      | Experimental Settings for Algorithm Design                   | Details in Appendix A.4                      |
| Reviewer bC3Z Q5                                             | Limitations and Future Directions                            | Details in Appendix H                        |
| Reviewer SBNr W1                                             | Comprehensive revision of the Method section with additional formalizations | Details in the Method section and Appendix G |
| Reviewer SBNr W4                                             | Robustness of LLM-LNS with Different LLMs                    | Details in Appendix F.6                      |
| Reviewer SBNr W2F, Reviewer a92q Q1F                         | Comparison with ReEvo                                        | Details in Appendix F.7                      |
| Reviewer SBNr W4F                                            | Performance Comparison with EoH Using Different LLMs         | Details in Appendix F.8                      |
| Reviewer SBNr W4F                                            | Comparison with Standalone LLMs                              | Details in Appendix F.9                      |

We deeply value the reviewers' feedback and believe the revisions significantly enhance the quality and transparency of the manuscript. We look forward to addressing any further questions or comments to ensure the work meets the highest standards of clarity and rigor.

---

### Meta-Review · Area_Chair_Bebk · 2024-12-20

**Metareview:**

This paper proposed a large language model (LLM) driven large neighborhood search (LNS) method for large-scale Mixed-Integer Linear Program (MILP) problems. The core contribution is a dual-level framework, where the outer layer optimizes prompt strategies using differential evolution, while the inner layer generates heuristic strategies guided by the optimized prompts. The proposed method demonstrated good performance on three types of problems, including online bin packing, TSP, and large-scale MILPs.

Reviewers acknowledged the novelty of the proposed dual-level scheme and its good empirical performance. However, they have major concerns including 1) limited contribution since the proposed method combines existing techniques, 2) lacking of proper ablation study, and 3) unclear presentation. These concerns largely remains after rebuttal. Since 3 out of 4 reviewers are negative, I recommend rejection.

**Additional Comments On Reviewer Discussion:**

Authors provided detailed reponses to reviewers' comments, with abundant new experimental results. Unfortunately, reviewers are not fully satisfied with the responses, especially in terms of the ablation study and significance of contribution.

---

### Decision · Program_Chairs · 2025-01-22

Reject